# On Zero-Initialized Attention: Optimal Prompt and Gating Factor Estimation

**Nghiem T. Diep** [* 1 2 3]  **Huy Nguyen** [* 4]  **Chau Nguyen** [* 5 6]  **Minh Le** [5]  **Duy M. H. Nguyen** [1 7 8]  **Daniel Sonntag** [1 9]  **Mathias Niepert** [7 8]  **Nhat Ho** [4]

## Abstract

LLaMA-Adapter has recently emerged as an efficient fine-tuning technique for LLaMA models, leveraging zero-initialized attention to stabilize training and enhance performance. However, despite its empirical success, the theoretical foundations of zero-initialized attention remain largely unexplored. In this paper, we provide a rigorous theoretical analysis, establishing a connection between zero-initialized attention and mixture-of-expert models. We prove that both linear and non-linear prompts, along with gating functions, can be optimally estimated, with non-linear prompts offering greater flexibility for future applications. Empirically, we validate our findings on the open LLM benchmarks, demonstrating that non-linear prompts outperform linear ones. Notably, even with limited training data, both prompt types consistently surpass vanilla attention, highlighting the robustness and adaptability of zero-initialized attention. Our implementation is publicly available on GitHub.

## 1. Introduction

Large Language Models (LLMs) have revolutionized natural language processing (Radford et al., 2019; Raffel et al., 2020; Anil et al., 2023; Minaee et al., 2024), demonstrating remarkable understanding and generative capabilities across various tasks. While proprietary models like ChatGPT (OpenAI, 2025) and GPT-4 (OpenAI et al., 2024) set new records in instruction-following performance (Peng et al., 2023), their closed-source nature and high development costs limit accessibility. To address this, several efforts have explored fine-tuning open-source LLMs, e.g., LLaMA (Touvron et al., 2023a), on large-scale instruction datasets where small human-annotated samples are expanded into massive training corpora using self-instruct methods (Taori et al., 2023). This approach has enabled models like LLaMA to achieve instruction-following performance comparable to GPT-3.5 and, with further advancements, shows promise in approaching GPT-4's capabilities. However, full fine-tuning of large-scale LLMs remains computationally intensive, highlighting the need for more efficient adaptation techniques to unlock their full potential across diverse downstream applications.

To address this challenge, researchers have increasingly turned to parameter-efficient fine-tuning (PEFT) techniques, which update only a small subset of parameters while keeping most of the backbone frozen, enabling efficient and scalable adaptation (Houlsby et al., 2019; Lester et al., 2021; Hu et al., 2021). Among these approaches, the LLaMA-Adapter (Zhang et al., 2024) has emerged as a specialized PEFT method for transforming LLaMA into an instruction-following model, demonstrating strong performance across multiple benchmarks. Beyond adaptation prompts, its key innovation lies in the *introduction of zero-initialized attention with zero gating*, which facilitates the seamless integration of new instructional prompts while preserving the model's pre-existing knowledge. However, despite its empirical success, the theoretical foundations of zero-initialized attention remain largely unexplored, limiting a deeper understanding of its underlying mechanisms and potential for further advancements.

In this work, we conduct a rigorous theoretical and empirical investigation into zero-initialized attention, showing that it is not merely an engineering trick. Building on recent findings (Le et al., 2024; 2025) that draw connections between the attention mechanism (Vaswani, 2017) and the mixture of experts (MoE) model (Jacobs et al., 1991; Jordan & Jacobs, 1994; Shazeer et al., 2017), we investigate how zero-initialized attention in the LLaMA-Adapter (Zhang et al., 2024) can be interpreted within this framework. Specifically, we show that *zero-initialized attention can be viewed as a special formulation of the MoE model* and that under lin-

---

[*]Equal contribution  [1]German Research Center for Artificial Intelligence (DFKI) [2]University of Science, VNU-HCM, Ho Chi Minh City, Vietnam [3]Viet Nam National University, Ho Chi Minh City, Vietnam [4]The University of Texas at Austin [5]Qualcomm AI Research, an initiative of Qualcomm Technologies, Inc. [6]Work was completed while an employee at Qualcomm [7]Max Planck Research School for Intelligent Systems (IMPRS-IS) [8]University of Stuttgart [9]Oldenburg University. Correspondence to: Huy Nguyen <huynm@utexas.edu>.

*Proceedings of the 42nd International Conference on Machine Learning*, Vancouver, Canada. PMLR 267, 2025. Copyright 2025 by the author(s).

ear prompt settings, both the prompts and the gating factor can be optimally estimated, offering significant statistical benefits. Furthermore, we *extend this analysis to scenarios in which the trainable prompts are nonlinear (e.g., implemented via an MLP)* and prove that the optimal estimation of the prompts and gating parameters remains achievable, thus enhancing the flexibility for future applications.

Our empirical evaluations further substantiate these theoretical findings. Across multiple datasets in the open LLM benchmark (Beeching et al., 2024), zero-initialized attention consistently demonstrates better performance than random-initialized attention. Additionally, non-linear prompts exhibit improved results compared to their linear counterparts, thereby offering the potential to boost the LLaMA-Adapter's capabilities. Notably, on various small rates of limited training data, both linear and non-linear prompts combined with zero-initialized attention substantially outperform standard attention, reinforcing our theoretical findings and demonstrating the robustness of the proposed approach.

**Contribution.** Our contributions are four-fold can be summarized as follows: **(i)** We develop a theoretical framework that examines the connection between zero-initialized attention and MoE models; **(ii)** We demonstrate the statistical advantages of zero-initialized attention over conventional attention, enabling optimal estimation of prompt parameters and gating factors; **(iii)** We extend our analysis to accommodate non-linear prompts, offering increased flexibility for a broader range of applications; **(iv)** Finally, extensive experiments on multiple question-answering datasets validate our theoretical insights, highlighting the robustness and significance of our findings.

**Organization.** The remainder of this paper is organized as follows: Section 2 reviews related work. In Section 3, we establish the connection between zero-initialized attention and the mixture-of-experts model. Section 4 presents our theoretical results. In Section 4.2, we extend the analysis for the non-linear setting and outline the main algorithm for the non-linear case, followed by empirical evaluations in Section 5. Finally, Section 6 discusses limitations and future research directions. Detailed proofs and additional experimental results are provided in the Appendix.

**Notation.** For any $n \in \mathbb{N}$, we denote $[n]$ as the set $\{1, 2, \ldots, n\}$. For any set $S$, we let $|S|$ be its cardinality. For any vectors $u := (u_1, u_2, \ldots, u_d) \in \mathbb{R}^d$ and $\alpha := (\alpha_1, \alpha_2, \ldots, \alpha_d) \in \mathbb{N}^d$, we let $u^\alpha = u_1^{\alpha_1} u_2^{\alpha_2} \ldots u_d^{\alpha_d}$, $|u| := u_1 + u_2 + \ldots + u_d$, and $\alpha! := \alpha_1! \alpha_2! \ldots \alpha_d!$, while $\|u\|$ denotes the $\ell_2$-norm value of $u$. Lastly, for any two positive sequences $(a_n)_{n \geq 1}$ and $(b_n)_{n \geq 1}$, we write $a_n = \mathcal{O}(b_n)$ or $a_n \lesssim b_n$ if $a_n \leq C b_n$ for all $n \in \mathbb{N}$, where $C > 0$ is some constant. Finally, $a_n = \mathcal{O}_P(b_n)$ means $a_n/b_n$ is stochastically bounded.

## 2. Related Work

**Parameter Efficient Fine-Tuning for LLMs.** Large foundational models have demonstrated remarkable generalization capabilities across a wide range of tasks (Devlin, 2018; Brown et al., 2020; Touvron et al., 2023b;c). However, as model sizes grow exponentially, fine-tuning all parameters in a large-scale model becomes increasingly impractical. In contrast, PEFT techniques (Hu et al., 2021; Karimi Mahabadi et al., 2021; Gao et al., 2024; Houlsby et al., 2019; Li et al., 2024c; Li & Liang, 2021; Zhou et al., 2022; Mangrulkar et al., 2022) have emerged as a promising strategy to adapt these models to downstream tasks while freezing most of the backbone's parameters. These approaches can be categorized into three directions, namely (i) *low-rank decomposition* like LoRA (Hu et al., 2021) and its advanced versions (Karimi Mahabadi et al., 2021; Zhang et al., 2023); (ii) *adapters* which insert lightweight adaptation modules into each block of the transformer and have been applied across numerous domains (Gao et al., 2024; Houlsby et al., 2019; Li et al., 2024c); and (iii) *prompt tuning* where trainable tokens are appended to the input embeddings (Lester et al., 2021; Nguyen et al., 2024b) or at intermediate layers, i.e., pre-fixed tuning (Liu et al., 2021; Li & Liang, 2021; Shi & Lipani, 2023).

Unlike the aforementioned PEFT methods, the LLaMA-Adapter (Zhang et al., 2024) is specifically designed to enhance instruction-following capabilities, where the model learns to generate contextually relevant responses based on natural language instructions. This is done by introducing a concept of zero-initialized attention to integrate new instructional prompts while preserving the model's existing knowledge. Through this mechanism, the algorithm starts with minimal impact and prevents training instability and catastrophic forgetting by selectively activating relevant information while allowing the model to incorporate instructions incrementally. In this work, we conduct a comprehensive theoretical and empirical investigation into zero-initialized attention, demonstrating that it is more than just an engineering trick and uncovering its fundamental properties and advantages.

**Mixture of Experts (MoE) in PEFT.** Recent research has explored MoE in PEFT to enhance the adaptability of large pre-trained models while minimizing computational costs. MoE-based approaches, such as Switch Transformers (Fedus et al., 2022) leveraged sparse activation of expert networks to achieve efficient scaling. In the context of PEFT, techniques like AdapterDrop (Rücklé et al., 2020) and LoRA (Hu et al., 2021) have been combined with MoE to dynamically allocate resources to task-specific experts, reducing the number of trainable parameters (Li et al., 2024a; Chen et al., 2024; 2023). These works demonstrate that MoE can significantly enhance parameter efficiency without compro-

mising performance, making it a promising direction for fine-tuning large-scale models.

In another line of research, the MoE framework has been leveraged in (Le et al., 2024; 2025) to investigate the convergence behavior of learnable prompt vectors in the context of prefix tuning method, which are attached to the key and value matrices of self-attention mechanism to learn downstream tasks. In particular, by showing rigorously that each row of an attention head can be represented as an MoE, they demonstrate theoretically and empirically that the prompt convergence will be significantly accelerated if there exists a shared structure among the prompt vectors. However, although the zero-initialized attention has been widely used as an PEFT method, its theoretical understanding has remained missing in the literature. To close this gap, we provide a comprehensive study on the convergence of prompt vectors within the zero-initialized attention by establishing a connection between this model and MoE in Section 3. Our theory indicates that linear prompts and non-linear prompts share the same convergence behavior and can be both optimally estimated. On the empirical side, we observe that the non-linear prompts in zero-initalized attention work favorably compared to linear prompts in several benchmark datasets.

Additional discussion on related work of the theory of mixture of experts is in Appendix B.

## 3. Zero-initialized Attention meets Mixture of Experts

**Zero-initialized attention.** We revisit the zero-initialized attention formulation introduced by Zhang et al. (2024). Let $\mathbf{X}_l = [\boldsymbol{x}_{l,1}, \ldots, \boldsymbol{x}_{l,N}]^\top \in \mathbb{R}^{N \times d}$ denote the input tokens at the $l$-th layer of LLaMA's transformer model, where $N$ is the input sequence length, and $d$ represents the feature dimensionality. Similarly, let $\mathbf{P}_l = [\boldsymbol{p}_{l,1}, \ldots, \boldsymbol{p}_{l,L}]^\top \in \mathbb{R}^{L \times d}$ denote the learnable adaptation prompt used to fine-tune the model. This prompt is concatenated with $\mathbf{X}_l$ along the token dimension, serving as a prefix, where $L$ denotes the prompt length. For simplicity, we omit the subscript $l$ in subsequent notations.

Suppose the model is generating the $(N+1)$-th word based on $(\mathbf{P}, \mathbf{X})$ at the $l$-th layer. We denote the corresponding $(N+1)$-th word token as $\boldsymbol{t} \in \mathbb{R}^d$. Within the attention mechanism, linear projection layers are applied to the input tokens, transforming them into queries, keys, and values, defined as follows:

$$\boldsymbol{Q} = \boldsymbol{t}^\top W^Q \in \mathbb{R}^{1 \times d_k}, \tag{1}$$

$$\boldsymbol{K} = \left[\mathbf{P}^\top, \mathbf{X}^\top, \boldsymbol{t}\right]^\top W^K \in \mathbb{R}^{(L+N+1) \times d_k}, \tag{2}$$

$$\boldsymbol{V} = \left[\mathbf{P}^\top, \mathbf{X}^\top, \boldsymbol{t}\right]^\top W^V \in \mathbb{R}^{(L+N+1) \times d_v}, \tag{3}$$

where $W^Q \in \mathbb{R}^{d \times d_k}$, $W^K \in \mathbb{R}^{d \times d_k}$, and $W^V \in \mathbb{R}^{d \times d_v}$

are pre-trained projection matrices. The attention scores computed between $\boldsymbol{Q}$ and $\boldsymbol{K}$ before applying the softmax function are given by:

$$\boldsymbol{S} = \frac{\boldsymbol{Q}\boldsymbol{K}^\top}{\sqrt{d_k}} = [\boldsymbol{S}_P, \ \boldsymbol{S}_X] \in \mathbb{R}^{1 \times (L+N+1)}, \tag{4}$$

where $\boldsymbol{S}_P \in \mathbb{R}^{1 \times L}$ and $\boldsymbol{S}_X \in \mathbb{R}^{1 \times (N+1)}$ denote the attention scores of $L$ adaption prompts and $N+1$ word tokens, respectively. Rather than applying the softmax function directly, Zhang et al. (2024) propose an alternative approach. They suggest computing the softmax independently over the two components $\boldsymbol{S}_P$ and $\boldsymbol{S}_X$, and incorporating a learnable gating factor $\alpha \in \mathbb{R}$ as follows:

$$\boldsymbol{S}_g = [\text{softmax}(\boldsymbol{S}_P) \cdot \tanh(\alpha), \ \text{softmax}(\boldsymbol{S}_X)]. \tag{5}$$

This approach aims to decouple the knowledge contained within the pre-trained model and the adaptation prompts, thereby preserving the pre-trained model's original knowledge. The activation function $\tanh(\cdot)$ is used to regulate the scale of $\alpha$ within the range $[-1, 1]$. Consequently, the output of the zero-initialized attention mechanism can be expressed as:

$$\mathbf{y} = \boldsymbol{S}_g \cdot \boldsymbol{V} \in \mathbb{R}^{d_v}. \tag{6}$$

During training, only the prompt parameters $\mathbf{P}$ are optimized, while all other parameters of the pre-trained model remain frozen. Next, we examine how zero-initialized attention can be interpreted through the lens of the mixture of experts framework.

**Connection to mixture of experts.** Recent studies (Le et al., 2024; 2025) have revealed a notable connection between the attention mechanisms and the mixture of experts (MoE) architectures (Jacobs et al., 1991; Jordan & Jacobs, 1994), showing that attention can be seen as a form of MoE, where attention weights act as gating functions over token interactions. This MoE perspective provides a useful framework for analyzing zero-initialized attention by viewing its components as gates and expert responses.

Specifically, let $\boldsymbol{X} = \left[\boldsymbol{x}_1^\top, \ldots, \boldsymbol{x}_N^\top, \boldsymbol{t}^\top\right]^\top \in \mathbb{R}^{(N+1)d}$, which is the concatenation of input tokens. For $i \in [N+1]$, define $E_i \in \mathbb{R}^{d \times (N+1)d}$ such that $E_i \boldsymbol{X} = \boldsymbol{x}_i$ for $i = 1, \ldots, N$, and $E_{N+1} \boldsymbol{X} = \boldsymbol{t}$. We then introduce a set of $L+N+1$ experts $f_j : \mathbb{R}^{(N+1)d} \to \mathbb{R}^{d_v}$, defined as:

$$f_j(\boldsymbol{X}) = W^{V^\top} E_j \boldsymbol{X} = W^{V^\top} \boldsymbol{x}_j, \ j \in [N] \tag{7}$$

$$f_{N+1}(\boldsymbol{X}) = W^{V^\top} E_{N+1} \boldsymbol{X} = W^{V^\top} \boldsymbol{t} \tag{8}$$

$$f_{N+1+j'}(\boldsymbol{X}) = W^{V^\top} \boldsymbol{p}_{j'}, \ j' \in [L]. \tag{9}$$

Based on equation (5), the weights $G_j : \mathbb{R}^{(N+1)d} \to \mathbb{R}$ associated with each expert are defined as follows:

$$G_j(\boldsymbol{X}) = \frac{\exp(\frac{\boldsymbol{X}^\top E_{N+1}^\top W^Q W^{K\top} E_j \boldsymbol{X}}{\sqrt{d_k}})}{\sum_{k=1}^{N+1} \exp(\frac{\boldsymbol{X}^\top E_{N+1}^\top W^Q W^{K\top} E_k \boldsymbol{X}}{\sqrt{d_k}})},$$

$$G_{N+1+j'}(\boldsymbol{X}) = \frac{\exp(\frac{\boldsymbol{X}^\top E_{N+1}^\top W^Q W^{K\top} \boldsymbol{p}_{j'}}{\sqrt{d_k}})}{\sum_{k'=1}^{L} \exp(\frac{\boldsymbol{X}^\top E_{N+1}^\top W^Q W^{K\top} \boldsymbol{p}_{k'}}{\sqrt{d_k}})},$$

for $j' \in [L]$ and $j \in [N+1]$. With these formulations, the output of zero-initialized attention, as described in equation (6), can be expressed as:

$$\mathbf{y} = \sum_{j=1}^{N+1} G_j(\boldsymbol{X}) \cdot f_j(\boldsymbol{X}) + \tanh(\alpha)$$
$$\times \left( \sum_{j'=1}^{L} G_{N+1+j'}(\boldsymbol{X}) \cdot f_{N+1+j'}(\boldsymbol{X}) \right). \quad (10)$$

From this formulation, zero-initialized attention can be interpreted as a specialized form of a mixture of experts model. The set of experts $f_1, \ldots, f_{N+1}$, along with their associated weight functions, are pre-trained and require no additional training, as their parameters are encoded within the pre-trained transformer model, representing existing knowledge. In contrast, the prompt experts $f_{N+2}, \ldots, f_{N+1+L}$ and their weight functions work in conjunction with the pre-trained experts, effectively integrating newly acquired information into the model through learnable prompts. Viewing zero-initialized attention through this lens as a specialized mixture of experts model provides the foundation for further theoretical analysis, as demonstrated in the next section.

## 4. Convergence Analysis of Prompt and Gating Factor Estimations

In this section, we provide a theoretical analysis for linear prompts and non-linear prompts through the connection between the zero-initialized attention and the mixture of experts in equation (10), demonstrating the sample-efficiency of the zero-initalized attention over the random-initalized attention. In addition, the theoretical benefits of non-linear prompts offer great flexibility in improving the practical performance of the zero-initialized attention, which will be investigated extensively with several benchmark datasets and tasks in Section 5.

### 4.1. Analytics for Linear Prompts

We first consider the original setting of zero-initialized attention when the prompts are linear.

**Problem setting.** Assume that $(\boldsymbol{X}_1, Y_1), (\boldsymbol{X}_2, Y_2), \ldots,$ $(\boldsymbol{X}_n, Y_n) \in \mathbb{R}^d \times \mathbb{R}^{d'}$ are i.i.d. samples of size $n$ generated from the following regression model:

$$Y_i = f_{G_*, \alpha_*}(\boldsymbol{X}_i) + \varepsilon_i, \quad i = 1, 2, \ldots, n, \quad (11)$$

where the variables $\varepsilon_1, \ldots, \varepsilon_n$ are independent Gaussian noise satisfying $\mathbb{E}[\varepsilon_i|\boldsymbol{X}_i] = 0$ and $\mathrm{Var}(\varepsilon_i|\boldsymbol{X}_i) = \sigma^2 I_{d'}$ for all $i \in [n]$. Additionally, we assume that $\boldsymbol{X}_1, \boldsymbol{X}_2, \ldots, \boldsymbol{X}_n$ are i.i.d. samples from some probability distribution $\mu$. The regression function $f_{G_*, \alpha_*}(\cdot)$ in equation (11) takes the form of the MoE model with $N$ pre-trained experts and $L$ unknown experts, which is given by:

$$f_{G_*, \alpha_*}(\boldsymbol{X}) := \sum_{j=1}^{N} \frac{\exp(\boldsymbol{X}^\top \bar{A}_j^0 \boldsymbol{X} + \bar{a}_j^0)}{\sum_{k=1}^{N} \exp(\boldsymbol{X}^\top \bar{A}_k^0 \boldsymbol{X} + \bar{a}_k^0)} \cdot h(\boldsymbol{X}, \bar{\eta}_j^0)$$
$$+ \tanh(\alpha_*) \cdot \sum_{j'=1}^{L} \frac{\exp((\bar{B}\boldsymbol{p}_{*,j'})^\top \boldsymbol{X} + \bar{b}_{*,j'})}{\sum_{k'=1}^{L} \exp((\bar{B}\boldsymbol{p}_{*,k'})^\top \boldsymbol{X} + \bar{b}_{*,k'})} \cdot \bar{C}\boldsymbol{p}_{*,j'},$$
$$(12)$$

where $G_* := \sum_{j'=1}^{L} \exp(\bar{b}_{*,j'})\delta_{\boldsymbol{p}_{*,j'}}$ denotes a true but unknown *mixing measure*, which is a weighted sum of Dirac measures $\delta$, associated with unknown biases and prompts $(\bar{b}_{*,j'}, \boldsymbol{p}_{*,j'})_{j'=1}^{L}$ in the parameter space $\Theta \subset \mathbb{R} \times \mathbb{R}^d$. Furthermore, the *gating factor* $\alpha^*$ is unknown and belongs to the parameter space $\Omega \subset \mathbb{R}$. In the model (12), the matrices $\bar{A}_j^0$, the expert parameters $\bar{\eta}_j^0$, and the bias parameters $\bar{a}_j^0$ are known for all $j \in [N]$. Finally, $\bar{B} \in \mathbb{R}^{d \times d}$ and $\bar{C} \in \mathbb{R}^{d' \times d}$ are given and they play the role of pre-trained projection matrices in the context of zero-initialized attention.

**Least-square estimator.** We can estimate the unknown prompts and gating factor in the regression model (11) via estimating the mixing measure $G_*$ using least-square method as follows:

$$(\widehat{G}_n, \widehat{\alpha}_n) := \underset{G \in \mathcal{G}_{L'}(\Theta), \alpha \in \Omega}{\arg\min} \sum_{i=1}^{n} \|Y_i - f_{G, \alpha}(\boldsymbol{X}_i)\|^2, \quad (13)$$

where $\mathcal{G}_{L'}(\Theta) := \{G = \sum_{i=1}^{\ell} \exp(\bar{b}_i)\delta_{\boldsymbol{p}_i} : 1 \leq \ell \leq L', (b_i, \boldsymbol{p}_i) \in \Theta\}$ denotes the set of all mixing measures with at most $L'$ prompts. For practical purpose, the number of chosen prompts $L'$ is generally larger than the number of true prompts $L$, i.e., $L' \geq L$, to guarantee that the estimated prompts and gating factor from the least-square method converge to the true prompts and gating factor.

**Convergence rate of regression estimator.** We first show that the regression estimator $f_{\widehat{G}_n, \widehat{\alpha}_n}$ can still estimate the true regression function $f_{G_*, \alpha_*}$ at the standard parametric rate in terms of the sample size $n$ though we overspecify the number of prompts, i.e., $L' > L$.

**Proposition 4.1.** *The convergence rate of the regression estimator $f_{\widehat{G}_n, \widehat{\alpha}_n}(\cdot)$ to the true regression function $f_{G_*, \alpha_*}(\cdot)$ under the $L^2(\mu)$ norm is of parametric order, that is,*

$$\|f_{\widehat{G}_n, \widehat{\alpha}_n} - f_{G_*, \alpha_*}\|_{L^2(\mu)} = \mathcal{O}_P(\sqrt{\log(n)/n}). \quad (14)$$

Proof of Proposition 4.1 is in Appendix A.3. Given the above convergence rate of the regression estimator, we aim to construct a loss function among parameters, denoted by $\mathcal{D}(\widehat{G}_n, G_*)$, such that $\|f_{\widehat{G}_n, \widehat{\alpha}_n} - f_{G_*, \alpha_*}\|_{L^2(\mu)} \gtrsim [\mathcal{D}(\widehat{G}_n, G_*) + |\widehat{\alpha}_n - \alpha_*|]$. Then, this lower bound together with the bound (14) will lead to our desired prompt convergence rates. To this end, we will build a loss function based on the concept of Voronoi cells as in (Manole & Ho, 2022).

**Voronoi loss function.** For a mixing measure $G \in \mathcal{G}_{L'}(\Theta)$, we distribute its atoms across the Voronoi cells $\{\mathcal{C}_j \equiv \mathcal{C}_j(G), j \in [L]\}$ generated by the atoms of $G_*$, where

$$\mathcal{C}_j := \{i \in [L'] : \|\boldsymbol{p}_i - \boldsymbol{p}_{*,j}\| \leq \|\boldsymbol{p}_i - \boldsymbol{p}_{*,\ell}\|, \forall \ell \neq j\}.$$

Then, the Voronoi loss function is given by

$$\mathcal{D}(G, G_*) := \sum_{j'=1}^{L} \Big| \sum_{i \in \mathcal{C}_{j'}} \exp(b_i) - \exp(b_{*,j'}) \Big|$$
$$+ \sum_{j' \in [L]:|\mathcal{C}_{j'}|=1} \sum_{i \in \mathcal{C}_{j'}} \exp(b_i)\|\Delta \boldsymbol{p}_{ij'}\|$$
$$+ \sum_{j' \in [L]:|\mathcal{C}_{j'}|>1} \sum_{i \in \mathcal{C}_{j'}} \exp(b_i)\|\Delta \boldsymbol{p}_{ij'}\|^2, \quad (15)$$

where $\Delta \boldsymbol{p}_{ij'} = \boldsymbol{p}_i - \boldsymbol{p}_{*,j'}$ for all $i \in \mathcal{C}_{j'}$ and $j' \in [L]$. Given the above loss function, we are now ready to capture the convergence behavior of linear prompts in Theorem 4.2.

**Theorem 4.2.** *Assume that $L' > L$, i.e., the number of prompts is unknown and is overspecified by $L'$ prompts. Then, the least square estimator $(\widehat{G}_n, \widehat{\alpha}_n)$ defined in equation (13) satisfies*

$$\mathcal{D}(\widehat{G}_n, G_*) = \mathcal{O}_P(\sqrt{\log(n)/n}),$$
$$|\widehat{\alpha}_n - \alpha_*| = \mathcal{O}_P(\sqrt{\log(n)/n}).$$

Proof of Theorem 4.2 is in Appendix A.1. Putting the first bound and the formulation of the Voronoi loss function $\mathcal{D}$ together, we observe that the convergence rates of estimating linear prompts range from the order $\mathcal{O}_P([\log(n)/n]^{\frac{1}{2}})$ to $\mathcal{O}_P([\log(n)/n]^{\frac{1}{4}})$, which are optimal. Therefore, we need a polynomial number of data, either $\mathcal{O}(\epsilon^{-2})$ or $\mathcal{O}(\epsilon^{-4})$, to approximate the linear prompts with a given error $\epsilon$.

**Sample complexity of random-initialized attention v.s. zero-initialized attention:** From the above the convergence analysis of prompt estimation under the zero-initialized attention and that under the random-initialized attention in (Akbarian et al., 2024; Le et al., 2024), we claim that using the zero-initialized attention is more sample-efficient than using the random-initialized attention in terms of prompt convergence due to the following reasons:

*(i) Prompt convergence in random-initialized attention*: The convergence rates of prompt estimation are significantly slow, standing at the order of $\mathcal{O}_P(1/\log^\tau(n))$ for some constant $\tau > 0$, where $n$ is the sample size. Thus, to approximate prompts with a given error $\epsilon$, we need exponentially many data points $\mathcal{O}(\exp(\epsilon^{-1/\tau}))$, which is not sample-efficient.

*(ii) Prompt convergence in zero-initialized attention*: As shown in Theorem 4.2, the convergence rates of linear prompt estimations are of polynomial orders, ranging from $\mathcal{O}_P(n^{-1/2})$ to $\mathcal{O}_P(n^{-1/4})$ (up to some logarithmic term), which are substantially faster than those under the random-initialized attention. Therefore, we only need polynomially many data points $\mathcal{O}(\epsilon^{-2})$ or $\mathcal{O}(\epsilon^{-4})$ to approximate the prompts with a given error $\epsilon$.

### 4.2. Theoretical Benefits of Non-Linear Prompts

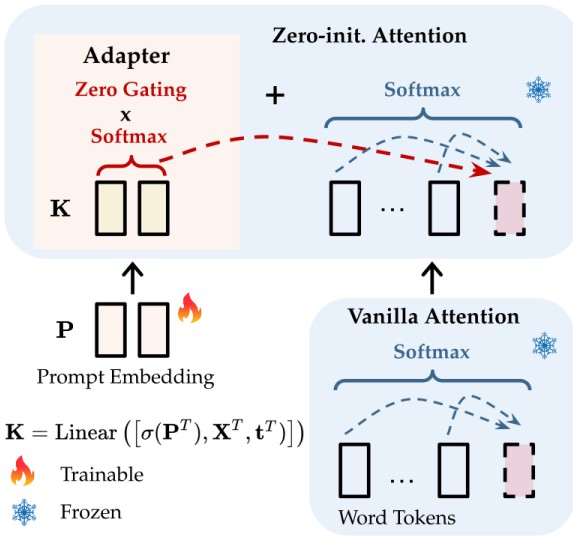

*Figure 1.* LLaMA-Adapter with non-linear prompt structures. Trainable prompts are integrated into the final layers of the LLaMA model, where a zero-gating mechanism modulates the added prompts. This approach enables progressive learning of instructional knowledge while keeping the remaining model parameters frozen.

While the original zero-initialization approach considered only linear prompts (Zhang et al., 2024), most current prompt-based techniques commonly reparameterize the prompt parameters with an MLP rather than optimizing them directly, in order to enhance training stability (Li & Liang, 2021; Liu et al., 2021; Le et al., 2025). To increase both the flexibility and practical relevance of our results, we extend our analysis to zero-initialized attention equipped with non-linear prompts.

**Problem setting.** Suppose that the data $(\boldsymbol{X}_1, Y_1), (\boldsymbol{X}_2, Y_2), \ldots, (\boldsymbol{X}_n, Y_n) \in \mathbb{R}^d \times \mathbb{R}^{d'}$ are

i.i.d. samples of size $n$ generated from the model:

$$Y_i = g_{G_*,\alpha_*}(\boldsymbol{X}_i) + \varepsilon_i, \quad i = 1, 2, \ldots, n. \quad (16)$$

Here, we impose the same assumptions on the noise variables $\varepsilon_i$ and the input $\boldsymbol{X}_i$ as in Section 4.1. Nevertheless, the regression function $g_{G_*,\alpha_*}(\cdot)$ in equation (16) now takes the form of a prefix MoE model with $N$ pre-trained experts and $L$ unknown experts,

$$g_{G_*,\alpha_*}(\boldsymbol{X}) := \sum_{j=1}^{N} \frac{\exp(\boldsymbol{X}^\top \bar{A}_j^0 \boldsymbol{X} + \bar{a}_j^0)}{\sum_{k=1}^{N} \exp(\boldsymbol{X}^\top \bar{A}_k^0 \boldsymbol{X} + \bar{a}_k^0)} \cdot h(\boldsymbol{X}, \bar{\eta}_j^0)$$

$$+ \tanh(\alpha_*)$$

$$\times \left( \sum_{j'=1}^{L} \frac{\exp((\bar{B}\sigma(\boldsymbol{p}_{*,j'}))^\top \boldsymbol{X} + \bar{b}_{*,j'})}{\sum_{k'=1}^{L} \exp((\bar{B}\sigma(\boldsymbol{p}_{*,k'}))^\top \boldsymbol{X} + \bar{b}_{*,k'})} \cdot \bar{C}\sigma(\boldsymbol{p}_{*,j'}) \right),$$

$$(17)$$

where $\sigma : \mathbb{R}^d \to \mathbb{R}^{d'}$ is some activation function applied element-wise to the prompts $\boldsymbol{p}_{*,j'}$. According to the change of the regression function, the least-square estimator of the true mixing measure $G_*$ under this setting becomes

$$(\widetilde{G}_n, \widetilde{\alpha}_n) := \underset{G \in \mathcal{G}_{L'}(\Theta), \alpha \in \Omega}{\arg\min} \sum_{i=1}^{n} \left( Y_i - g_{G,\alpha}(\boldsymbol{X}_i) \right)^2. \quad (18)$$

In the following proposition, we will illustrate that the parametric convergence rate of the regression function estimator still holds true under the setting of non-linear prompts.

**Proposition 4.3.** *The convergence rate of the model estimation $g_{\widetilde{G}_n, \widetilde{\alpha}_n}(\cdot)$ to the true model $g_{G_*,\alpha_*}(\cdot)$ under the $L^2(\mu)$ norm is parametric on the sample size, that is,*

$$\|g_{\widetilde{G}_n, \widetilde{\alpha}_n} - g_{G_*,\alpha_*}\|_{L^2(\mu)} = \mathcal{O}_P(\sqrt{\log(n)/n}). \quad (19)$$

Similar to Section 4.1, by utilizing the Voronoi loss function $\mathcal{D}$ defined in Eq.(15), we are able to investigate the convergence behavior of non-linear prompts in the zero-initialized attention in Theorem 4.4 whose proof can be found in Appendix A.2.

**Theorem 4.4.** *Assume that the activation function $\sigma$ satisfies the Assumptions (A.1)-(A.2) specified in Appendix A.2. Then, the least square estimator $(\widetilde{G}_n, \widetilde{\alpha}_n)$ defined in equation (18) satisfies*

$$\mathcal{D}(\widetilde{G}_n, G_*) = \mathcal{O}_P(\sqrt{\log(n)/n}),$$
$$|\widetilde{\alpha}_n - \alpha_*| = \mathcal{O}_P(\sqrt{\log(n)/n}).$$

It can be seen that the convergence of prompt parameters under this setting behaves analogously to that in Theorem 4.2. In particular, the prompt parameters $\boldsymbol{p}_{*,j}$ still admit the estimation rates of order $\mathcal{O}_P([\log(n)/n]^{\frac{1}{2}})$ or $\mathcal{O}_P([\log(n)/n]^{\frac{1}{4}})$. Thus, a polynomial number of data points ranging from $\mathcal{O}(\varepsilon^{-2})$ to $\mathcal{O}(\varepsilon^{-4})$ is required to

achieve the prompt approximation with a given error $\epsilon$. As a result, we claim that the zero-initialized attention with non-linear prompts is also more sample-efficient than the random-initialized attention in terms of prompt convergence. Moreover, although sharing the same sample complexity as when using linear prompts, the zero-initialized attention with non-linear prompts will be shown to offer greater flexibility in practical applications, which will be discussed further in Section 5.

**Non-Linear prompt optimization in practice.** Motivated by the theoretical benefits of using non-linear prompts in zero-initialized attention in Eq.(17), one replaces the linear prompts $\mathbf{P}$ in zero-initialized attention with non-linear prompts (Figure 1), which is given by:

$$\tilde{\mathbf{P}} = \sigma(\mathbf{P}) \in \mathbb{R}^{L \times d}, \quad (20)$$

where $\sigma(\cdot)$ is a non-linear activation function or a lightweight MLP, in line with common practice in most current prompt-based techniques, and $\mathbf{P}$ can be a layer embedding vector or a set of embedding vectors equal to the length of prompt for each layer. For instance, we can choose $\sigma$ as a lightweight MLP with 2 layers combined with non-linear activation such as Tanh, ReLU, and Leaky-ReLU:

$$\sigma(\mathbf{P}) = f_2(\phi((f_1(\mathbf{P})))), \quad (21)$$

where $f_1(.)$, $f_2(.)$ are separate linear transformations, $\phi(.)$ represents the non-linear activation function, e.g., ReLU or Leaky-ReLU, and $\mathbf{P}$ is defined as a layer embedding vector. To ensure parameter efficiency and facilitate knowledge sharing across layers, this MLP can be shared among the layers that utilize the prompts.

As established in Theorem 4.4, this non-linear formulation retains the same estimation rates as the linear variant, thus offering greater flexibility for practical applications.

## 5. Experiments

To highlight the statistical advantages of zero-initialized attention and explore the potential of non-linear prompts, we conduct a series of question-answering experiments on LLM tasks. Section 5.1 provides an overview of our experimental setup, while the main results are presented in Section 5.2. Additional details and prompt templates are included in Appendix C.

### 5.1. Experimental Setup

**Datasets and Evaluations.** We use the Open LLM benchmarks as in Beeching et al. (2024). These benchmarks evaluate the generative abilities of LLMs in four different tasks, including (i) AI2 Reasoning Challenge (ARC) with Easy (eas) and Challenge (cha) types (Clark et al., 2018), (ii) HellaSwag (Zellers et al., 2019), (iii)

*Table 1.* Commparison between **Linear prompt** (zero-initialized mechanism) and **Random-Init** prompt on 4 LLM tasks using LLaMA-7B and LLaMA-13B models.

| Method | ARC | | | MMLU | Hellaswag | TruthfullQA | Average |
|---|---|---|---|---|---|---|---|
| | *Acc (eas)* | *Acc (cha)* | *Acc (aver)* | *Acc* | *Acc* | *Acc* | |
| LLaMA-7B + zero-init | 62.29 ↑ 1.64 | 43.17 ↑ 2.47 | 52.73 ↑ 2.06 | 36.28 ↑ 1.16 | 76.79 ↑ 4.17 | 45.53 ↑ 7.71 | 52.83 ↑ 3.77 |
| LLaMA-7B + rand-init | 60.65 | 40.7 | 50.67 | 35.12 | 72.62 | 37.82 | 49.06 |
| LLaMA-13B + zero-init | 81.78 ↑ 0.17 | 64.33 ↑ 0.42 | 73.06 ↑ 0.3 | 49.64 ↑ 1.62 | 81.21 ↑ 0.05 | 34.88 ↑ 0.36 | 59.70 ↑ 0.58 |
| LLaMA-13B + rand-init | 81.61 | 63.91 | 72.76 | 48.02 | 81.16 | 34.52 | 59.12 |

MMLU (Hendrycks et al., 2020), and (iv) TruthfulQA (Lin et al., 2021). All these tasks evaluate the model through multiple-choice questions, where ARC and MMLU test the LLM's knowledge, HellaSwag tests the model's ability to finish sentences, and TruthfulQA measures whether an LLM is truthful in generating answers to given questions.

We follow the experimental setup of LLaMA-Adapter (Zhang et al., 2024) by fine-tuning LLaMA on the Alpaca dataset (Taori et al., 2023). The model performance is evaluated on the test set by conducting a zero-shot evaluation for ARC, MMLU, and TruthfulQA while using a 10-shot setting for HellaSwag. Here, $n$-shot refers to incorporating $n$ instruction-following samples into the prompt question.

**Architectures Training.** We employ experiments on two LLaMA versions, LLaMA-7B with 32 Transformer layers and LLaMA-13B with 40 Transformer layers (Touvron et al., 2023b;c). The models are trained with 4 A100-GPUs for 5 epochs. The training configuration includes a warmup period of 2 epochs, a total batch size of 64, a learning rate of 0.009, and a weight decay of 0.02. With LLaMA-7B, we use a prompt with length $L = 10$ and integrate adaptation prompts into the last $K = 30$ layers. On LLaMA-13B, we use $L = 10$ and insert prompts at the last $K = 38$ layers.

**Baselines.** To access the effectiveness of zero-initialized attention and demonstrate the potential of integrating it with the proposed non-linear prompt, we conduct experiments using the following configurations:

- (1) **Linear prompt**: i.e., the default setting as LLaMA-Adapter (Zhang et al., 2024), where prompt vectors $\mathbf{P}$ are zero-initialized and used directly.

- (2) **Non-Linear prompt**: use zero-initialized mechanism and apply a nonlinear MLP on input prompts $\mathbf{P}$ to generate prompt vectors $\tilde{\mathbf{P}} = \sigma(\mathbf{P})$ and the MLP layers are shared among layers.

- (3) **Random-Init prompt**: use the input prompts with conventional randomly-initialization, i.e., forms in Eq.(4) rather than the zero-initialized mechanism.

- (4) **Finetuning & low-rank decomposition**: We compare with a fully fine-tuned LLaMA model trained on

the Alpaca dataset (Taori et al., 2023), where all model parameters are updated. Additionally, we benchmark against LoRA (Hu et al., 2021), a PEFT method that inserts trainable low-rank decomposition matrices into each layer's weights, and VeRA (Kopiczko et al., 2023), another PEFT method that uses a single random pair of low-rank matrices shared across all layers, while learning separate scaling vectors for each layer.

- (5) **Other Prompt Tuning Methods**: including (Lester et al., 2021) which optimize continuous prompts added to input text; IA3 (Liu et al., 2022) a method using three learnable vectors to rescale the keys, values, and position-wise feed-forward activations in the attention layers.

## 5.2. Main Results

**I. Zero-initialized attention is essential in prompt-tuning, enhancing both *robustness* and *effectiveness* compared to random-initialized attention.**

We begin by investigating the impact of zero-initialized prompt-tuning on LLaMA-7B and LLaMA-13B using the linear prompt setting, comparing its performance against conventional random-initialization strategies.

As shown in Table 1, the zero-initialized mechanism enhances stability in LLaMA-13B and significantly boosts the performance of LLaMA-Adapter when using LLaMA-7B, compared to conventional random-initialization strategies. For instance, with LLaMA-7B on Hellaswag and TruthfulQA, the Linear prompt surpasses the Random-Init prompt by $4.17\%$ and $7.71\%$, respectively. The varying impact of zero-initialization between LLaMA-7B and LLaMA-13B can be attributed to differences in model capacity and expressiveness. LLaMA-13B, with its larger parameter space, naturally generalizes better even with randomly initialized prompts, reducing the relative advantage of zero-initialization. In contrast, LLaMA-7B, with its lower capacity, benefits more from zero-initialization, as it relies heavily on efficient adaptation mechanisms to optimize learning.

In summary, those results align with our theoretical findings in Section 4, which demonstrate how zero-initialized

attention improves both robustness and sample efficiency in parameter estimation.

**II. Non-linear prompts provide the potential to improve the performance of the zero-initialized mechanism and achieve competitive performance with full fine-tuning.**

We evaluate the impact of the `non-linear prompt` setting on the LLaMA-Adapter by comparing it with the `linear prompt` setting on LLaMA-7B and LLaMA-13B. Additionally, we benchmark against a fully fine-tuned LLaMA model and a version fine-tuned using LoRA. All models are trained with the Alpaca dataset.

Tables 2 and 3 presents our results with the following observations: first, the `non-linear prompt` consistently matches or outperforms the `linear prompt` across both LLaMA-7B and LLaMA-13B, with performance gains ranging from 1–2%. Notably, on the MMLU and TruthfulQA datasets with LLaMA-13B, the non-linear prompt achieves scores of 51.32 and 38.92, respectively, compared to 49.64 and 34.88 for the linear prompt, demonstrating its effectiveness. Table 3 also shows that the non-linear prompt does not significantly increase training time compared to the linear version.

Secondly, compared to other PEFT methods such as Prompt Tuning, IA3, and VeRA, we see that the `non-linear prompt` significantly outperforms all these PEFT methods in LLaMA-7B version. For example, on the MMLU and TruthfulQA, the `non-linear prompt` surpasses Prompt Tuning by 4.1% and 10.28%, respectively, demonstrating the effectiveness of the new PEFT design.

Finally, when compared to fully fine-tuned and LoRA-based versions of LLaMA-7B, our approach using a non-linear prompt achieves performance nearly on par with full fine-tuning while surpassing LoRA by an average accuracy margin of 0.64%

In summary, these findings validate our theoretical observations and highlight the effectiveness of combining non-linear prompts with zero-initialized attention and improving prompt-tuning performance in LLMs.

**III. Sample Efficiency of Zero-Initialized Attention vs. Random-Initialized Attention.**

Figures 2 and 3 provide a systematic analysis of the sample efficiency of zero-initialized attention by evaluating its performance under varying data availability. Specifically, we randomly subsample the Alpaca dataset at different fractions {1%, 10%, 30%, 50%, 100%} to simulate low-data scenarios. We then fine-tune the `Non-Linear`, `Linear`, and `Random-Init` prompts on these subsets for both LLaMA-7B and LLaMA-13B and evaluate their performance on the ARC dataset. This experiment allows us to assess how

well each initialization strategy adapts to limited data and whether zero-initialized attention provides a consistent advantage in sample efficiency.

We observe that both Non-Linear and Linear prompts significantly enhance sample efficiency in parameter estimation for prefix-tuning in LLMs compared to the Random-Init prompt (which uses conventional attention). In the LLaMA-7B setting (Figure 2), both Non-Linear and Linear prompts outperform Random-Init across all fractions of the Alpaca training set. For example, the Non-Linear prompt exceeds Random-Init by 3.77% when trained on 100% of the dataset, while Linear exceeds it by 2.05%. When trained on 50% of the dataset, Non-Linear and Linear outperform Random-Init by 4.72% and 2.16%, respectively.

In the LLaMA-13B setting (Figure 3), a similar trend is observed, with zero-initialized attention showing consistent advantages over Random-Init, and the Non-Linear prompt slightly outperforming the Linear prompt in most cases. The only exception is at 1% training data, where Linear surpasses Non-Linear by 1.99% in accuracy. In short, these findings corroborate our theoretical results in Section 4, which demonstrate the sample efficiency of the zero-initialized attention mechanism in parameter estimation.

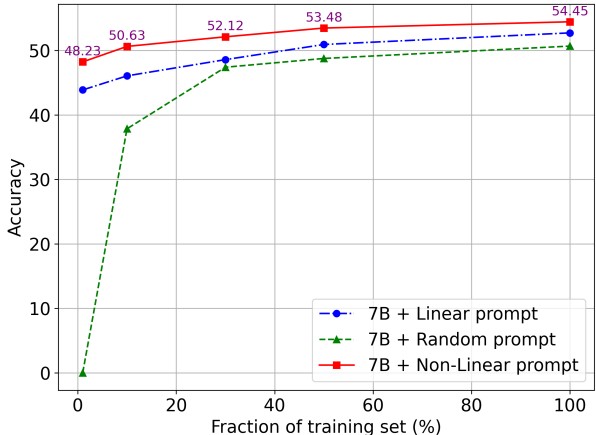

*Figure 2.* Accuracy of different prompt strategies (Linear, Random, and Non-Linear) with a **LLaMa-7B** model across varying fractions of the training set. The Non-Linear prompt consistently outperforms the other methods, especially in low-data regimes.

## 6. Conclusion and Limitations

In this paper, we demonstrate that zero-initialization prompt-tuning for adapting LLMs is not just an engineering trick but can be rigorously explained through theoretical properties by drawing connections between attention mechanisms and the mixture-of-experts perspective. Based on these insights, we introduce a novel non-linear prompt-tuning approach that outperforms linear prompts in terms of both performance and robustness while achieving competitive results compared to the full fine-tuning of LLaMA on the

*Table 2.* Comparison of `Non-Linear prompt`, `Linear prompt`, and various fine-tuning methods. **Params** denote the total number of parameters updated during the fine-tuning process. **Bold** values indicate better scores between linear and non-linear settings.

| Method | Params | ARC | | | MMLU | Hellaswag | TruthfullQA | Average |
| --- | --- | --- | --- | --- | --- | --- | --- | --- |
| | | *Acc (eas)* | *Acc (cha)* | *Acc (aver)* | *Acc* | *Acc* | *Acc* | |
| LLaMA-7B, Fully Fine-tuning Alpaca | 7B | 67.47 | 46.25 | 56.86 | 37.25 | 77.09 | 42.35 | 53.39 |
| LLaMA-7B, LoRA Alpaca | 4.2M | 61.91 | 42.15 | 52.03 | 34.87 | 77.53 | 46.14 | 52.64 |
| LLaMA-7B, Prompt Tuning Alpaca | 42K | 55.35 | 37.46 | 46.41 | 32.85 | 75.88 | 34.76 | 47.48 |
| LLaMA-7B, IA3 Alpaca | 524K | 52.06 | 35.92 | 43.99 | 31.65 | 75.73 | 32.8 | 46.04 |
| LLaMA-7B, VeRA Alpaca | 541K | 49.2 | 35.49 | 42.35 | 30.88 | 75.59 | 31.95 | 45.19 |
| LLaMA-7B + zero-init + linear | 1.2M | 62.29 | 43.17 | 52.73 | 36.28 | **76.79** | **45.53** | 52.83 |
| LLaMA-7B + zero-init + non-linear | 2.6M | **63.51** | **45.39** | **54.45** | **36.95** | 76.67 | 45.04 | **53.28** |
| LLaMA-13B + zero-init + linear | 1.9M | 81.78 | 64.33 | 73.06 | 49.64 | 81.21 | 34.88 | 59.70 |
| LLaMA-13B + zero-init + non-linear | 3.3M | **82.87** | **66.55** | **74.71** | **51.32** | **81.72** | **38.92** | **61.67** |

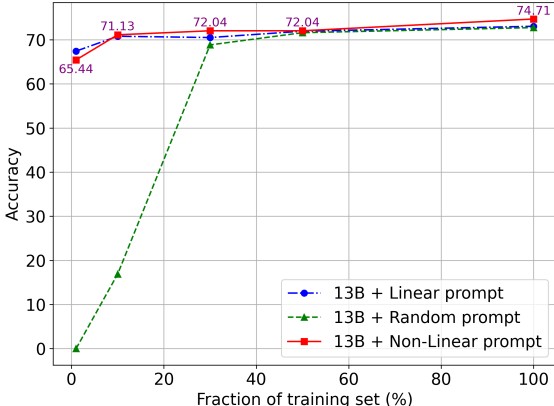

*Figure 3.* Accuracy of different prompt strategies (Linear, Random, and Non-Linear) using the **LLaMa-13B** model across varying fractions of the training set. The Non-Linear prompt consistently achieves the highest accuracy, particularly in low-resource settings. In contrast, the Random prompt performs significantly worse when only a small portion of the training data is available, but gradually improves as more data is introduced.

*Table 3.* Efficiency Comparison. The training is conducted on 4 GPUs A100-80GB.

| Method | Tuned Params | Storage Space | Training Time |
| --- | --- | --- | --- |
| LLaMA-7B + zero-init + linear | 1.2M | 4.4M | 1h54' |
| LLaMA-7B + zero-init + non-linear | 2.6M | 9.5M | 1h54' |
| LLaMA-13B + zero-init + linear | 1.9M | 6.9M | 3h17' |
| LLaMA-13B + zero-init + non-linear | 3.3M | 12M | 3h17' |

Alpaca dataset. Our findings are validated across several question-answering tasks, tested on both LLaMA-7B and LLaMA-13B architectures, and align with our theoretical analysis in practical settings. We believe our results will encourage further exploration into initialization techniques and their potential for improving other parameter-efficient fine-tuning methods.

Our study also has some limitations that need further exploration. First, it has only been tested on unimodal LLMs,

leaving the performance on multi-modal LLMs unexamined. Multi-modal tasks may require adjustments to the framework in both theoretical perspectives and practical implementations (Li et al., 2023; Nguyen et al., 2024a). Second, the integration of adapters and the optimal number of prompt embeddings across layers have not been systematically explored. Further empirical experiments are needed to refine these components for better efficiency and adaptability in more complex models. Finally, the current study primarily focuses on performance metrics, but the interpretability and stability of non-linear prompts in real-world deployment scenarios require further analysis. Addressing these limitations will contribute to a deeper understanding of the robustness and generalizability of PEFT for LLaMA models.

## Impact Statement

This paper presents work whose goal is to advance the field of Machine Learning. There are many potential societal consequences of our work, none which we feel must be specifically highlighted here.

## Acknowledgment

This work was supported by Deutsche Forschungsgemeinschaft (DFG, German Research Foundation) under Germany's Excellence Strategy - EXC 2075 – 390740016, the DARPA ANSR program under award FA8750-23-2-0004, the DARPA CODORD program under award HR00112590089. The authors thank the International Max Planck Research School for Intelligent Systems (IMPRS-IS) for supporting Duy M. H. Nguyen. Duy M. H. Nguyen and Daniel Sonntag are also supported by the XAINES project (BMBF, 01IW20005), No-IDLE project (BMBF, 01IW23002), and the Endowed Chair of Applied Artificial Intelligence, Oldenburg University. This research is also supported by research funding from Faculty of Information Technology, University of Science, Vietnam National University - Ho Chi Minh City.

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

# Supplement to "On Zero-Initialized Attention: Optimal Prompt and Gating Factor Estimation"

In this supplementary material, we provide detailed proofs of the **main results** in Appendix A. Additional discussion on related works is in Appendix B and additional experimental details are in Appendix C.

## A. Proofs

### A.1. Proof of Theorem 4.2

Based on the convergence rate of $f_{\widehat{G}_n, \widehat{\alpha}_n}$ to $f_{G_*, \alpha_*}$ in Proposition 4.1, to obtain the conclusion of Theorem 4.2, we only need to demonstrate that

$$\|f_{G,\alpha} - f_{G_*,\alpha_*}\|_{L^2(\mu)} \geq C \cdot (\mathcal{D}(G, G_*) + |\alpha - \alpha_*|)$$

for any $(G, \alpha) \in \mathcal{G}_{L'}(\Theta) \times \Omega$ for some universal constant $C$. It is equivalent to proving that:

$$\inf_{(G,\alpha)\in\mathcal{G}_{L'}(\Theta)\times\Omega} \|f_{G,\alpha} - f_{G_*,\alpha_*}\|_{L^2(\mu)}/(\mathcal{D}(G, G_*) + |\alpha - \alpha_*|) > 0.$$

To obtain the conclusion for the above inequality, we consider two parts: (i) local part, namely,

$$\lim_{\varepsilon\to 0} \inf_{(G,\alpha)\in\mathcal{G}_{L'}(\Theta)\times\Omega:\mathcal{D}(G,G_*)+|\alpha-\alpha_*|\leq\varepsilon} \|f_{G,\alpha} - f_{G_*,\alpha_*}\|_{L^2(\mu)}/(\mathcal{D}(G, G_*) + |\alpha - \alpha_*|) > 0;$$

(ii) global part, namely, for any $\varepsilon > 0$

$$\inf_{(G,\alpha)\in\mathcal{G}_{L'}(\Theta)\times\Omega:\mathcal{D}(G,G_*)+|\alpha-\alpha_*|>\varepsilon} \|f_{G,\alpha} - f_{G_*,\alpha_*}\|_{L^2(\mu)}/(\mathcal{D}(G, G_*) + |\alpha - \alpha_*|) > 0;$$

**Local part:** We first start with the local part, which is equivalent to demonstrating that

$$\lim_{\varepsilon\to 0} \inf_{(G,\alpha)\in\mathcal{G}_{L'}(\Theta)\times\Omega:\mathcal{D}(G,G_*)+|\alpha-\alpha_*|\leq\varepsilon} \|f_{G,\alpha} - f_{G_*,\alpha_*}\|_{L^2(\mu)}/(\mathcal{D}(G, G_*) + |\alpha - \alpha_*|) > 0.$$

We prove the above claim by contradiction. Assume by contrary that the above claim does not hold. It indicates that we can find a sequence of mixing measures $G_n := \sum_{j'=1}^{L_n} \exp(\bar{b}_{n,j'})\delta_{\boldsymbol{p}_{n,j'}}$ in $\mathcal{G}_{L'}(\Theta)$ and a sequence of $\alpha_n \in \Omega$ such that when $n \to \infty$, the following limits hold:

$$\begin{cases} \mathcal{D}(G_n, G_*) + |\alpha_n - \alpha_*| \to 0, \\ \|f_{G_n,\alpha_n} - f_{G_*,\alpha_*}\|_{L^2(\mu)}/(\mathcal{D}(G_n, G_*) + |\alpha_n - \alpha_*|) \to 0. \end{cases}$$

The first limit indicates that $\mathcal{D}_n := \mathcal{D}(G_n, G_*) \to 0$ and $\alpha_n \to \alpha_*$ as $n \to \infty$.

For the simplicity of the ensuing presentation, we denote $\mathcal{C}_j^n := \mathcal{C}_j(G_n)$ as a Voronoi cell of $G_n$ induced by the $j$-th components of $G_*$. Without loss of generality, we assume that those Voronoi cells do not depend on the sample size, i.e., $\mathcal{C}_j = \mathcal{C}_j^n$, which is possible since our arguments are asymptotic. Therefore, we can rewrite the Voronoi loss $\mathcal{D}_n$ as follows:

$$\mathcal{D}_n := \sum_{j'=1}^{L} \Big| \sum_{i\in\mathcal{C}_{j'}} \exp(\bar{b}_{n,i}) - \exp(\bar{b}_{*,j'}) \Big| + \sum_{j'\in[L]:|\mathcal{C}_{j'}|=1} \sum_{i\in\mathcal{C}_{j'}} \exp(\bar{b}_{n,i})\|\Delta\boldsymbol{p}_{n,ij'}\|$$

$$+ \sum_{j'\in[L]:|\mathcal{C}_{j'}|>1} \sum_{i\in\mathcal{C}_{j'}} \exp(\bar{b}_{n,i})\|\Delta\boldsymbol{p}_{n,ij'}\|^2,$$

where $\Delta\boldsymbol{p}_{n,ij'} = \boldsymbol{p}_{n,i} - \boldsymbol{p}_{*,j'}$ for all $i \in \mathcal{C}_{j'}$.

From the hypothesis, we have $\mathcal{D}_n \to 0$, which implies that $\sum_{i\in\mathcal{C}_j} \exp(\bar{b}_{n,i}) \to \exp(\bar{b}_{*,j})$ and $\boldsymbol{p}_{n,i} \to \boldsymbol{p}_{*,j}$ for any $i \in \mathcal{C}_j, j \in [L]$. To establish the contradiction, our proof consists of three main steps.

**Step 1 - Taylor expansion.** To ease the presentation, let us denote

$$f_{G_n}(\boldsymbol{X}) := \sum_{j=1}^{L_n} \frac{\exp((\bar{B}\boldsymbol{p}_{n,j})^\top \boldsymbol{X} + \bar{b}_{n,j})}{\sum_{k=1}^{L_n} \exp((\bar{B}\boldsymbol{p}_{n,k})^\top \boldsymbol{X} + \bar{b}_{n,k})} \cdot \bar{C}\boldsymbol{p}_{n,j},$$

$$f_{G_*}(\boldsymbol{X}) := \sum_{j'=1}^{L} \frac{\exp((\bar{B}\boldsymbol{p}_{*,j'})^\top \boldsymbol{X} + \bar{b}_{*,j'})}{\sum_{k'=1}^{L} \exp((\bar{B}\boldsymbol{p}_{*,k'})^\top \boldsymbol{X} + \bar{b}_{*,k'})} \cdot \bar{C}\boldsymbol{p}_{*,j'}.$$

We now decompose the function $\bar{Q}_n(\boldsymbol{X})$ as follows:

$$\begin{aligned}
\bar{Q}_n(\boldsymbol{X}) :&= \Big[ \sum_{k'=1}^{L} \exp((\bar{B}\boldsymbol{p}_{*,k'})^\top \boldsymbol{X} + \bar{b}_{*,k'}) \Big] \cdot [f_{G_n,\alpha_n}(\boldsymbol{X}) - f_{G_*,\alpha_*}(\boldsymbol{X})] \\
&= \Big[ \sum_{k'=1}^{L} \exp((\bar{B}\boldsymbol{p}_{*,k'})^\top \boldsymbol{X} + \bar{b}_{*,k'}) \Big] \cdot \Big( \tanh(\alpha_n) f_{G_n}(\boldsymbol{X}) - \tanh(\alpha_*) f_{G_*}(\boldsymbol{X}) \Big) \\
&= \Big[ \sum_{k'=1}^{L} \exp((\bar{B}\boldsymbol{p}_{*,k'})^\top \boldsymbol{X} + \bar{b}_{*,k'}) \Big] \cdot \tanh(\alpha_n) \Big[ f_{G_n}(\boldsymbol{X}) - f_{G_*}(\boldsymbol{X}) \Big] \\
&\quad + \Big[ \sum_{k'=1}^{L} \exp((\bar{B}\boldsymbol{p}_{*,k'})^\top \boldsymbol{X} + \bar{b}_{*,k'}) \Big] \cdot [\tanh(\alpha_n) - \tanh(\alpha_*)] f_{G_*}(\boldsymbol{X}) \\
&:= \bar{Q}_{n,1}(\boldsymbol{X}) + \bar{Q}_{n,2}(\boldsymbol{X}).
\end{aligned}$$

For that purpose, we will decompose the two terms $\bar{Q}_{n,1}(\boldsymbol{X})$ and $\bar{Q}_{n,2}(\boldsymbol{X})$, respectively.

**Decomposition of the function $\bar{Q}_{n,1}(\boldsymbol{X})$.** We have

$$\begin{aligned}
\bar{Q}_{n,1}(\boldsymbol{X}) &= \sum_{j=1}^{L} \sum_{i\in\mathcal{C}_j} \tanh(\alpha_n) \exp(\bar{b}_{n,i}) \Big[ \exp((\bar{B}\boldsymbol{p}_{n,i})^\top \boldsymbol{X}) \bar{C}\boldsymbol{p}_{n,i} - \exp((\bar{B}\boldsymbol{p}_{*,j})^\top \boldsymbol{X}) \bar{C}\boldsymbol{p}_{*,j} \Big] \\
&\quad - \sum_{j=1}^{L} \sum_{i\in\mathcal{C}_j} \tanh(\alpha_n) \exp(\bar{b}_{n,i}) \Big[ \exp((\bar{B}\boldsymbol{p}_{n,i})^\top \boldsymbol{X}) - \exp((\bar{B}\boldsymbol{p}_{*,j})^\top \boldsymbol{X}) \Big] f_{G_n}(\boldsymbol{X}) \\
&\quad + \sum_{j=1}^{L} \tanh(\alpha_n) \Big( \sum_{i\in\mathcal{C}_j} \exp(\bar{b}_{n,i}) - \exp(\bar{b}_{*,j}) \Big) \exp((\bar{B}\boldsymbol{p}_{*,j})^\top \boldsymbol{X}) \Big[ \bar{C}\boldsymbol{p}_{*,j} - f_{G_n}(\boldsymbol{X}) \Big] \\
&:= \bar{A}_n(\boldsymbol{X}) - \bar{B}_n(\boldsymbol{X}) + \bar{C}_n(\boldsymbol{X}).
\end{aligned} \tag{22}$$

We now proceed to decompose the functions $\bar{A}_n(.)$ and $\bar{B}_n(.)$ via Taylor expansion.

**Decomposition of the function $\bar{A}_n(\boldsymbol{X})$.** We first define the following functions $\bar{U}(\boldsymbol{X};\boldsymbol{p}) := \exp((\bar{B}\boldsymbol{p})^\top \boldsymbol{X})$ and $\bar{V}(\boldsymbol{p}) = \bar{C}\boldsymbol{p}$. Then, we denote the product of these functions as $\bar{F}(\boldsymbol{X};\boldsymbol{p}) = \bar{U}(\boldsymbol{X};\boldsymbol{p})\bar{V}(\boldsymbol{p})$. To decompose $\bar{A}_n(\boldsymbol{X})$, we separately consider Voronoi cells with exactly one element and those with more than one element. It leads to the following decomposition of the function $\bar{A}_n(\boldsymbol{X})$:

$$\begin{aligned}
\bar{A}_n(\boldsymbol{X}) &= \sum_{j:|\mathcal{C}_j|=1} \sum_{i\in\mathcal{C}_j} \tanh(\alpha_n) \exp(\bar{b}_{n,i}) \Big[ \bar{F}(\boldsymbol{X};\boldsymbol{p}_{n,i}) - \bar{F}(\boldsymbol{X};\boldsymbol{p}_{*,j}) \Big] \\
&\quad + \sum_{j:|\mathcal{C}_j|>1} \sum_{i\in\mathcal{C}_j} \tanh(\alpha_n) \exp(\bar{b}_{n,i}) \Big[ \bar{F}(\boldsymbol{X};\boldsymbol{p}_{n,i}) - \bar{F}(\boldsymbol{X};\boldsymbol{p}_{*,j}) \Big] \\
&:= \bar{A}_{n,1}(\boldsymbol{X}) + \bar{A}_{n,2}(\boldsymbol{X}),
\end{aligned}$$

where we denote $\bar{A}_{n,1}(\boldsymbol{X}) = \sum_{j:|\mathcal{C}_j|=1} \sum_{i\in\mathcal{C}_j} \tanh(\alpha_n) \exp(\bar{b}_{n,i}) \Big[ \bar{F}(\boldsymbol{X};\boldsymbol{p}_{n,i}) - \bar{F}(\boldsymbol{X};\boldsymbol{p}_{*,j}) \Big]$ and $\bar{A}_{n,2}(\boldsymbol{X}) = \sum_{j:|\mathcal{C}_j|>1} \sum_{i\in\mathcal{C}_j} \tanh(\alpha_n) \exp(\bar{b}_{n,i}) \Big[ \bar{F}(\boldsymbol{X};\boldsymbol{p}_{n,i}) - \bar{F}(\boldsymbol{X};\boldsymbol{p}_{*,j}) \Big]$.

For the function $\bar{A}_{n,1}(\boldsymbol{X})$, for any indices $i \in \mathcal{C}_j$ and $j$ such that $|\mathcal{C}_j| = 1$, the first-order Taylor expansion entails that

$$\bar{U}(\boldsymbol{X}; \boldsymbol{p}_{n,i}) = \bar{U}(\boldsymbol{X}; \boldsymbol{p}_{*,j}) + \sum_{|\alpha|=1} (\Delta \boldsymbol{p}_{n,ij})^\alpha \frac{\partial^{|\alpha|} \bar{U}}{\partial \boldsymbol{p}^\alpha}(\boldsymbol{X}; \boldsymbol{p}_{*,j}) + \bar{R}_{ij,1}(\boldsymbol{X}),$$

$$\bar{V}(\boldsymbol{p}_{n,i}) = \bar{V}(\boldsymbol{p}_{*,j}) + \sum_{|\alpha|=1} (\Delta \boldsymbol{p}_{n,ij})^\alpha \frac{\partial^{|\alpha|} \bar{V}}{\partial \boldsymbol{p}^\alpha}(\boldsymbol{p}_{*,j}) + \bar{R}_{ij,2},$$

where the terms $\bar{R}_{ij,1}(\boldsymbol{X})$ and $\bar{R}_{ij,2}$ are Taylor remainders.

Combining the above results leads to the following formulation of the function $\bar{A}_{n,1}(\boldsymbol{X})$:

$$\bar{A}_{n,1}(\boldsymbol{X}) = \sum_{j:|\mathcal{C}_j|=1} \sum_{i \in \mathcal{C}_j} \frac{\tanh(\alpha_n) \exp(\bar{b}_{n,i})}{\alpha!} \sum_{|\alpha|=1} \left\{ (\Delta \boldsymbol{p}_{n,ij})^\alpha \frac{\partial^{|\alpha|} \bar{U}}{\partial \boldsymbol{p}^\alpha}(\boldsymbol{X}; \boldsymbol{p}_{*,j}) \bar{V}(\boldsymbol{p}_{*,j}) \right.$$

$$\left. + (\Delta \boldsymbol{p}_{n,ij})^\alpha \frac{\partial^{|\alpha|} \bar{V}}{\partial \boldsymbol{p}^\alpha}(\boldsymbol{p}_{*,j}) \bar{U}(\boldsymbol{X}; \boldsymbol{p}_{*,j}) \right\} + \bar{R}_{n,1}(\boldsymbol{X})$$

$$= \sum_{j:|\mathcal{C}_j|=1} \sum_{|\alpha|=1} \left\{ \bar{M}_{n,j,\alpha} \frac{\partial^{|\alpha|} \bar{U}}{\partial \boldsymbol{p}^\alpha}(\boldsymbol{X}; \boldsymbol{p}_{*,j}) \bar{V}(\boldsymbol{p}_{*,j}) + \bar{M}_{n,j,\alpha} \frac{\partial^{|\alpha|} \bar{V}}{\partial \boldsymbol{p}^\alpha}(\boldsymbol{p}_{*,j}) \bar{U}(\boldsymbol{X}; \boldsymbol{p}_{*,j}) \right\} + \bar{R}_{n,1}(\boldsymbol{X})$$

where the function $\bar{R}_{n,1}(\boldsymbol{X})$ is the combination of Taylor remainders and satisfies that $\bar{R}_{n,1}(\boldsymbol{X})/(\mathcal{D}_n + |\alpha_n - \alpha_*|) \to 0$ when $n \to \infty$. Furthermore, the formulations of $\bar{M}_{n,j,\alpha}$ are as follows:

$$\bar{M}_{n,j,\alpha} = \sum_{i \in \mathcal{C}_j} \frac{\tanh(\alpha_n) \exp(\bar{b}_{n,i})}{\alpha!} (\Delta \boldsymbol{p}_{n,ij})^\alpha,$$

for any $|\alpha| = 1$.

Moving to the function $\bar{A}_{n,2}(\boldsymbol{X})$, the second-order Taylor expansions of the function $\bar{U}(\boldsymbol{X}; \boldsymbol{p}_{n,i})$ around the function $\bar{U}(\boldsymbol{X}; \boldsymbol{p}_{*,j})$ and the function $\bar{V}(\boldsymbol{p}_{n,i})$ around the function $\bar{V}(\boldsymbol{p}_{*,j})$ for any indices $i \in \mathcal{C}_j$ and $j$ such that $|\mathcal{C}_j| > 1$, we obtain the following formulation of the function $\bar{A}_{n,2}(\boldsymbol{X})$:

$$\bar{A}_{n,2}(\boldsymbol{X}) = \sum_{j:|\mathcal{C}_j|>1} \sum_{1 \leq |\alpha| \leq 2} \left\{ \bar{M}_{n,j,\alpha} \frac{\partial^{|\alpha|} \bar{U}}{\partial \boldsymbol{p}^\alpha}(\boldsymbol{X}; \boldsymbol{p}_{*,j}) \bar{V}(\boldsymbol{p}_{*,j}) + \bar{M}_{n,j,\alpha} \frac{\partial^{|\alpha|} \bar{V}}{\partial \boldsymbol{p}^\alpha}(\boldsymbol{p}_{*,j}) \bar{U}(\boldsymbol{X}; \boldsymbol{p}_{*,j}) \right\}$$

$$+ \sum_{j:|\mathcal{C}_j|>1} \sum_{|\alpha|=1, |\beta|=1} \bar{M}_{n,j,\alpha,\beta} \frac{\partial^{|\alpha|} \bar{U}}{\partial \boldsymbol{p}^\alpha}(\boldsymbol{X}; \boldsymbol{p}_{*,j}) \frac{\partial^{|\beta|} \bar{V}}{\partial \boldsymbol{p}^\beta}(\boldsymbol{p}_{*,j}) + \bar{R}_{n,2}(\boldsymbol{X})$$

where the function $\bar{R}_{n,2}(\boldsymbol{X})$ is a combination of Taylor remainders and satisfies $\bar{R}_{n,2}(\boldsymbol{X})/(\mathcal{D}_n + |\alpha_n - \alpha_*|) \to 0$ when $n \to \infty$. Furthermore, we define

$$\bar{M}_{n,j,\alpha} = \sum_{i \in \mathcal{C}_j} \frac{\tanh(\alpha_n) \exp(b_{n,i})}{\alpha!} (\Delta \boldsymbol{p}_{n,ij})^\alpha,$$

for any $|\alpha| = 2$ and

$$\bar{M}_{n,j,\alpha,\beta} = \sum_{i \in \mathcal{C}_j} \frac{\tanh(\alpha_n) \exp(\bar{b}_{n,i})}{\alpha! \beta!} (\Delta \boldsymbol{p}_{n,ij})^{\alpha+\beta},$$

for any $|\alpha| = |\beta| = 1$. From the formulations of the functions $\bar{U}(\boldsymbol{X}; \boldsymbol{p})$ and $\bar{V}(\boldsymbol{p})$, we obtain the following explicit forms of their partial derivatives:

$$\frac{\partial \bar{U}}{\partial \boldsymbol{p}^{(u)}}(\boldsymbol{X}; \boldsymbol{p}) = \exp((\bar{B}\boldsymbol{p})^\top \boldsymbol{X})(\bar{B}1_u)^\top \boldsymbol{X},$$

$$\frac{\partial^2 \bar{U}}{\partial \boldsymbol{p}^{(u)} \partial \boldsymbol{p}^{(v)}}(\boldsymbol{X}; \boldsymbol{p}) = \exp((\bar{B}\boldsymbol{p})^\top \boldsymbol{X}) \boldsymbol{X}^\top (\bar{B}1_u)(\bar{B}1_v)^\top \boldsymbol{X},$$

$$\frac{\partial \bar{V}}{\partial \boldsymbol{p}^{(u)}}(\boldsymbol{p}) = \bar{C}1_u,$$

$$\frac{\partial^2 \bar{V}}{\partial \boldsymbol{p}^{(u)} \partial \boldsymbol{p}^{(v)}}(\boldsymbol{p}) = 0.$$

In these formulations, we use $1_u$ to denote the vector that its $u$-th element is 1 and its other elements are 0 for $1 \leq u \leq d$. Plugging these explicit formulations of the derivatives of the functions $\bar{U}(\boldsymbol{X}; \boldsymbol{p})$ and $\bar{V}(\boldsymbol{p})$, we can express the functions $\bar{A}_{n,1}(\boldsymbol{X})$ and $\bar{A}_{n,2}(\boldsymbol{X})$ as follows:

$$\bar{A}_{n,1}(\boldsymbol{X}) = \sum_{j:|\mathcal{C}_j|=1} \exp((\bar{B}\boldsymbol{p}_{*,j})^\top \boldsymbol{X}) \big[\mathcal{L}_{1,n}(\boldsymbol{p}_{*,j}) + \mathcal{L}_{2,n}(\boldsymbol{p}_{*,j})^\top \bar{B}^\top \boldsymbol{X}\big) + \bar{R}_{n,1}(\boldsymbol{X}),$$

$$\bar{A}_{n,2}(\boldsymbol{X}) = \sum_{j:|\mathcal{C}_j|>1} \exp((\bar{B}\boldsymbol{p}_{*,j})^\top \boldsymbol{X}) \big[\bar{\mathcal{L}}_{1,n}(\boldsymbol{p}_{*,j}) + \bar{\mathcal{L}}_{2,n}(\boldsymbol{p}_{*,j})^\top \bar{B}^\top \boldsymbol{X}$$

$$+ (\bar{B}^\top \boldsymbol{X})^\top \bar{\mathcal{L}}_{3,n}(\boldsymbol{p}_{*,j}) \bar{B}^\top \boldsymbol{X}\big] + \bar{R}_{n,2}(\boldsymbol{X}),$$

where the functions $\mathcal{L}_{1,n}(\boldsymbol{p}), \mathcal{L}_{2,n}(\boldsymbol{p}), \bar{\mathcal{L}}_{1,n}(\boldsymbol{p}), \bar{\mathcal{L}}_{2,n}(\boldsymbol{p})$, and $\bar{\mathcal{L}}_{3,n}(\boldsymbol{p})$ are defined as follows:

$$\mathcal{L}_{1,n}(\boldsymbol{p}) = \sum_{u=1}^{d} \bar{M}_{n,j,1_u} \bar{C}1_u,$$

$$\mathcal{L}_{2,n}(\boldsymbol{p}) = \sum_{u=1}^{d} \bar{M}_{n,j,1_u} 1_u \bar{C}\boldsymbol{p},$$

$$\bar{\mathcal{L}}_{1,n}(\boldsymbol{p}) = \sum_{u=1}^{d} \bar{M}_{n,j,1_u} C1_u,$$

$$\bar{\mathcal{L}}_{2,n}(\boldsymbol{p}) = \sum_{u=1}^{d} \bar{M}_{n,j,1_u} 1_u \bar{C}\boldsymbol{p} + \sum_{1 \leq u,v \leq d} \bar{M}_{n,j,1_v,1_u} \bar{C}1_u 1_v$$

$$\bar{\mathcal{L}}_{3,n}(\boldsymbol{p}) = \sum_{1 \leq u,v \leq d} \bar{M}_{n,j,1_{uv}} 1_u 1_v^\top \bar{C}\boldsymbol{p}.$$

In these formulations, we denote $1_{uv}$ as the matrix that its $(u,v)$-th element is 1 and its other elements are 0 for any $1 \leq u,v \leq d$.

**Decomposition of the function $\bar{B}_n(\boldsymbol{X})$.** Similar to the decomposition of the function $\bar{A}_n(\boldsymbol{X})$, we can decompose the function $\bar{B}_n(\boldsymbol{X})$ as follows:

$$\bar{B}_n(\boldsymbol{X}) = \sum_{j:|\mathcal{C}_j|=1} \sum_{i \in \mathcal{C}_j} \tanh(\alpha_n) \exp(\bar{b}_{n,i}) \big[\bar{U}(\boldsymbol{X}; \boldsymbol{p}_{n,i}) - \bar{U}(\boldsymbol{X}; \boldsymbol{p}_{*,j})\big] f_{G_n}(\boldsymbol{X})$$

$$+ \sum_{j:|\mathcal{C}_j|>1} \sum_{i \in \mathcal{C}_j} \tanh(\alpha_n) \exp(\bar{b}_{n,i}) \big[\bar{U}(\boldsymbol{X}; \boldsymbol{p}_{n,i}) - \bar{U}(\boldsymbol{X}; \boldsymbol{p}_{*,j})\big] f_{G_n}(\boldsymbol{X})$$

$$:= \bar{B}_{n,1}(\boldsymbol{X}) + \bar{B}_{n,2}(\boldsymbol{X})$$

where we denote $\bar{B}_{n,1}(\boldsymbol{X}) = \sum_{j:|\mathcal{C}_j|=1} \sum_{i \in \mathcal{C}_j} \tanh(\alpha_n) \exp(\bar{b}_{n,i}) \big[\bar{U}(\boldsymbol{X}; \boldsymbol{p}_{n,i}) - \bar{U}(\boldsymbol{X}; \boldsymbol{p}_{*,j})\big] f_{G_n}(\boldsymbol{X})$ and $\bar{B}_{n,2}(\boldsymbol{X}) = \sum_{j:|\mathcal{C}_j|>1} \sum_{i \in \mathcal{C}_j} \tanh(\alpha_n) \exp(\bar{b}_{n,i}) \big[\bar{U}(\boldsymbol{X}; \boldsymbol{p}_{n,i}) - \bar{U}(\boldsymbol{X}; \boldsymbol{p}_{*,j})\big] f_{G_n}(\boldsymbol{X})$.

Similar to the Taylor expansions for the functions $\bar{A}_{n,1}(\boldsymbol{X})$ and $\bar{A}_{n,2}(\boldsymbol{X})$, by using the first-order Taylor expansion to $\bar{B}_{n,1}(\boldsymbol{X})$ and the second-order Taylor expansion to $\bar{B}_{n,2}(\boldsymbol{X})$, we obtain that

$$\bar{B}_{n,1}(\boldsymbol{X}) = \sum_{j:|\mathcal{C}_j|=1} \sum_{|\alpha|=1} \bar{M}_{n,j,\alpha} \frac{\partial^{|\alpha|} \bar{U}}{\partial \boldsymbol{p}^\alpha}(\boldsymbol{X}; \boldsymbol{p}_{*,j}) f_{G_n}(\boldsymbol{X}) + \bar{R}_{n,3}(\boldsymbol{X})$$

$$\bar{B}_{n,2}(\boldsymbol{X}) = \sum_{j:|\mathcal{C}_j|=1} \sum_{1\leq|\alpha|\leq 2} \bar{M}_{n,j,\alpha} \frac{\partial^{|\alpha|}\bar{U}}{\partial \boldsymbol{p}^{\alpha}}(\boldsymbol{X};\boldsymbol{p}_{*,j}) f_{G_n}(\boldsymbol{X}) + \bar{R}_{n,4}(\boldsymbol{X})$$

where the functions $\bar{R}_{n,3}(\boldsymbol{X})$, $\bar{R}_{n,4}(\boldsymbol{X})$ are Taylor remainders. Furthermore, they satisfy that $\bar{R}_{n,3}(\boldsymbol{X})/(\mathcal{D}_n+|\alpha_n-\alpha_*|) \to 0$ and $\bar{R}_{n,4}(\boldsymbol{X})/(\mathcal{D}_n+|\alpha_n-\alpha_*|) \to 0$ when $n \to \infty$. Given the explicit formulations of the derivatives of the functions $\bar{U}(\boldsymbol{X};\boldsymbol{p})$ and $\bar{V}(\boldsymbol{p})$, the functions $\bar{B}_{n,1}(\boldsymbol{X})$ and $\bar{B}_{n,2}(\boldsymbol{X})$ can be then rewritten as follows:

$$\bar{B}_{n,1}(\boldsymbol{X}) = \sum_{j:|\mathcal{C}_j|=1} \exp((\bar{B}\boldsymbol{p}_{*,j})^\top \boldsymbol{X}) \mathcal{N}_{1,n}(\boldsymbol{p}_{*,j})^\top \boldsymbol{X} f_{G_n}(\boldsymbol{X}) + \bar{R}_{n,3}(\boldsymbol{X}),$$

$$\bar{B}_{n,2}(\boldsymbol{X}) = \sum_{j:|\mathcal{C}_j|>1} \exp((\bar{B}\boldsymbol{p}_{*,j})^\top \boldsymbol{X}) \big[ \bar{\mathcal{N}}_{1,n}(\boldsymbol{p}_{*,j})^\top \bar{B}^\top \boldsymbol{X}$$

$$+ (\bar{B}^\top \boldsymbol{X})^\top \bar{\mathcal{N}}_{2,n}(\boldsymbol{p}_{*,j})(\bar{B}^\top \boldsymbol{X}) \big] f_{G_n}(\boldsymbol{X}) + \bar{R}_{n,4}(\boldsymbol{X}).$$

Here, the functions $\mathcal{N}_{1,n}(\boldsymbol{X})$, $\bar{\mathcal{N}}_{1,n}(\boldsymbol{X})$, and $\bar{\mathcal{N}}_{2,n}(\boldsymbol{X})$ have the following formulations:

$$\mathcal{N}_{1,n}(\boldsymbol{p}) = \sum_{u=1}^{d} \bar{M}_{n,j,1_u} \mathbf{1}_u,$$

$$\bar{\mathcal{N}}_{1,n}(\boldsymbol{p}) = \sum_{u=1}^{d} \bar{M}_{n,j,1_u} \mathbf{1}_u,$$

$$\bar{\mathcal{N}}_{2,n}(\boldsymbol{p}) = \sum_{1\leq u,v\leq d} \bar{M}_{n,j,1_{uv}} \mathbf{1}_u \mathbf{1}_v^\top.$$

Collecting all of the above results with the decomposition of the functions $\bar{A}_n(\boldsymbol{X})$ and $\bar{B}_n(\boldsymbol{X})$, we can represent the function $\bar{Q}_{n,1}(\boldsymbol{X})$ in equation (22) as follows:

$$\bar{Q}_{n,1}(\boldsymbol{X}) = \sum_{j:|\mathcal{C}_j|=1} \exp((\bar{B}\boldsymbol{p}_{*,j})^\top \boldsymbol{X}) \big[ \mathcal{L}'_{1,n}(\boldsymbol{p}_{*,j}) + \mathcal{L}_{2,n}(\boldsymbol{p}_{*,j})^\top \bar{B}^\top \boldsymbol{X} \big)$$

$$+ \sum_{j:|\mathcal{C}_j|>1} \exp((\bar{B}\boldsymbol{p}_{*,j})^\top \boldsymbol{X}) \big[ \bar{\mathcal{L}}'_{1,n}(\boldsymbol{p}_{*,j}) + \bar{\mathcal{L}}_{2,n}(\boldsymbol{p}_{*,j})^\top \bar{B}^\top \boldsymbol{X} + (\bar{B}^\top \boldsymbol{X})^\top \bar{\mathcal{L}}_{3,n}(\boldsymbol{p}_{*,j}) \bar{B}^\top \boldsymbol{X} \big]$$

$$- \sum_{j:|\mathcal{C}_j|=1} \exp((\bar{B}\boldsymbol{p}_{*,j})^\top \boldsymbol{X}) \big[ \bar{M}_{n,j,0_d} + \mathcal{N}_{1,n}(\boldsymbol{p}_{*,j})^\top \bar{B}^\top \boldsymbol{X} \big] f_{G_n}(\boldsymbol{X})$$

$$- \sum_{j:|\mathcal{C}_j|>1} \exp((\bar{B}\boldsymbol{p}_{*,j})^\top \boldsymbol{X}) \big[ \bar{M}_{n,j,0_d} + \bar{\mathcal{N}}_{1,n}(\boldsymbol{p}_{*,j})^\top \bar{B}^\top \boldsymbol{X} + (\bar{B}^\top \boldsymbol{X})^\top \bar{\mathcal{N}}_{2,n}(\boldsymbol{p}_{*,j}) \bar{B}^\top \boldsymbol{X} \big] f_{G_n}(\boldsymbol{X})$$

$$+ \bar{R}_{n,1}(\boldsymbol{X}) + \bar{R}_{n,2}(\boldsymbol{X}) - \bar{R}_{n,3}(\boldsymbol{X}) - \bar{R}_{n,4}(\boldsymbol{X}) \tag{23}$$

where we define $\bar{M}_{n,j,0_d} = \tanh(\alpha_n)\big( \sum_{i\in\mathcal{C}_j} \exp(\bar{b}_{n,i}) - \exp(\bar{b}_{*,j}) \big)$ for any index $1 \leq j \leq L$, $\mathcal{L}'_{1,n}(\boldsymbol{p}_{*,j}) = \mathcal{L}_{1,n}(\boldsymbol{p}_{*,j}) + \bar{M}_{n,j,0_d} \bar{C} \boldsymbol{p}_{*,j}$, and $\bar{\mathcal{L}}'_{1,n}(\boldsymbol{p}_{*,j}) = \bar{\mathcal{L}}_{1,n}(\boldsymbol{p}_{*,j}) + \bar{M}_{n,j,0_d} \bar{C} \boldsymbol{p}_{*,j}$.

**Decomposition of the function $\bar{Q}_{n,2}(\boldsymbol{X})$.** An application of the first-order Taylor expansion leads to the following expression for the function $\bar{Q}_{n,2}(\boldsymbol{X})$:

$$\bar{Q}_{n,2}(\boldsymbol{X}) = [\tanh(\alpha_n) - \tanh(\alpha_*)] \cdot \Big[ \sum_{k'=1}^{L} \exp((\bar{B}\boldsymbol{p}_{*,k'})^\top \boldsymbol{X} + \bar{b}_{*,k'}) \Big] \cdot f_{G_*}(\boldsymbol{X})$$

$$= (\alpha_n - \alpha_*)[1 - \tanh^2(\alpha_*)] \cdot \Big[ \sum_{k'=1}^{L} \exp((\bar{B}\boldsymbol{p}_{*,k'})^\top \boldsymbol{X} + \bar{b}_{*,k'}) \Big] \cdot f_{G_*}(\boldsymbol{X}) + \bar{R}_5(\boldsymbol{X}), \tag{24}$$

where the function $\bar{R}_{n,5}(\boldsymbol{X})$ is Taylor remainder and satisfies that $\bar{R}_{n,5}(\boldsymbol{X})/(\mathcal{D}_n + |\alpha_n - \alpha_*|) \to 0$ as $n \to \infty$.

**Step 2 - Non-vanishing coefficients.** The results of equations (23) and (24) indicate that $[\bar{Q}_n(\boldsymbol{X}) - \sum_{i=1}^{5} \bar{R}_{n,i}(\boldsymbol{X})]/(\mathcal{D}_n + |\alpha_n - \alpha_*|)$ can be represented as a combination of the linearly independent functions $\exp((\bar{B}\boldsymbol{p}_{*,j})^{\top}\boldsymbol{X})$, $(\bar{B}^{\top}\boldsymbol{X})^{(u)}\exp((\bar{B}\boldsymbol{p}_{*,j})^{\top}\boldsymbol{X})$, $(\bar{B}^{\top}\boldsymbol{X})^{(u)}(\bar{B}^{\top}\boldsymbol{X})^{(v)}\exp((\bar{B}\boldsymbol{p}_{*,j})^{\top}\boldsymbol{X})$, $\exp((\bar{B}\boldsymbol{p}_{*,j})^{\top}\boldsymbol{X})f_{G_n}(\boldsymbol{X})$,

$(\bar{B}^{\top}\boldsymbol{X})^{(u)}\exp((\bar{B}\boldsymbol{p}_{*,j})^{\top}\boldsymbol{X})f_{G_n}(\boldsymbol{X})$, $(\bar{B}^{\top}\boldsymbol{X})^{(u)}(\bar{B}^{\top}\boldsymbol{X})^{(v)}\exp((\bar{B}\boldsymbol{p}_{*,j})^{\top}\boldsymbol{X})f_{G_n}(\boldsymbol{X})$, and $[1 - \tanh^2(\alpha_*)] \cdot \left[\sum_{k'=1}^{L}\exp((\bar{B}\boldsymbol{p}_{*,k'})^{\top}\boldsymbol{X} + \bar{b}_{*,k'})\right] \cdot f_{G_*}(\boldsymbol{X})$ for any $1 \leq j \leq L$ and $1 \leq u, v \leq d$.

Our claim is that at least one of the coefficients of these linearly independent terms in the formulation of $[\bar{Q}_n(\boldsymbol{X}) - \sum_{i=1}^{5}\bar{R}_{n,i}(\boldsymbol{X})]/(\mathcal{D}_n + |\alpha_n - \alpha_*|)$ does not go to 0 as $n \to \infty$. Assume by contrary that this claim does not hold, which means that all the coefficients of these linearly independent terms go to 0 as $n \to \infty$. Therefore, as $n \to \infty$ we obtain that

$$\mathcal{L}_{1,n}(\boldsymbol{p}_{*,j})/(\mathcal{D}_n + |\alpha_n - \alpha_*|) \to 0, \ \mathcal{L}_{2,n}(\boldsymbol{p}_{*,j})^{(u)}/(\mathcal{D}_n + |\alpha_n - \alpha_*|) \to 0, \ \bar{\mathcal{L}}_{1,n}(\boldsymbol{p}_{*,j})/(\mathcal{D}_n + |\alpha_n - \alpha_*|) \to 0,$$

$$\bar{\mathcal{L}}_{2,n}(\boldsymbol{p}_{*,j})^{(u)}/(\mathcal{D}_n + |\alpha_n - \alpha_*|) \to 0, \ \bar{\mathcal{L}}_{2,n}(\boldsymbol{p}_{*,j})^{(u)}/(\mathcal{D}_n + |\alpha_n - \alpha_*|) \to 0,$$

$$\bar{\mathcal{L}}_{3,n}(\boldsymbol{p}_{*,j})^{(uv)}/(\mathcal{D}_n + |\alpha_n - \alpha_*|) \to 0, \ \mathcal{N}_{1,n}(\boldsymbol{p}_{*,j})/(\mathcal{D}_n + |\alpha_n - \alpha_*|) \to 0, \ \bar{\mathcal{N}}_{1,n}((\boldsymbol{p}_{*,j})^{(u)}/(\mathcal{D}_n + |\alpha_n - \alpha_*|) \to 0,$$

$$\bar{\mathcal{N}}_{2,n}(\boldsymbol{p}_{*,j})^{(uv)}/(\mathcal{D}_n + |\alpha_n - \alpha_*|) \to 0, \ \bar{M}_{n,j,0_d}/\mathcal{D}_n \to 0, \ (\alpha_n - \alpha_*)/(\mathcal{D}_n + |\alpha_n - \alpha_*|) \to 0$$

for any $1 \leq u, v \leq d$ and $1 \leq j \leq L$.

As $(\alpha_n - \alpha_*)/(\mathcal{D}_n + |\alpha_n - \alpha_*|) \to 0$, we deduce that

$$\frac{|\alpha_n - \alpha_*|}{(\mathcal{D}_n + |\alpha_n - \alpha_*|)} \to 0. \tag{25}$$

Note that since $\alpha_n \to \alpha_* \neq 0$ as $n \to \infty$, we have $1/\tanh(\alpha_n) \not\to \infty$. Then, as $\bar{M}_{n,j,0_d}/(\mathcal{D}_n + |\alpha_n - \alpha_*|) \to 0$, it implies that

$$\frac{|\sum_{i \in \mathcal{C}_j}\exp(\bar{b}_{n,i}) - \exp(\bar{b}_{*,j})|}{(\mathcal{D}_n + |\alpha_n - \alpha_*|)} = \frac{1}{\tanh(\alpha_n)} \cdot \frac{|\bar{M}_{n,j,0_d}|}{(\mathcal{D}_n + |\alpha_n - \alpha_*|)} \to 0,$$

for any $1 \leq j \leq L$. By varying the index $j$ from 1 to $L$ in these limits and summing them up, we obtain that

$$\frac{\sum_{j=1}^{L}|\sum_{i \in \mathcal{C}_j}\exp(\bar{b}_{n,i}) - \exp(\bar{b}_{*,j})|}{(\mathcal{D}_n + |\alpha_n - \alpha_*|)} \to 0. \tag{26}$$

Now, we consider indices $j \in [L]$ such that $|\mathcal{C}_j| = 1$, i.e., the corresponding Voronoi cell has only one element. From the hypothesis, we have $\mathcal{L}_{2,n}(\boldsymbol{p}_{*,j})^{(u)}/(\mathcal{D}_n + |\alpha_n - \alpha_*|) \to 0$, which leads to $\bar{M}_{n,j,1_u}/(\mathcal{D}_n + |\alpha_n - \alpha_*|) \to 0$. Hence, we find that

$$\frac{\sum_{i \in \mathcal{C}_j}\exp(\bar{b}_{n,i})\|\Delta\boldsymbol{p}_{n,ij}\|}{(\mathcal{D}_n + |\alpha_n - \alpha_*|)} = \frac{\sum_{u=1}^{d}|\bar{M}_{n,j,1_u}|}{\tanh(\alpha_n)(\mathcal{D}_n + |\alpha_n - \alpha_*|)} \to 0.$$

That limit directly implies the following result:

$$\frac{\sum_{j:|\mathcal{C}_j|=1}\sum_{i \in \mathcal{C}_j}\exp(\bar{b}_{n,i})\|\Delta\boldsymbol{p}_{n,ij}\|}{(\mathcal{D}_n + |\alpha_n - \alpha_*|)} \to 0. \tag{27}$$

We now move to the Voronoi cells having more than one element, namely, we consider indices $j \in [L]$ satisfying $|\mathcal{C}_j| > 1$. The limit $\bar{\mathcal{L}}_{3,n}(\boldsymbol{p}_{*,j})^{(uu)}/(\mathcal{D}_n + |\alpha_n - \alpha_*|) \to 0$ induces that

$$\frac{\sum_{i \in \mathcal{C}_j}\exp(\bar{b}_{n,i})\|\Delta\boldsymbol{p}_{n,ij}\|^2}{(\mathcal{D}_n + |\alpha_n - \alpha_*|)} = \frac{\sum_{u=1}^{d}\bar{\mathcal{L}}_{3,n}(\boldsymbol{p}_{*,j})^{(uu)}}{\tanh(\alpha_n)(\mathcal{D}_n + |\alpha_n - \alpha_*|)} \to 0.$$

By varying the indices $j$ in these limits over all the Voronoi cells $\mathcal{C}_j$ having more than one element, we find that

$$\frac{\sum_{j:|\mathcal{C}_j|>1}\sum_{i \in \mathcal{C}_j}\exp(\bar{b}_{n,i})\|\Delta\boldsymbol{p}_{n,ij}\|^2}{(\mathcal{D}_n + |\alpha_n - \alpha_*|)} \to 0. \tag{28}$$

Combining the results from equations (25), (26), (27), and (28) leads to

$$1 = \frac{\mathcal{D}_n + |\alpha_n - \alpha_*|}{\mathcal{D}_n + |\alpha_n - \alpha_*|} \to 0$$

as $n \to \infty$, which cannot hold. As a consequence, at least one of the coefficients of the terms in the formulations of $[Q_n(\boldsymbol{X}) - \sum_{i=1}^{5} \bar{R}_{n,i}(\boldsymbol{X})]/(\mathcal{D}_n + |\alpha_n - \alpha_*|)$ does not go to 0 as $n \to \infty$.

**Step 3 - Application of the Fatou's lemma.** We denote $m_n$ as the maximum of the absolute values of $\mathcal{L}'_{1,n}(\boldsymbol{p}_{*,j})/(\mathcal{D}_n + |\alpha_n - \alpha_*|)$, $\mathcal{L}_{2,n}(\boldsymbol{p}_{*,j})^{(u)}/(\mathcal{D}_n + |\alpha_n - \alpha_*|)$, $\bar{\mathcal{L}}'_{1,n}(\boldsymbol{p}_{*,j})/(\mathcal{D}_n + |\alpha_n - \alpha_*|)$, $\bar{\mathcal{L}}_{2,n}(\boldsymbol{p}_{*,j})^{(u)}/(\mathcal{D}_n + |\alpha_n - \alpha_*|)$, $\bar{\mathcal{L}}_{3,n}(\boldsymbol{p}_{*,j})^{(uv)}/(\mathcal{D}_n + |\alpha_n - \alpha_*|)$, $\mathcal{N}_{1,n}(\boldsymbol{p}_{*,j})/(\mathcal{D}_n + |\alpha_n - \alpha_*|)$, $\bar{\mathcal{N}}_{1,n}((\boldsymbol{p}_{*,j})^{(u)}/(\mathcal{D}_n + |\alpha_n - \alpha_*|)$, $\bar{\mathcal{N}}_{2,n}(\boldsymbol{p}_{*,j})^{(uv)}/(\mathcal{D}_n + |\alpha_n - \alpha_*|)$, $\bar{M}_{n,j,0_d}/(\mathcal{D}_n + |\alpha_n - \alpha_*|)$, and $(\alpha_n - \alpha_*)/(\mathcal{D}_n + |\alpha_n - \alpha_*|)$ for all $1 \le u, v \le d$. From the result of Step 2 in the proof, we have $1/m_n \not\to \infty$ as $n \to \infty$.

Recall that $\|f_{G_n,\alpha_n} - f_{G_*,\alpha_*}\|_{L^2(\mu)}/(\mathcal{D}_n + |\alpha_n - \alpha_*|) \to 0$ as $n \to \infty$, which indicates that $\|f_{G_n,\alpha_n} - f_{G_*,\alpha_*}\|_{L^2(\mu)}/(m_n(\mathcal{D}_n + |\alpha_n - \alpha_*|)) \to 0$. Furthermore, since the $L^2(\mu)$ norm is equivalent to the $L^1(\mu)$ norm, we have $\|f_{G_n,\alpha_n} - f_{G_*,\alpha_*}\|_{L^1(\mu)}/(m_n(\mathcal{D}_n + |\alpha_n - \alpha_*|)) \to 0$. An application of Fatou's lemma leads to

$$0 = \lim_{n\to\infty} \frac{\|f_{G_n,\alpha_n} - f_{G_*,\alpha_*}\|_{L^1(\mu)}}{m_n(\mathcal{D}_n + |\alpha_n - \alpha_*|)} \ge \int \liminf_{n\to\infty} \frac{\|f_{G_n,\alpha_n}(\boldsymbol{X}) - f_{G_*,\alpha_*}(\boldsymbol{X})\|_1}{m_n(\mathcal{D}_n + |\alpha_n - \alpha_*|)} d\mu(\boldsymbol{X}) \ge 0.$$

It indicates that $\liminf_{n\to\infty} \frac{\|f_{G_n}(\boldsymbol{X}) - f_{G_*}(\boldsymbol{X})\|_1}{m_n(\mathcal{D}_n + |\alpha_n - \alpha_*|)} = 0$ for almost surely $\boldsymbol{X}$. As $n \to \infty$, we denote

$$\frac{\mathcal{L}'_{1,n}(\boldsymbol{p}_{*,j})}{m_n(\mathcal{D}_n + |\alpha_n - \alpha_*|)} \to \widehat{\alpha}_j, \qquad \frac{\mathcal{L}_{2,n}(\boldsymbol{p}_{*,j})}{m_n(\mathcal{D}_n + |\alpha_n - \alpha_*|)} \to \beta_j,$$

$$\frac{\bar{\mathcal{L}}'_{1,n}(\boldsymbol{p}_{*,j})}{m_n(\mathcal{D}_n + |\alpha_n - \alpha_*|)} \to \bar{\alpha}_j, \qquad \frac{\bar{\mathcal{L}}_{2,n}(\boldsymbol{p}_{*,j})}{m_n(\mathcal{D}_n + |\alpha_n - \alpha_*|)} \to \bar{\beta}_j, \qquad \frac{\bar{\mathcal{L}}_{3,n}(\boldsymbol{p}_{*,j})}{m_n(\mathcal{D}_n + |\alpha_n - \alpha_*|)} \to \bar{\gamma}_j,$$

$$\frac{\bar{M}_{n,j,0_d}}{m_n(\mathcal{D}_n + |\alpha_n - \alpha_*|)} \to \tilde{\alpha}_j, \qquad \frac{\mathcal{N}_{1,n}(\boldsymbol{p}_{*,j})}{m_n(\mathcal{D}_n + |\alpha_n - \alpha_*|)} \to \tilde{\beta}_j,$$

$$\frac{\bar{\mathcal{N}}_{1,n}(\boldsymbol{p}_{*,j})}{m_n(\mathcal{D}_n + |\alpha_n - \alpha_*|)} \to \widehat{\beta}_j, \qquad \frac{\bar{\mathcal{N}}_{2,n}(\boldsymbol{p}_{*,j})}{m_n(\mathcal{D}_n + |\alpha_n - \alpha_*|)} \to \widehat{\gamma}_j, \qquad \frac{\alpha_n - \alpha_*}{m_n(\mathcal{D}_n + |\alpha_n - \alpha_*|)} \to \tau$$

for any $1 \le j \le L$. Here, from the definition of $m_n$, at least one coefficient among $\{\widehat{\alpha}_j, \beta_j, \tilde{\alpha}_j, \tilde{\beta}_j\}_{j:|\mathcal{C}_j|=1}$, $\{\bar{\alpha}_j, \bar{\beta}_j, \bar{\gamma}_j, \tilde{\alpha}_j, \widehat{\beta}_j, \widehat{\gamma}_j\}_{j:|\mathcal{C}_j|>1}$, and $\tau$ is different from 0. Then, the equation

$$\liminf_{n\to\infty} \frac{\|Q_n(\boldsymbol{X})\|_1}{m_n(\mathcal{D}_n + |\alpha_n - \alpha_*|)} = \Big[ \sum_{k'=1}^{L} \exp((\bar{B}\boldsymbol{p}_{*,k'})^\top \boldsymbol{X} + \bar{b}_{*,k'}) \Big] \cdot \liminf_{n\to\infty} \frac{\|f_{G_n,\alpha_n}(\boldsymbol{X}) - f_{G_*,\alpha_*}(\boldsymbol{X})\|_1}{m_n(\mathcal{D}_n + |\alpha_n - \alpha_*|)} = 0$$

leads to

$$\sum_{j:|\mathcal{C}_j|=1} \exp((\bar{B}\boldsymbol{p}_{*,j})^\top \boldsymbol{X})(\alpha_j + \beta_j^\top(\bar{B}^\top \boldsymbol{X}))$$

$$+ \sum_{j:|\mathcal{C}_j|>1} \exp((\bar{B}\boldsymbol{p}_{*,j})^\top \boldsymbol{X})\big[\bar{\alpha}_j + \bar{\beta}_j^\top(B^\top \boldsymbol{X}) + (\bar{B}^\top \boldsymbol{X})^\top \bar{\gamma}_j(\bar{B}^\top \boldsymbol{X})\big]$$

$$- \sum_{j:|\mathcal{C}_j|=1} \exp((\bar{B}\boldsymbol{p}_{*,j})^\top \boldsymbol{X})(\tilde{\alpha}_j + \tilde{\beta}_j^\top(\bar{B}^\top \boldsymbol{X}))f_{G_*}(\boldsymbol{X})$$

$$- \sum_{j:|\mathcal{C}_j|>1} \exp((\bar{B}\boldsymbol{p}_{*,j})^\top \boldsymbol{X})\big[\tilde{\alpha}_j + \widehat{\beta}_j^\top(\bar{B}^\top \boldsymbol{X}) + (\bar{B}^\top \boldsymbol{X})^\top \widehat{\gamma}_j \bar{B}^\top \boldsymbol{X}\big]f_{G_*}(\boldsymbol{X})$$

$$+ \tau[1 - \tanh^2(\alpha_*)] \cdot \Big[ \sum_{k'=1}^{L} \exp((\bar{B}\boldsymbol{p}_{*,k'})^\top \boldsymbol{X} + \bar{b}_{*,k'}) \Big] \cdot f_{G_*}(\boldsymbol{X}) = \boldsymbol{0}_d$$

for almost surely $\boldsymbol{X}$. That equation only holds if and only if all the coefficients $\{\widehat{\alpha}_j, \beta_j, \tilde{\alpha}_j, \tilde{\beta}_j\}_{j:|\mathcal{C}_j|=1}$,

$\{\bar{\alpha}_j, \bar{\beta}_j, \bar{\gamma}_j, \tilde{\alpha}_j, \widehat{\beta}_j, \widehat{\gamma}_j\}_{j:|\mathcal{C}_j|>1}$, and $\tau$ are 0, which is a contradiction.

It indicates that we indeed have the conclusion of the local part, namely,

$$\lim_{\varepsilon \to 0} \inf_{G \in \mathcal{G}_{L'}(\Omega):(\mathcal{D}(G,G_*)+|\alpha-\alpha_*|)\leq\varepsilon} \|f_{G,\alpha} - f_{G_*,\alpha_*}\|_{L^2(\mu)}/(\mathcal{D}(G,G_*)+|\alpha-\alpha_*|) > 0.$$

**Global part:** The result of the local part implies that we can find a positive constant $\varepsilon'$ such that

$$\inf_{G \in \mathcal{G}_{L'}(\Omega):(\mathcal{D}(G,G_*)+|\alpha-\alpha_*|)\leq\varepsilon'} \|f_{G,\alpha} - f_{G_*,\alpha_*}\|_{L^2(\mu)}/(\mathcal{D}(G,G_*)+|\alpha-\alpha_*|) > 0.$$

Therefore, to obtain the conclusion of the theorem it is sufficient to prove that

$$\inf_{G \in \mathcal{G}_{L'}(\Omega):(\mathcal{D}(G,G_*)+|\alpha-\alpha_*|)>\varepsilon'} \|f_{G,\alpha} - f_{G_*,\alpha_*}\|_{L^2(\mu)}/(\mathcal{D}(G,G_*)+|\alpha-\alpha_*|) > 0.$$

Assume by contrary that the above claim does not hold. Then there exists a sequence of measures $G'_n := \sum_{j'=1}^{L'} \exp(\bar{b}_{n,j'})\delta_{\boldsymbol{p}_{n,j'}}$ in $\mathcal{G}_{L'}(\Theta)$ and $\alpha'_n \in \Omega$ such that we have

$$\begin{cases} \mathcal{D}(G'_n, G_*) + |\alpha'_n - \alpha_*| > \varepsilon' \\ \|f_{G'_n,\alpha'_n} - f_{G_*,\alpha_*}\|_{L^2(\mu)}/(\mathcal{D}(G'_n, G_*) + |\alpha'_n - \alpha_*|) \to 0. \end{cases}$$

These limits indicate that $\|f_{G'_n,\alpha'_n} - f_{G_*,\alpha_*}\|_{L^2(\mu)} \to 0$ as $n \to \infty$.
Recall that the sets $\Theta$ and $\Omega$ are compact. Therefore, there exists a mixing measure $G'$ in $\mathcal{G}_{L'}(\Omega)$ such that one of $(G'_n, \alpha'_n)$'s subsequences converges to $(G', \alpha')$. Since $\mathcal{D}(G'_n, G_*) + |\alpha'_n - \alpha_*| > \varepsilon'$, we deduce that $\mathcal{D}(G', G_*) + |\alpha' - \alpha_*| > \varepsilon'$.
An application of the Fatou's lemma leads to

$$0 = \lim_{n\to\infty} \|f_{G'_n,\alpha'_n} - f_{G_*,\alpha_*}\|_{L^2(\mu)} \geq \int \liminf_{n\to\infty} \|f_{G'_n,\alpha'_n}(\boldsymbol{X}) - f_{G_*,\alpha_*}(\boldsymbol{X})\|_2^2 \, d\mu(\boldsymbol{X}).$$

Hence, we have $f_{G',\alpha'}(\boldsymbol{X}) = f_{G_*,\alpha_*}(\boldsymbol{X})$ for $\mu$−almost surely $\boldsymbol{X}$. From the identifiability property (cf. the end of this proof), we deduce that $(G', \alpha') \equiv (G_*, \alpha_*)$. It follows that $\mathcal{D}(G', G_*) + |\alpha' - \alpha_*| = 0$. It contradicts to the hypothesis that $\mathcal{D}(G', G_*) + |\alpha' - \alpha_*| > \varepsilon' > 0$.
As a consequence, the proof of the global part is completed. We obtain the conclusion of the theorem.

**Proof for the identifiability property.** We now demonstrate that if $f_{\bar{G},\bar{\alpha}}(\boldsymbol{X}) = f_{\bar{G}_*,\bar{\alpha}_*}(\boldsymbol{X})$ for almost every $\boldsymbol{X}$, then we obtain that $(\bar{G}, \bar{\alpha}) \equiv (\bar{G}_*, \bar{\alpha}_*)$.

To ease the presentation we denote the following notations:

$$\text{softmax}_{\bar{G}}^{\text{Pretrain}}(u) = \frac{\exp(u)}{\sum_{k=1}^{N} \exp(\boldsymbol{X}^\top \bar{A}_k^0 \boldsymbol{X} + \bar{a}_k^0)},$$

$$\text{softmax}_{\bar{G}}^{\text{Prompt}}(u') = \frac{\exp(u)}{\sum_{j'=1}^{L'} \exp((\bar{B}\boldsymbol{p}_{j'})^\top \boldsymbol{X} + \bar{b}_{j'})},$$

$$\text{softmax}_{\bar{G}_*}^{\text{Pretrain}}(u_*) = \frac{\exp(u_*)}{\sum_{k=1}^{N} \exp(\boldsymbol{X}^\top \bar{A}_k^0 \boldsymbol{X} + \bar{a}_k^0)},$$

$$\text{softmax}_{\bar{G}_*}^{\text{Prompt}}(u'_*) = \frac{\exp(u_*)}{\sum_{j'=1}^{L} \exp((\bar{B}\boldsymbol{p}_{*,j'})^\top \boldsymbol{X} + \bar{b}_{*,j'})}.$$

Here, $u$, $u'$, $u_*$, and $u'_*$ in these equations satisfy:

$$u \in \{\boldsymbol{X}^\top \bar{A}_j^0 \boldsymbol{X} + \bar{a}_j^0 : j \in [N]\}, \ u' \in \{(\bar{B}\boldsymbol{p}_{j'})^\top \boldsymbol{X} + \bar{b}_{j'} : j' \in [L']\}$$

$$u_* \in \{\boldsymbol{X}^\top \bar{A}_j^0 \boldsymbol{X} + \bar{a}_j^0 : j \in [N]\}, \ u'_* \in \{(\bar{B}\boldsymbol{p}_{*,j'})^\top \boldsymbol{X} + \bar{b}_{*,j'} : j' \in [L]\}.$$

The equation $f_{\bar{G},\bar{\alpha}}(\boldsymbol{X}) = f_{\bar{G}_*,\bar{\alpha}_*}(\boldsymbol{X})$ for almost every $\boldsymbol{X}$ indicates that

$$\sum_{j=1}^{N} \text{softmax}_{\bar{G}}^{\text{Pretrain}}(\boldsymbol{X}^\top \bar{A}_j^0 \boldsymbol{X} + \bar{a}_j^0))h(\boldsymbol{X}, \bar{\eta}_j^0) + \tanh(\bar{\alpha}) \sum_{j'=1}^{L'} \text{softmax}_{\bar{G}}^{\text{Prompt}}((\bar{B}\boldsymbol{p}_{j'})^\top \boldsymbol{X} + \bar{b}_{j'})\bar{C}\boldsymbol{p}_{j'}$$

$$= \sum_{j=1}^{N} \text{softmax}_{\bar{G}_*}^{\text{Pretrain}}(\boldsymbol{X}^\top \bar{A}_j^0 \boldsymbol{X} + \bar{a}_j^0))h(\boldsymbol{X}, \bar{\eta}_j^0) + \tanh(\bar{\alpha}_*) \sum_{j'=1}^{L} \text{softmax}_{\bar{G}_*}^{\text{Prompt}}((\bar{B}\boldsymbol{p}_{*,j'})^\top \boldsymbol{X} + \bar{b}_{*,j'})\bar{C}\boldsymbol{p}_{*,j'}. \quad (29)$$

The above equation only holds when $L = L'$. Furthermore, we also have that

$$\{\text{softmax}_{\bar{G}}^{\text{Prompt}}((\bar{B}\boldsymbol{p}_{j'})^\top \boldsymbol{X} + \bar{b}_{j'}) : j' \in [L']\} = \{\text{softmax}_{\bar{G}_*}^{\text{Prompt}}((\bar{B}\boldsymbol{p}_{*,j'})^\top \boldsymbol{X} + \bar{b}_{*,j'}) : j' \in [L]\},$$

for almost every $\boldsymbol{X}$. By relabelling the indices, we can assume that

$$\text{softmax}_{\bar{G}}^{\text{Prompt}}((\bar{B}\boldsymbol{p}_{j'})^\top \boldsymbol{X} + \bar{b}_{j'}) = \text{softmax}_{\bar{G}_*}^{\text{Prompt}}((\bar{B}\boldsymbol{p}_{*,j'})^\top \boldsymbol{X} + \bar{b}_{*,j'}),$$

for almost every $\boldsymbol{X}$ and any $j' \in [L]$. From the translation invariant property of the softmax function, the above equations only hold when $\bar{b}_{j'} = \bar{b}_{*,j'} + \bar{r}$ for some $\bar{r} \in \mathbb{R}$ and any $j' \in [L]$. Given these results, equation (29) leads to

$$\tanh(\bar{\alpha}) \sum_{j=1}^{L} \exp(\bar{b}_j) \exp((\bar{B}\boldsymbol{p}_j)^\top \boldsymbol{X})\bar{C}\boldsymbol{p}_j = \tanh(\bar{\alpha}_*) \sum_{j=1}^{L} \exp(\bar{b}_{*,j}) \exp((\bar{B}\boldsymbol{p}_{*,j})^\top \boldsymbol{X})\bar{C}\boldsymbol{p}_{*,j}, \quad (30)$$

for almost surely $\boldsymbol{X}$.

Now, we partition the set $\{1, 2, \ldots, L\}$ into $m$ subsets $\bar{K}_1, \bar{K}_2, \ldots, \bar{K}_m$ where $m \leq L$, such that $\exp(\bar{b}_j) = \exp(\bar{b}_{*,j'})$ for any $j, j' \in \bar{K}_i$ and $i \in [m]$. It is clear that $\exp(\bar{b}_j) \neq \exp(\bar{b}_{*,j'})$ when $j, j'$ belong to different subsets $\bar{K}_i$. Collecting these results, equation (30) can be rewritten as follows:

$$\tanh(\bar{\alpha}) \sum_{i=1}^{m} \sum_{j \in \bar{K}_i} \exp(\bar{b}_j) \exp((\bar{B}\boldsymbol{p}_j)^\top \boldsymbol{X})\bar{C}\boldsymbol{p}_j$$

$$= \tanh(\bar{\alpha}_*) \sum_{i=1}^{m} \sum_{j \in \bar{K}_i} \exp(\bar{b}_{*,j}) \exp((\bar{B}\boldsymbol{p}_{*,j})^\top \boldsymbol{X})\bar{C}\boldsymbol{p}_{*,j},$$

for almost surely $\boldsymbol{X}$. Hence, we achieve that

$$\{((\bar{B}\boldsymbol{p}_j)^\top, \boldsymbol{p}_j) : j \in \bar{K}_i\} = \{((\bar{B}\boldsymbol{p}_{*,j})^\top, \boldsymbol{p}_{*,j}) : j \in \bar{K}_i\} \quad \text{and} \quad \tanh(\bar{\alpha}) = \tanh(\bar{\alpha}_*).$$

It naturally leads to

$$\{\boldsymbol{p}_j : j \in \bar{K}_i\} = \{\boldsymbol{p}_{*,j} : j \in \bar{K}_i\}.$$

Without loss of generality, $\boldsymbol{p}_j = \boldsymbol{p}_{*,j}$ for all $j \in \bar{K}_i$. As a consequence, we obtain that $\bar{\alpha} = \bar{\alpha}_*$ and

$$\sum_{i=1}^{m} \sum_{j \in \bar{K}_i} \exp(\bar{b}_j)\delta_{\boldsymbol{p}_j} = \sum_{i=1}^{m} \sum_{j \in \bar{K}_i} \exp(\bar{b}_{*,j})\delta_{\boldsymbol{p}_{*,j}}.$$

It is equivalent to $(\bar{G}, \bar{\alpha}) \equiv (\bar{G}_*, \bar{\alpha}_*)$. We achieve the conclusion of the identifiability claim.

## A.2. Proof of Theorem 4.4

Similar to the proof of Theorem 4.2 in Appendix A.1, we only need to demonstrate that

$$\|g_{G,\alpha} - g_{G_*,\alpha_*}\|_{L^2(\mu)} \geq C \cdot (\mathcal{D}(G, G_*) + |\alpha - \alpha_*|)$$

for any $(G, \alpha) \in \mathcal{G}_{L'}(\Theta) \times \Omega$ for some universal constant $C$. It is equivalent to proving that:

$$\inf_{(G,\alpha) \in \mathcal{G}_{L'}(\Theta) \times \Omega} \|g_{G,\alpha} - g_{G_*,\alpha_*}\|_{L^2(\mu)} / (\mathcal{D}(G, G_*) + |\alpha - \alpha_*|) > 0.$$

To obtain the conclusion for the above inequality, we consider two parts: (i) local part, namely,

$$\lim_{\varepsilon \to 0} \inf_{(G,\alpha) \in \mathcal{G}_{L'}(\Theta) \times \Omega : \mathcal{D}(G,G_*) + |\alpha - \alpha_*| \leq \varepsilon} \|g_{G,\alpha} - g_{G_*,\alpha_*}\|_{L^2(\mu)} / (\mathcal{D}(G, G_*) + |\alpha - \alpha_*|) > 0;$$

(ii) global part, namely, for any $\varepsilon > 0$

$$\inf_{(G,\alpha)\in\mathcal{G}_{L'}(\Theta)\times\Omega:\mathcal{D}(G,G_*)+|\alpha-\alpha_*|>\varepsilon} \|g_{G,\alpha} - g_{G_*,\alpha_*}\|_{L^2(\mu)}/(\mathcal{D}(G,G_*) + |\alpha - \alpha_*|) > 0;$$

Since the global part can be argued in a similar fashion to Appendix A.1, we will focus only on proving the local part in this appendix. Additionally, we will impose the following essential yet mild assumptions in the activation function $\sigma$ to facilitate our arguments:

**Assumptions.** We assume that the activation function $\sigma$ meet the following assumptions:

*(A.1) (Uniform Lipschitz) Let $F(\boldsymbol{X};\boldsymbol{p}) := \exp((B\sigma(\boldsymbol{p}))^\top \boldsymbol{X})C\sigma(\boldsymbol{p})$. Then, for any $r \in \{1,2\}$, we have*

$$\sum_{|\alpha|=r} \left| \left( \frac{\partial^{|\alpha|} F}{\partial \boldsymbol{p}^\alpha}(\boldsymbol{X};\boldsymbol{p}) - \frac{\partial^{|\alpha|} F}{\partial \boldsymbol{p}^\alpha}(\boldsymbol{X};\boldsymbol{p}') \right)\gamma^\alpha \right| \le c\|\boldsymbol{p}-\boldsymbol{p}'\|^\zeta \|\gamma\|^r,$$

*for any vector $\gamma \in \mathbb{R}^d$ and for some positive constants $\zeta$ and $c$ which are independent of $\boldsymbol{X}$ and $\boldsymbol{p}, \boldsymbol{p}'$. Here, $\alpha \in \mathbb{N}^d$.*

*(A.2) (Injective) If there exist parameters $\boldsymbol{p}$ and $\boldsymbol{p}'$ such that $\sigma(\boldsymbol{p}) = \sigma(\boldsymbol{p}')$, then we obtain that $\boldsymbol{p} = \boldsymbol{p}'$.*

**Local part:** We first start with the local part, which is equivalent to demonstrating that

$$\lim_{\varepsilon\to 0} \inf_{(G,\alpha)\in\mathcal{G}_{L'}(\Theta)\times\Omega:\mathcal{D}(G,G_*)+|\alpha-\alpha_*|\le\varepsilon} \|g_{G,\alpha} - g_{G_*,\alpha_*}\|_{L^2(\mu)}/(\mathcal{D}(G,G_*) + |\alpha - \alpha_*|) > 0.$$

We prove the above claim by contradiction. Assume by contrary that the above claim does not hold. It indicates that we can find a sequence of mixing measures $G_n := \sum_{j'=1}^{L_n} \exp(\bar{b}_{n,j'})\delta_{\boldsymbol{p}_{n,j'}}$ in $\mathcal{G}_{L'}(\Theta)$ and a sequence of $\alpha_n \in \Omega$ such that when $n \to \infty$, the following limits hold:

$$\begin{cases} \mathcal{D}(G_n, G_*) + |\alpha_n - \alpha_*| \to 0, \\ \|g_{G_n,\alpha_n} - g_{G_*,\alpha_*}\|_{L^2(\mu)}/(\mathcal{D}(G_n, G_*) + |\alpha_n - \alpha_*|) \to 0. \end{cases}$$

The first limit indicates that $\mathcal{D}_n := \mathcal{D}(G_n, G_*) \to 0$ and $\alpha_n \to \alpha_*$ as $n \to \infty$.

For the simplicity of the ensuing presentation, we denote $\mathcal{C}_j^n := \mathcal{C}_j(G_n)$ as a Voronoi cell of $G_n$ induced by the $j$-th components of $G_*$. Without loss of generality, we assume that those Voronoi cells do not depend on the sample size, i.e., $\mathcal{C}_j = \mathcal{C}_j^n$, which is possible since our arguments are asymptotic. Therefore, we can rewrite the Voronoi loss $\mathcal{D}_n$ as follows:

$$\mathcal{D}_n := \sum_{j'=1}^{L} \left| \sum_{i\in\mathcal{C}_{j'}} \exp(\bar{b}_{n,i}) - \exp(\bar{b}_{*,j'}) \right| + \sum_{j'\in[L]:|\mathcal{C}_{j'}|=1} \sum_{i\in\mathcal{C}_{j'}} \exp(\bar{b}_{n,i})\|\Delta\boldsymbol{p}_{n,ij'}\|$$
$$+ \sum_{j'\in[L]:|\mathcal{C}_{j'}|>1} \sum_{i\in\mathcal{C}_{j'}} \exp(\bar{b}_{n,i})\|\Delta\boldsymbol{p}_{n,ij'}\|^2,$$

where $\Delta\boldsymbol{p}_{n,ij'} = \boldsymbol{p}_{n,i} - \boldsymbol{p}_{*,j'}$ for all $i \in \mathcal{C}_{j'}$.

From the hypothesis, we have $\mathcal{D}_n \to 0$, which implies that $\sum_{i\in\mathcal{C}_j} \exp(\bar{b}_{n,i}) \to \exp(\bar{b}_{*,j})$ and $\boldsymbol{p}_{n,i} \to \boldsymbol{p}_{*,j}$ for any $i \in \mathcal{C}_j, j \in [L]$. Similar to the proof of Theorem 4.2, to establish the contradiction our proof consists of three main steps.

**Step 1 - Taylor expansion.** To ease the presentation, let us denote

$$g_{G_n}(\boldsymbol{X}) := \sum_{j=1}^{L_n} \frac{\exp((\bar{B}\sigma(\boldsymbol{p}_{n,j}))^\top \boldsymbol{X} + \bar{b}_{n,j})}{\sum_{k=1}^{L_n} \exp((\bar{B}\sigma(\boldsymbol{p}_{n,k}))^\top \boldsymbol{X} + \bar{b}_{n,k})} \cdot \bar{C}\sigma(\boldsymbol{p}_{n,j}),$$

$$g_{G_*}(\boldsymbol{X}) := \sum_{j'=1}^{L} \frac{\exp((\bar{B}\sigma(\boldsymbol{p}_{*,j'}))^\top \boldsymbol{X} + \bar{b}_{*,j'})}{\sum_{k'=1}^{L} \exp((\bar{B}\sigma(\boldsymbol{p}_{*,k'}))^\top \boldsymbol{X} + \bar{b}_{*,k'})} \cdot \bar{C}\sigma(\boldsymbol{p}_{*,j'}).$$

We now perform the following decomposition:

$$
\begin{aligned}
\widetilde{Q}_n(\boldsymbol{X}) &:= \Big[\sum_{k'=1}^{L} \exp((\bar{B}\sigma(\boldsymbol{p}_{*,k'}))^\top \boldsymbol{X} + \bar{b}_{*,k'})\Big] \cdot [g_{G_n,\alpha_n}(\boldsymbol{X}) - g_{G_*,\alpha_*}(\boldsymbol{X})] \\
&= \Big[\sum_{k'=1}^{L} \exp((\bar{B}\sigma(\boldsymbol{p}_{*,k'}))^\top \boldsymbol{X} + \bar{b}_{*,k'})\Big] \cdot \Big(\tanh(\alpha_n)g_{G_n}(\boldsymbol{X}) - \tanh(\alpha_*)g_{G_*}(\boldsymbol{X})\Big) \\
&= \Big[\sum_{k'=1}^{L} \exp((\bar{B}\sigma(\boldsymbol{p}_{*,k'}))^\top \boldsymbol{X} + \bar{b}_{*,k'})\Big] \cdot \tanh(\alpha_n)\Big[g_{G_n}(\boldsymbol{X}) - g_{G_*}(\boldsymbol{X})\Big] \\
&\quad + \Big[\sum_{k'=1}^{L} \exp((\bar{B}\sigma(\boldsymbol{p}_{*,k'}))^\top \boldsymbol{X} + \bar{b}_{*,k'})\Big] \cdot [\tanh(\alpha_n) - \tanh(\alpha_*)]g_{G_*}(\boldsymbol{X}) \\
&:= \widetilde{Q}_{n,1}(\boldsymbol{X}) + \widetilde{Q}_{n,2}(\boldsymbol{X}).
\end{aligned}
$$

For that purpose, we will decompose the two terms $\widetilde{Q}_{n,1}(\boldsymbol{X})$ and $\widetilde{Q}_{n,2}(\boldsymbol{X})$, respectively.

**Decomposition of the function $\widetilde{Q}_{n,1}(\boldsymbol{X})$.** We have

$$
\begin{aligned}
\widetilde{Q}_{n,1}(\boldsymbol{X}) &= \sum_{j=1}^{L}\sum_{i\in\mathcal{C}_j} \tanh(\alpha_n)\exp(\bar{b}_{n,i})\Big[\exp((\bar{B}\sigma(\boldsymbol{p}_{n,i}))^\top \boldsymbol{X})\bar{C}\sigma(\boldsymbol{p}_{n,i}) - \exp((\bar{B}\sigma(\boldsymbol{p}_{*,j}))^\top \boldsymbol{X})\bar{C}\sigma(\boldsymbol{p}_{*,j})\Big] \\
&\quad - \sum_{j=1}^{L}\sum_{i\in\mathcal{C}_j} \tanh(\alpha_n)\exp(\bar{b}_{n,i})\Big[\exp((\bar{B}\sigma(\boldsymbol{p}_{n,i}))^\top \boldsymbol{X}) - \exp((\bar{B}\sigma(\boldsymbol{p}_{*,j}))^\top \boldsymbol{X})\Big]g_{G_n}(\boldsymbol{X}) \\
&\quad + \sum_{j=1}^{L} \tanh(\alpha_n)\Big(\sum_{i\in\mathcal{C}_j}\exp(\bar{b}_{n,i}) - \exp(\bar{b}_{*,j})\Big)\exp((\bar{B}\sigma(\boldsymbol{p}_{*,j}))^\top \boldsymbol{X})\Big[\bar{C}\sigma(\boldsymbol{p}_{*,j}) - g_{G_n}(\boldsymbol{X})\Big] \\
&:= \widetilde{A}_n(\boldsymbol{X}) - \widetilde{B}_n(\boldsymbol{X}) + \widetilde{C}_n(\boldsymbol{X}). \tag{31}
\end{aligned}
$$

**Decomposition of the function $\widetilde{A}_n(\boldsymbol{X})$.** To ease the presentation, we define the following functions $\widetilde{U}(\boldsymbol{X};\boldsymbol{p}) := \exp((\bar{B}\sigma(\boldsymbol{p}))^\top \boldsymbol{X})$ and $\widetilde{V}(\boldsymbol{p}) = \bar{C}\sigma(\boldsymbol{p})$. Then, we denote the product of these functions as $\widetilde{F}(\boldsymbol{X};\boldsymbol{p}) = \widetilde{U}(\boldsymbol{X};\boldsymbol{p})\widetilde{V}(\boldsymbol{p})$. To decompose $\widetilde{A}_n(\boldsymbol{X})$, we separately consider Voronoi cells with exactly one element and those with more than one element. It leads to the following decomposition of the function $\widetilde{A}_n(\boldsymbol{X})$:

$$
\begin{aligned}
\widetilde{A}_n(\boldsymbol{X}) &= \sum_{j:|\mathcal{C}_j|=1}\sum_{i\in\mathcal{C}_j} \tanh(\alpha_n)\exp(\bar{b}_{n,i})\Big[\widetilde{F}(\boldsymbol{X};\boldsymbol{p}_{n,i}) - \widetilde{F}(\boldsymbol{X};\boldsymbol{p}_{*,j})\Big] \\
&\quad + \sum_{j:|\mathcal{C}_j|>1}\sum_{i\in\mathcal{C}_j} \tanh(\alpha_n)\exp(\bar{b}_{n,i})\Big[\widetilde{F}(\boldsymbol{X};\boldsymbol{p}_{n,i}) - \widetilde{F}(\boldsymbol{X};\boldsymbol{p}_{*,j})\Big] \\
&:= \widetilde{A}_{n,1}(\boldsymbol{X}) + \widetilde{A}_{n,2}(\boldsymbol{X}),
\end{aligned}
$$

where we denote $\widetilde{A}_{n,1}(\boldsymbol{X}) = \sum_{j:|\mathcal{C}_j|=1}\sum_{i\in\mathcal{C}_j} \tanh(\alpha_n)\exp(\bar{b}_{n,i})\Big[\widetilde{F}(\boldsymbol{X};\boldsymbol{p}_{n,i}) - \widetilde{F}(\boldsymbol{X};\boldsymbol{p}_{*,j})\Big]$ and $\widetilde{A}_{n,2}(\boldsymbol{X}) = \sum_{j:|\mathcal{C}_j|>1}\sum_{i\in\mathcal{C}_j} \tanh(\alpha_n)\exp(\bar{b}_{n,i})\Big[\widetilde{F}(\boldsymbol{X};\boldsymbol{p}_{n,i}) - \widetilde{F}(\boldsymbol{X};\boldsymbol{p}_{*,j})\Big]$.

For the function $\widetilde{A}_{n,1}(\boldsymbol{X})$, for any indices $i\in\mathcal{C}_j$ and $j$ such that $|\mathcal{C}_j|=1$, the first-order Taylor expansion entails that

$$
\widetilde{U}(\boldsymbol{X};\boldsymbol{p}_{n,i}) = \widetilde{U}(\boldsymbol{X};\boldsymbol{p}_{*,j}) + \sum_{|\alpha|=1}(\Delta\boldsymbol{p}_{n,ij})^\alpha \frac{\partial^{|\alpha|}\widetilde{U}}{\partial\boldsymbol{p}^\alpha}(\boldsymbol{X};\boldsymbol{p}_{*,j}) + \widetilde{R}_{ij,1}(\boldsymbol{X}),
$$

$$
\widetilde{V}(\boldsymbol{p}_{n,i}) = \widetilde{V}(\boldsymbol{p}_{*,j}) + \sum_{|\alpha|=1}(\Delta\boldsymbol{p}_{n,ij})^\alpha \frac{\partial^{|\alpha|}\widetilde{V}}{\partial\boldsymbol{p}^\alpha}(\boldsymbol{p}_{*,j}) + \widetilde{R}_{ij,2},
$$

where the terms $\widetilde{R}_{ij,1}(\boldsymbol{X})$ and $\widetilde{R}_{ij,2}$ are Taylor remainders.

Combining the above results leads to the following formulation of the function $\widetilde{A}_{n,1}(\boldsymbol{X})$:

$$
\widetilde{A}_{n,1}(\boldsymbol{X}) = \sum_{j:|\mathcal{C}_j|=1} \sum_{i \in \mathcal{C}_j} \frac{\tanh(\alpha_n)\exp(\bar{b}_{n,i})}{\alpha!} \sum_{|\alpha|=1} \left\{ (\Delta \boldsymbol{p}_{n,ij})^\alpha \frac{\partial^{|\alpha|}\widetilde{U}}{\partial \boldsymbol{p}^\alpha}(\boldsymbol{X};\boldsymbol{p}_{*,j})\widetilde{V}(\boldsymbol{p}_{*,j}) \right.
$$
$$
\left. + (\Delta \boldsymbol{p}_{n,ij})^\alpha \frac{\partial^{|\alpha|}\widetilde{V}}{\partial \boldsymbol{p}^\alpha}(\boldsymbol{p}_{*,j})\widetilde{U}(\boldsymbol{X};\boldsymbol{p}_{*,j}) \right\} + \widetilde{R}_{n,1}(\boldsymbol{X})
$$
$$
= \sum_{j:|\mathcal{C}_j|=1} \sum_{|\alpha|=1} \left\{ \bar{M}_{n,j,\alpha} \frac{\partial^{|\alpha|}\widetilde{U}}{\partial \boldsymbol{p}^\alpha}(\boldsymbol{X};\boldsymbol{p}_{*,j})\widetilde{V}(\boldsymbol{p}_{*,j}) + \bar{M}_{n,j,\alpha} \frac{\partial^{|\alpha|}\widetilde{V}}{\partial \boldsymbol{p}^\alpha}(\boldsymbol{p}_{*,j})\widetilde{U}(\boldsymbol{X};\boldsymbol{p}_{*,j}) \right\} + \widetilde{R}_{n,1}(\boldsymbol{X})
$$

where the function $\widetilde{R}_{n,1}(\boldsymbol{X})$ is the combination of Taylor remainders and satisfies that $\widetilde{R}_{n,1}(\boldsymbol{X})/(\mathcal{D}_n + |\alpha_n - \alpha_*|) \to 0$ when $n \to \infty$. Furthermore, the formulations of $\bar{M}_{n,j,\alpha}$ are as follows:

$$
\bar{M}_{n,j,\alpha} = \sum_{i \in \mathcal{C}_j} \frac{\tanh(\alpha_n)\exp(\bar{b}_{n,i})}{\alpha!}(\Delta \boldsymbol{p}_{n,ij})^\alpha,
$$

for any $|\alpha| = 1$.

Moving to the function $\widetilde{A}_{n,2}(\boldsymbol{X})$, the second-order Taylor expansions of the function $\widetilde{U}(\boldsymbol{X};\boldsymbol{p}_{n,i})$ around the function $\widetilde{U}(\boldsymbol{X};\boldsymbol{p}_{*,j})$ and the function $\widetilde{V}(\boldsymbol{p}_{n,i})$ around the function $\widetilde{V}(\boldsymbol{p}_{*,j})$ for any indices $i \in \mathcal{C}_j$ and $j$ such that $|\mathcal{C}_j| > 1$, we obtain the following formulation of the function $\widetilde{A}_{n,2}(\boldsymbol{X})$:

$$
\widetilde{A}_{n,2}(\boldsymbol{X}) = \sum_{j:|\mathcal{C}_j|>1} \sum_{1 \le |\alpha| \le 2} \left\{ \bar{M}_{n,j,\alpha} \frac{\partial^{|\alpha|}\widetilde{U}}{\partial \boldsymbol{p}^\alpha}(\boldsymbol{X};\boldsymbol{p}_{*,j})\widetilde{V}(\boldsymbol{p}_{*,j}) + \bar{M}_{n,j,\alpha} \frac{\partial^{|\alpha|}\widetilde{V}}{\partial \boldsymbol{p}^\alpha}(\boldsymbol{p}_{*,j})\widetilde{U}(\boldsymbol{X};\boldsymbol{p}_{*,j}) \right\}
$$
$$
+ \sum_{j:|\mathcal{C}_j|>1} \sum_{|\alpha|=1,|\beta|=1} \bar{M}_{n,j,\alpha,\beta} \frac{\partial^{|\alpha|}\widetilde{U}}{\partial \boldsymbol{p}^\alpha}(\boldsymbol{X};\boldsymbol{p}_{*,j}) \frac{\partial^{|\beta|}\widetilde{V}}{\partial \boldsymbol{p}^\beta}(\boldsymbol{p}_{*,j}) + \widetilde{R}_{n,2}(\boldsymbol{X})
$$

where the function $\widetilde{R}_{n,2}(\boldsymbol{X})$ is a combination of Taylor remainders and satisfies $\widetilde{R}_{n,2}(\boldsymbol{X})/(\mathcal{D}_n + |\alpha_n - \alpha_*|) \to 0$ when $n \to \infty$. Furthermore, we define

$$
\bar{M}_{n,j,\alpha} = \sum_{i \in \mathcal{C}_j} \frac{\tanh(\alpha_n)\exp(b_{n,i})}{\alpha!}(\Delta \boldsymbol{p}_{n,ij})^\alpha,
$$

for any $|\alpha| = 2$ and

$$
\bar{M}_{n,j,\alpha,\beta} = \sum_{i \in \mathcal{C}_j} \frac{\tanh(\alpha_n)\exp(\bar{b}_{n,i})}{\alpha!\beta!}(\Delta \boldsymbol{p}_{n,ij})^{\alpha+\beta},
$$

for any $|\alpha| = |\beta| = 1$. From the formulations of the functions $\widetilde{U}(\boldsymbol{X};\boldsymbol{p})$ and $\widetilde{V}(\boldsymbol{p})$, we obtain the following explicit forms of their partial derivatives:

$$
\frac{\partial \widetilde{U}}{\partial \boldsymbol{p}^{(u)}}(\boldsymbol{X};\boldsymbol{p}) = \exp((\bar{B}\sigma(\boldsymbol{p}))^\top \boldsymbol{X})\left(\bar{B}\frac{\partial \sigma}{\partial \boldsymbol{p}^{(u)}}(\boldsymbol{p})\right)^\top \boldsymbol{X},
$$
$$
\frac{\partial^2 \widetilde{U}}{\partial \boldsymbol{p}^{(u)} \partial \boldsymbol{p}^{(v)}}(\boldsymbol{X};\boldsymbol{p}) = \exp((\bar{B}\sigma(\boldsymbol{p}))^\top \boldsymbol{X})\left\{ \left(\bar{B}\frac{\partial^2 \sigma}{\partial \boldsymbol{p}^{(u)} \partial \boldsymbol{p}^{(v)}}(\boldsymbol{p})\right)^\top \boldsymbol{X} + \boldsymbol{X}^\top \left(\bar{B}\frac{\partial \sigma}{\partial \boldsymbol{p}^{(u)}}(\boldsymbol{p})\right)\left(\bar{B}\frac{\partial \sigma}{\partial \boldsymbol{p}^{(v)}}(\boldsymbol{p})\right)^\top \boldsymbol{X} \right\},
$$
$$
\frac{\partial \widetilde{V}}{\partial \boldsymbol{p}^{(u)}}(\boldsymbol{p}) = \bar{C}\frac{\partial \sigma}{\partial \boldsymbol{p}^{(u)}}(\boldsymbol{p}),
$$
$$
\frac{\partial^2 \widetilde{V}}{\partial \boldsymbol{p}^{(u)} \partial \boldsymbol{p}^{(v)}}(\boldsymbol{p}) = \bar{C}\frac{\partial^2 \sigma}{\partial \boldsymbol{p}^{(u)} \partial \boldsymbol{p}^{(v)}}(\boldsymbol{p}).
$$

Plugging these explicit formulations of the derivatives of the functions $\widetilde{U}(\boldsymbol{X};\boldsymbol{p})$ and $\widetilde{V}(\boldsymbol{p})$, we can express the functions

$\widetilde{A}_{n,1}(\boldsymbol{X})$ and $\widetilde{A}_{n,2}(\boldsymbol{X})$ as follows:

$$\widetilde{A}_{n,1}(\boldsymbol{X}) = \sum_{j:|\mathcal{C}_j|=1} \exp((\bar{B}\sigma(\boldsymbol{p}_{*,j}))^\top \boldsymbol{X})\big[\mathcal{L}_{1,n}(\boldsymbol{p}_{*,j}) + \mathcal{L}_{2,n}(\boldsymbol{p}_{*,j})^\top \bar{B}^\top \boldsymbol{X}\big) + \widetilde{R}_{n,1}(\boldsymbol{X}),$$

$$\widetilde{A}_{n,2}(\boldsymbol{X}) = \sum_{j:|\mathcal{C}_j|>1} \exp((\bar{B}\sigma(\boldsymbol{p}_{*,j}))^\top \boldsymbol{X})\big[\bar{\mathcal{L}}_{1,n}(\boldsymbol{p}_{*,j}) + \bar{\mathcal{L}}_{2,n}(\boldsymbol{p}_{*,j})^\top \bar{B}^\top \boldsymbol{X}$$

$$+ (\bar{B}^\top \boldsymbol{X})^\top \bar{\mathcal{L}}_{3,n}(\boldsymbol{p}_{*,j})\bar{B}^\top \boldsymbol{X}\big] + \widetilde{R}_{n,2}(\boldsymbol{X}),$$

where the functions $\mathcal{L}_{1,n}(\boldsymbol{p}), \mathcal{L}_{2,n}(\boldsymbol{p}), \bar{\mathcal{L}}_{1,n}(\boldsymbol{p}), \bar{\mathcal{L}}_{2,n}(\boldsymbol{p})$, and $\bar{\mathcal{L}}_{3,n}(\boldsymbol{p})$ are defined as follows:

$$\mathcal{L}_{1,n}(\boldsymbol{p}) = \sum_{u=1}^{d} \bar{M}_{n,j,1_u}\bar{C}\frac{\partial\sigma}{\partial\boldsymbol{p}^{(u)}}(\boldsymbol{p}),$$

$$\mathcal{L}_{2,n}(\boldsymbol{p}) = \sum_{u=1}^{d} \bar{M}_{n,j,1_u}\frac{\partial\sigma}{\partial\boldsymbol{p}^{(u)}}(\boldsymbol{p})\bar{C}\sigma(\boldsymbol{p}),$$

$$\bar{\mathcal{L}}_{1,n}(\boldsymbol{p}) = \sum_{1\le u,v\le d} \bar{M}_{n,j,1_{uv}}\bar{C}\frac{\partial^2\sigma}{\partial\boldsymbol{p}^{(u)}\partial\boldsymbol{p}^{(v)}}(\boldsymbol{p}) = \sum_{u=1}^{d} M_{n,j,1_{uu}}\bar{C}\frac{\partial^2\sigma}{\partial\boldsymbol{p}^{(u)}\partial\boldsymbol{p}^{(u)}}(\boldsymbol{p}),$$

$$\bar{\mathcal{L}}_{2,n}(\boldsymbol{p}) = \sum_{u=1}^{d} \bar{M}_{n,j,1_u}\frac{\partial\sigma}{\partial\boldsymbol{p}^{(u)}}(\boldsymbol{p})\bar{C}\sigma(\boldsymbol{p}) + \sum_{1\le u,v\le d}\big[\bar{M}_{n,j,1_v,1_u}\bar{C}\frac{\partial\sigma}{\partial\boldsymbol{p}^{(u)}}(\boldsymbol{p})\frac{\partial\sigma}{\partial\boldsymbol{p}^{(v)}}(\boldsymbol{p})$$

$$+ \bar{M}_{n,j,1_{uv}}\frac{\partial^2\sigma}{\partial\boldsymbol{p}^{(u)}\partial\boldsymbol{p}^{(v)}}(\boldsymbol{p})\bar{C}\sigma(\boldsymbol{p})\big],$$

$$\bar{\mathcal{L}}_{3,n}(\boldsymbol{p}) = \sum_{1\le u,v\le d} \bar{M}_{n,j,1_{uv}}\frac{\partial\sigma}{\partial\boldsymbol{p}^{(u)}}(\boldsymbol{p})(\frac{\partial\sigma}{\partial\boldsymbol{p}^{(v)}}(\boldsymbol{p}))^\top \bar{C}\sigma(\boldsymbol{p}).$$

In these formulations, we use $1_u$ to denote the vector that its $u$-th element is 1 and its other elements are 0 for $1 \le u \le d$. Furthermore, we denote $1_{uv}$ as the matrix that its $(u,v)$-th element is 1 and its other elements are 0 for any $1 \le u, v \le d$.

**Decomposition of the function $\widetilde{B}_n(\boldsymbol{X})$.** Similar to the decomposition of the function $\widetilde{A}_n(\boldsymbol{X})$, we can decompose the function $\widetilde{B}_n(\boldsymbol{X})$ as follows:

$$\widetilde{B}_n(\boldsymbol{X}) = \sum_{j:|\mathcal{C}_j|=1}\sum_{i\in\mathcal{C}_j} \tanh(\alpha_n)\exp(\bar{b}_{n,i})\big[\widetilde{U}(\boldsymbol{X};\boldsymbol{p}_{n,i}) - \widetilde{U}(\boldsymbol{X};\boldsymbol{p}_{*,j})\big]g_{G_n}(\boldsymbol{X})$$

$$+ \sum_{j:|\mathcal{C}_j|>1}\sum_{i\in\mathcal{C}_j} \tanh(\alpha_n)\exp(\bar{b}_{n,i})\big[\widetilde{U}(\boldsymbol{X};\boldsymbol{p}_{n,i}) - \widetilde{U}(\boldsymbol{X};\boldsymbol{p}_{*,j})\big]g_{G_n}(\boldsymbol{X})$$

$$:= \widetilde{B}_{n,1}(\boldsymbol{X}) + \widetilde{B}_{n,2}(\boldsymbol{X})$$

where we denote $\widetilde{B}_{n,1}(\boldsymbol{X}) = \sum_{j:|\mathcal{C}_j|=1}\sum_{i\in\mathcal{C}_j} \tanh(\alpha_n)\exp(\bar{b}_{n,i})\big[\widetilde{U}(\boldsymbol{X};\boldsymbol{p}_{n,i}) - \widetilde{U}(\boldsymbol{X};\boldsymbol{p}_{*,j})\big]g_{G_n}(\boldsymbol{X})$ and $\bar{B}_{n,2}(\boldsymbol{X}) = \sum_{j:|\mathcal{C}_j|>1}\sum_{i\in\mathcal{C}_j} \tanh(\alpha_n)\exp(\bar{b}_{n,i})\big[\widetilde{U}(\boldsymbol{X};\boldsymbol{p}_{n,i}) - \widetilde{U}(\boldsymbol{X};\boldsymbol{p}_{*,j})\big]g_{G_n}(\boldsymbol{X})$.

Similar to the Taylor expansions for the functions $\widetilde{A}_{n,1}(\boldsymbol{X})$ and $\widetilde{A}_{n,2}(\boldsymbol{X})$, by using the first-order Taylor expansion to $\widetilde{B}_{n,1}(\boldsymbol{X})$ and the second-order Taylor expansion to $\widetilde{B}_{n,2}(\boldsymbol{X})$, we obtain that

$$\widetilde{B}_{n,1}(\boldsymbol{X}) = \sum_{j:|\mathcal{C}_j|=1}\sum_{|\alpha|=1} \bar{M}_{n,j,\alpha}\frac{\partial^{|\alpha|}\widetilde{U}}{\partial\boldsymbol{p}^\alpha}(\boldsymbol{X};\boldsymbol{p}_{*,j})g_{G_n}(\boldsymbol{X}) + \widetilde{R}_{n,3}(\boldsymbol{X}),$$

$$\widetilde{B}_{n,2}(\boldsymbol{X}) = \sum_{j:|\mathcal{C}_j|=1}\sum_{1\le|\alpha|\le2} \bar{M}_{n,j,\alpha}\frac{\partial^{|\alpha|}\widetilde{U}}{\partial\boldsymbol{p}^\alpha}(\boldsymbol{X};\boldsymbol{p}_{*,j})g_{G_n}(\boldsymbol{X}) + \widetilde{R}_{n,4}(\boldsymbol{X})$$

where the functions $\widetilde{R}_{n,3}(\boldsymbol{X}), \widetilde{R}_{n,4}(\boldsymbol{X})$ are Taylor remainders. Furthermore, they satisfy that $\widetilde{R}_{n,3}(\boldsymbol{X})/(\mathcal{D}_n + |\alpha_n - \alpha_*|) \to$

$0$ and $\widetilde{R}_{n,4}(\boldsymbol{X})/(\mathcal{D}_n + |\alpha_n - \alpha_*|) \to 0$ when $n \to \infty$. Given the explicit formulations of the derivatives of the functions $\widetilde{U}(\boldsymbol{X}; \boldsymbol{p})$ and $\widetilde{V}(\boldsymbol{p})$, the functions $\widetilde{B}_{n,1}(\boldsymbol{X})$ and $\widetilde{B}_{n,2}(\boldsymbol{X})$ can be then rewritten as follows:

$$\widetilde{B}_{n,1}(\boldsymbol{X}) = \sum_{j:|\mathcal{C}_j|=1} \exp((\bar{B}\sigma(\boldsymbol{p}_{*,j}))^\top \boldsymbol{X})\mathcal{N}_{1,n}(\boldsymbol{p}_{*,j})^\top \boldsymbol{X} g_{G_n}(\boldsymbol{X}) + \widetilde{R}_{n,3}(\boldsymbol{X}),$$

$$\widetilde{B}_{n,2}(\boldsymbol{X}) = \sum_{j:|\mathcal{C}_j|>1} \exp((\bar{B}\sigma(\boldsymbol{p}_{*,j}))^\top \boldsymbol{X})\big[\bar{\mathcal{N}}_{1,n}(\boldsymbol{p}_{*,j})^\top \bar{B}^\top \boldsymbol{X}$$
$$+ (\bar{B}^\top \boldsymbol{X})^\top \bar{\mathcal{N}}_{2,n}(\boldsymbol{p}_{*,j})(\bar{B}^\top \boldsymbol{X})\big] g_{G_n}(\boldsymbol{X}) + \widetilde{R}_{n,4}(\boldsymbol{X}).$$

Here, the functions $\mathcal{N}_{1,n}(\boldsymbol{X})$, $\bar{\mathcal{N}}_{1,n}(\boldsymbol{X})$, and $\bar{\mathcal{N}}_{2,n}(\boldsymbol{X})$ have the following formulations:

$$\mathcal{N}_{1,n}(\boldsymbol{p}) = \sum_{u=1}^d \bar{M}_{n,j,1_u} \frac{\partial \sigma}{\partial \boldsymbol{p}^{(u)}}(\boldsymbol{p}),$$

$$\bar{\mathcal{N}}_{1,n}(\boldsymbol{p}) = \sum_{u=1}^d \bar{M}_{n,j,1_u} \frac{\partial \sigma}{\partial \boldsymbol{p}^{(u)}}(\boldsymbol{p}) + \sum_{1 \le u,v \le d} \bar{M}_{n,j,1_{uv}} \frac{\partial^2 \sigma}{\partial \boldsymbol{p}^{(u)} \partial \boldsymbol{p}^{(v)}}(\boldsymbol{p}),$$

$$\bar{\mathcal{N}}_{2,n}(\boldsymbol{p}) = \sum_{1 \le u,v \le d} \bar{M}_{n,j,1_{uv}} \frac{\partial \sigma}{\partial \boldsymbol{p}^{(u)}}(\boldsymbol{p}) \frac{\partial \sigma}{\partial \boldsymbol{p}^{(v)}}(\boldsymbol{p})^\top.$$

Collecting all of the above results with the decomposition of the functions $\widetilde{A}_n(\boldsymbol{X})$ and $\widetilde{B}_n(\boldsymbol{X})$, we can represent the function $\widetilde{Q}_{n,1}(\boldsymbol{X})$ in equation (22) as follows:

$$\widetilde{Q}_{n,1}(\boldsymbol{X}) = \sum_{j:|\mathcal{C}_j|=1} \exp((\bar{B}\sigma(\boldsymbol{p}_{*,j}))^\top \boldsymbol{X})\big[\mathcal{L}'_{1,n}(\boldsymbol{p}_{*,j}) + \mathcal{L}_{2,n}(\boldsymbol{p}_{*,j})^\top \bar{B}^\top \boldsymbol{X}\big]$$
$$+ \sum_{j:|\mathcal{C}_j|>1} \exp((\bar{B}\sigma(\boldsymbol{p}_{*,j}))^\top \boldsymbol{X})\big[\bar{\mathcal{L}}'_{1,n}(\boldsymbol{p}_{*,j}) + \bar{\mathcal{L}}_{2,n}(\boldsymbol{p}_{*,j})^\top \bar{B}^\top \boldsymbol{X} + (\bar{B}^\top \boldsymbol{X})^\top \bar{\mathcal{L}}_{3,n}(\boldsymbol{p}_{*,j})\bar{B}^\top \boldsymbol{X}\big]$$
$$- \sum_{j:|\mathcal{C}_j|=1} \exp((\bar{B}\sigma(\boldsymbol{p}_{*,j}))^\top \boldsymbol{X})\big[\bar{M}_{n,j,0_d} + \mathcal{N}_{1,n}(\boldsymbol{p}_{*,j})^\top \bar{B}^\top \boldsymbol{X}\big] g_{G_n}(\boldsymbol{X})$$
$$- \sum_{j:|\mathcal{C}_j|>1} \exp((\bar{B}\sigma(\boldsymbol{p}_{*,j}))^\top \boldsymbol{X})\big[\bar{M}_{n,j,0_d} + \bar{\mathcal{N}}_{1,n}(\boldsymbol{p}_{*,j})^\top \bar{B}^\top \boldsymbol{X} + (\bar{B}^\top \boldsymbol{X})^\top \bar{\mathcal{N}}_{2,n}(\boldsymbol{p}_{*,j})\bar{B}^\top \boldsymbol{X}\big] g_{G_n}(\boldsymbol{X})$$
$$+ \widetilde{R}_{n,1}(\boldsymbol{X}) + \widetilde{R}_{n,2}(\boldsymbol{X}) - \widetilde{R}_{n,3}(\boldsymbol{X}) - \widetilde{R}_{n,4}(\boldsymbol{X}) \tag{32}$$

where we define $\bar{M}_{n,j,0_d} = \tanh(\alpha_n)\left(\sum_{i \in \mathcal{C}_j} \exp(\bar{b}_{n,i}) - \exp(\bar{b}_{*,j})\right)$ for any index $1 \le j \le L$, $\mathcal{L}'_{1,n}(\boldsymbol{p}_{*,j}) = \mathcal{L}_{1,n}(\boldsymbol{p}_{*,j}) + \bar{M}_{n,j,0_d}\bar{C}\sigma(\boldsymbol{p}_{*,j})$, and $\bar{\mathcal{L}}'_{1,n}(\boldsymbol{p}_{*,j}) = \bar{\mathcal{L}}_{1,n}(\boldsymbol{p}_{*,j}) + \bar{M}_{n,j,0_d}\bar{C}\sigma(\boldsymbol{p}_{*,j})$.

**Decomposition of the function $\widetilde{Q}_{n,2}(\boldsymbol{X})$.** An application of the first-order Taylor expansion leads to the following expression for the function $\widetilde{Q}_{n,2}(\boldsymbol{X})$:

$$\widetilde{Q}_{n,2}(\boldsymbol{X}) = [\tanh(\alpha_n) - \tanh(\alpha_*)] \cdot \Big[ \sum_{k'=1}^L \exp((\bar{B}\sigma(\boldsymbol{p}_{*,k'}))^\top \boldsymbol{X} + \bar{b}_{*,k'}) \Big] \cdot g_{G_*}(\boldsymbol{X})$$

$$= (\alpha_n - \alpha_*)[1 - \tanh^2(\alpha_*)] \cdot \Big[ \sum_{k'=1}^L \exp((\bar{B}\sigma(\boldsymbol{p}_{*,k'}))^\top \boldsymbol{X} + \bar{b}_{*,k'}) \Big] \cdot g_{G_*}(\boldsymbol{X}) + \widetilde{R}_{n,5}(\boldsymbol{X}), \tag{33}$$

where the function $\widetilde{R}_{n,5}(\boldsymbol{X})$ is Taylor remainder and satisfies that $\widetilde{R}_{n,5}(\boldsymbol{X})/(\mathcal{D}_n + |\alpha_n - \alpha_*|) \to 0$ as $n \to \infty$.

**Step 2 - Non-vanishing coefficients.** From the results of equations (32) and (33), we can express $[\widetilde{Q}_n(\boldsymbol{X}) - \sum_{i=1}^5 \widetilde{R}_{n,i}(\boldsymbol{X})]/(\mathcal{D}_n + |\alpha_n - \alpha_*|)$ as a combination of the linearly independent terms

$$\exp((\bar{B}\sigma(\boldsymbol{p}_{*,j}))^\top \boldsymbol{X}), \ (\bar{B}^\top \boldsymbol{X})^{(u)} \exp((\bar{B}\sigma(\boldsymbol{p}_{*,j}))^\top \boldsymbol{X}),$$

$$(\bar{B}^\top \boldsymbol{X})^{(u)} (\bar{B}^\top \boldsymbol{X})^{(v)} \exp((\bar{B}\sigma(\boldsymbol{p}_{*,j}))^\top \boldsymbol{X}), \; \exp((\bar{B}\sigma(\boldsymbol{p}_{*,j}))^\top \boldsymbol{X}) g_{G_n}(\boldsymbol{X}),$$

$$(\bar{B}^\top \boldsymbol{X})^{(u)} \exp((\bar{B}\sigma(\boldsymbol{p}_{*,j}))^\top \boldsymbol{X}) g_{G_n}(\boldsymbol{X}), \; (\bar{B}^\top \boldsymbol{X})^{(u)} (\bar{B}^\top \boldsymbol{X})^{(v)} \exp((\bar{B}\sigma(\boldsymbol{p}_{*,j}))^\top \boldsymbol{X}) g_{G_n}(\boldsymbol{X}),$$

$$[1 - \tanh^2(\alpha_*)] \cdot \Big[ \sum_{k'=1}^{L} \exp((\bar{B}\sigma(\boldsymbol{p}_{*,k'}))^\top \boldsymbol{X} + \bar{b}_{*,k'}) \Big] \cdot g_{G_*}(\boldsymbol{X}),$$

for any $1 \leq j \leq L$ and $1 \leq u, v \leq d$.

Our claim is that at least one of the coefficients of these linearly independent terms in the formulation of $[\widetilde{Q}_n(\boldsymbol{X}) - \sum_{i=1}^{5} R_{n,i}(\boldsymbol{X})]/(\mathcal{D}_n + |\alpha_n - \alpha_*|)$ does not go to 0 as $n \to \infty$. Assume by contrary that this claim does not hold, which means that all the coefficients of these linearly independent terms go to 0 as $n \to \infty$. Therefore, as $n \to \infty$ we obtain that

$$\mathcal{L}_{1,n}(\boldsymbol{p}_{*,j})/(\mathcal{D}_n + |\alpha_n - \alpha_*|) \to 0, \; \mathcal{L}_{2,n}(\boldsymbol{p}_{*,j})^{(u)}/(\mathcal{D}_n + |\alpha_n - \alpha_*|) \to 0, \; \bar{\mathcal{L}}_{1,n}(\boldsymbol{p}_{*,j})/(\mathcal{D}_n + |\alpha_n - \alpha_*|) \to 0,$$

$$\bar{\mathcal{L}}_{2,n}(\boldsymbol{p}_{*,j})^{(u)}/(\mathcal{D}_n + |\alpha_n - \alpha_*|) \to 0, \; \bar{\mathcal{L}}_{2,n}(\boldsymbol{p}_{*,j})^{(u)}/(\mathcal{D}_n + |\alpha_n - \alpha_*|) \to 0,$$

$$\bar{\mathcal{L}}_{3,n}(\boldsymbol{p}_{*,j})^{(uv)}/(\mathcal{D}_n + |\alpha_n - \alpha_*|) \to 0, \; \mathcal{N}_{1,n}(\boldsymbol{p}_{*,j})/(\mathcal{D}_n + |\alpha_n - \alpha_*|) \to 0, \; \bar{\mathcal{N}}_{1,n}((\boldsymbol{p}_{*,j})^{(u)}/(\mathcal{D}_n + |\alpha_n - \alpha_*|) \to 0,$$

$$\bar{\mathcal{N}}_{2,n}(\boldsymbol{p}_{*,j})^{(uv)}/(\mathcal{D}_n + |\alpha_n - \alpha_*|) \to 0, \; \bar{M}_{n,j,0_d}/\mathcal{D}_n \to 0, \; (\alpha_n - \alpha_*)/(\mathcal{D}_n + |\alpha_n - \alpha_*|) \to 0$$

for any $1 \leq u, v \leq d$ and $1 \leq j \leq L$.

Since $(\alpha_n - \alpha_*)/(\mathcal{D}_n + |\alpha_n - \alpha_*|)$, we obtain that

$$\frac{|\alpha_n - \alpha_*|}{(\mathcal{D}_n + |\alpha_n - \alpha_*|)} \to 0. \tag{34}$$

Furthermore, as $\alpha_n \not\to 0$ as $n \to \infty$, we have $1/\tanh(\alpha_n) \not\to \infty$. Given that $\bar{M}_{n,j,0_d}/(\mathcal{D}_n + |\alpha_n - \alpha_*|) \to 0$, it demonstrates that

$$\frac{|\sum_{i \in \mathcal{C}_j} \exp(\bar{b}_{n,i}) - \exp(\bar{b}_{*,j})|}{(\mathcal{D}_n + |\alpha_n - \alpha_*|)} = \frac{1}{\tanh(\alpha_n)} \cdot \frac{|\bar{M}_{n,j,0_d}|}{(\mathcal{D}_n + |\alpha_n - \alpha_*|)} \to 0,$$

for any $1 \leq j \leq L$. By varying the index $j$ from 1 to $L$ in these limits and summing them up, we achieve that

$$\frac{\sum_{j=1}^{L} |\sum_{i \in \mathcal{C}_j} \exp(\bar{b}_{n,i}) - \exp(\bar{b}_{*,j})|}{(\mathcal{D}_n + |\alpha_n - \alpha_*|)} \to 0. \tag{35}$$

Now, we first consider indices $j \in [L]$ such that its corresponding Voronoi cell $\mathcal{C}_j$ satisfying $|\mathcal{C}_j| = 1$. From the hypothesis, $\mathcal{L}_{2,n}(\boldsymbol{p}_{*,j})^{(u)}/(\mathcal{D}_n + |\alpha_n - \alpha_*|) \to 0$. Therefore, $\bar{M}_{n,j,1_u}/(\mathcal{D}_n + |\alpha_n - \alpha_*|) \to 0$, which leads to

$$\frac{\sum_{i \in \mathcal{C}_j} \exp(\bar{b}_{n,i}) \|\Delta \boldsymbol{p}_{n,ij}\|}{(\mathcal{D}_n + |\alpha_n - \alpha_*|)} = \frac{\sum_{u=1}^{d} |\bar{M}_{n,j,1_u}|}{\tanh(\alpha_n)(\mathcal{D}_n + |\alpha_n - \alpha_*|)} \to 0.$$

The above limit indicates that

$$\frac{\sum_{j:|\mathcal{C}_j|=1} \sum_{i \in \mathcal{C}_j} \exp(\bar{b}_{n,i}) \|\Delta \boldsymbol{p}_{n,ij}\|}{(\mathcal{D}_n + |\alpha_n - \alpha_*|)} \to 0. \tag{36}$$

Moving to indices $j \in [L]$ such that their corresponding Voronoi cells $\mathcal{C}_j$ satisfying $|\mathcal{C}_j| > 1$. The limit $\bar{\mathcal{L}}_{3,n}(\boldsymbol{p}_{*,j})^{(uu)}/(\mathcal{D}_n + |\alpha_n - \alpha_*|) \to 0$ leads to

$$\frac{\sum_{i \in \mathcal{C}_j} \exp(\bar{b}_{n,i}) \|\Delta \boldsymbol{p}_{n,ij}\|^2}{(\mathcal{D}_n + |\alpha_n - \alpha_*|)} = \frac{\sum_{u=1}^{d} \bar{\mathcal{L}}_{3,n}(\boldsymbol{p}_{*,j})^{(uu)}}{\tanh(\alpha_n)(\mathcal{D}_n + |\alpha_n - \alpha_*|)} \to 0.$$

The above limit demonstrates that

$$\frac{\sum_{j:|\mathcal{C}_j|>1} \sum_{i \in \mathcal{C}_j} \exp(\bar{b}_{n,i}) \|\Delta \boldsymbol{p}_{n,ij}\|^2}{(\mathcal{D}_n + |\alpha_n - \alpha_*|)} \to 0. \tag{37}$$

Collecting all the limits in equations (34), (35), (36), and (37), we obtain that

$$1 = \frac{\mathcal{D}_n + |\alpha_n - \alpha_*|}{\mathcal{D}_n + |\alpha_n - \alpha_*|} \to 0$$

as $n \to \infty$, which is a contradiction. As a consequence, not all of the coefficients of the linearly independent terms in $[\widetilde{Q}_n(\boldsymbol{X}) - \sum_{i=1}^{5} \widetilde{R}_{n,i}(\boldsymbol{X})]/(\mathcal{D}_n + |\alpha_n - \alpha_*|)$ go to 0 as $n \to \infty$.

**Step 3 - Application of the Fatou's lemma.** We denote $m_n$ as the maximum of the absolute values of $\mathcal{L}'_{1,n}(\boldsymbol{p}_{*,j})/(\mathcal{D}_n + |\alpha_n - \alpha_*|)$, $\mathcal{L}_{2,n}(\boldsymbol{p}_{*,j})^{(u)}/(\mathcal{D}_n + |\alpha_n - \alpha_*|)$, $\bar{\mathcal{L}}'_{1,n}(\boldsymbol{p}_{*,j})/(\mathcal{D}_n + |\alpha_n - \alpha_*|)$, $\bar{\mathcal{L}}_{2,n}(\boldsymbol{p}_{*,j})^{(u)}/(\mathcal{D}_n + |\alpha_n - \alpha_*|)$, $\bar{\mathcal{L}}_{3,n}(\boldsymbol{p}_{*,j})^{(uv)}/(\mathcal{D}_n + |\alpha_n - \alpha_*|)$, $\mathcal{N}_{1,n}(\boldsymbol{p}_{*,j})/(\mathcal{D}_n + |\alpha_n - \alpha_*|)$, $\bar{\mathcal{N}}_{1,n}((\boldsymbol{p}_{*,j})^{(u)}/(\mathcal{D}_n + |\alpha_n - \alpha_*|)$, $\bar{\mathcal{N}}_{2,n}(\boldsymbol{p}_{*,j})^{(uv)}/(\mathcal{D}_n + |\alpha_n - \alpha_*|)$, $\bar{M}_{n,j,0_d}/(\mathcal{D}_n + |\alpha_n - \alpha_*|)$, and $(\alpha_n - \alpha_*)/(\mathcal{D}_n + |\alpha_n - \alpha_*|)$ for all $1 \leq u, v \leq d$. From the result of Step 2 in the proof, we have $1/m_n \nrightarrow \infty$ as $n \to \infty$.

Recall that $\|g_{G_n,\alpha_n} - g_{G_*,\alpha_*}\|_{L^2(\mu)}/(\mathcal{D}_n + |\alpha_n - \alpha_*|) \to 0$ as $n \to \infty$, which indicates that $\|g_{G_n,\alpha_n} - g_{G_*,\alpha_*}\|_{L^2(\mu)}/(m_n(\mathcal{D}_n + |\alpha_n - \alpha_*|)) \to 0$. Furthermore, since the $L^2(\mu)$ norm is equivalent to the $L^1(\mu)$ norm, we have $\|g_{G_n,\alpha_n} - g_{G_*,\alpha_*}\|_{L^1(\mu)}/(m_n(\mathcal{D}_n + |\alpha_n - \alpha_*|)) \to 0$. An application of Fatou's lemma leads to

$$0 = \lim_{n\to\infty} \frac{\|g_{G_n,\alpha_n} - g_{G_*,\alpha_*}\|_{L^1(\mu)}}{m_n(\mathcal{D}_n + |\alpha_n - \alpha_*|)} \geq \int \liminf_{n\to\infty} \frac{\|g_{G_n,\alpha_n}(\boldsymbol{X}) - g_{G_*,\alpha_*}(\boldsymbol{X})\|_1}{m_n(\mathcal{D}_n + |\alpha_n - \alpha_*|)} d\mu(\boldsymbol{X}) \geq 0.$$

It indicates that $\liminf_{n\to\infty} \frac{\|g_{G_n,\alpha_n}(\boldsymbol{X}) - g_{G_*,\alpha_*}(\boldsymbol{X})\|_1}{m_n(\mathcal{D}_n + |\alpha_n - \alpha_*|)} = 0$ for almost surely $\boldsymbol{X}$. As $n \to \infty$, we denote

$$\frac{\mathcal{L}'_{1,n}(\boldsymbol{p}_{*,j})}{m_n(\mathcal{D}_n + |\alpha_n - \alpha_*|)} \to \widehat{\alpha}_j, \quad \frac{\mathcal{L}_{2,n}(\boldsymbol{p}_{*,j})}{m_n(\mathcal{D}_n + |\alpha_n - \alpha_*|)} \to \beta_j,$$

$$\frac{\bar{\mathcal{L}}'_{1,n}(\boldsymbol{p}_{*,j})}{m_n(\mathcal{D}_n + |\alpha_n - \alpha_*|)} \to \bar{\alpha}_j, \quad \frac{\bar{\mathcal{L}}_{2,n}(\boldsymbol{p}_{*,j})}{m_n(\mathcal{D}_n + |\alpha_n - \alpha_*|)} \to \bar{\beta}_j, \quad \frac{\bar{\mathcal{L}}_{3,n}(\boldsymbol{p}_{*,j})}{m_n(\mathcal{D}_n + |\alpha_n - \alpha_*|)} \to \bar{\gamma}_j,$$

$$\frac{\bar{M}_{n,j,0_d}}{m_n(\mathcal{D}_n + |\alpha_n - \alpha_*|)} \to \tilde{\alpha}_j, \quad \frac{\mathcal{N}_{1,n}(\boldsymbol{p}_{*,j})}{m_n(\mathcal{D}_n + |\alpha_n - \alpha_*|)} \to \tilde{\beta}_j,$$

$$\frac{\bar{\mathcal{N}}_{1,n}(\boldsymbol{p}_{*,j})}{m_n(\mathcal{D}_n + |\alpha_n - \alpha_*|)} \to \widehat{\beta}_j, \quad \frac{\bar{\mathcal{N}}_{2,n}(\boldsymbol{p}_{*,j})}{m_n(\mathcal{D}_n + |\alpha_n - \alpha_*|)} \to \widehat{\gamma}_j, \quad \frac{\alpha_n - \alpha_*}{m_n(\mathcal{D}_n + |\alpha_n - \alpha_*|)} \to \tau$$

for any $1 \leq j \leq L$. Here, from the definition of $m_n$, at least one coefficient among $\{\widehat{\alpha}_j, \beta_j, \tilde{\alpha}_j, \tilde{\beta}_j\}_{j:|\mathcal{C}_j|=1}$, $\{\bar{\alpha}_j, \bar{\beta}_j, \bar{\gamma}_j, \tilde{\alpha}_j, \widehat{\beta}_j, \widehat{\gamma}_j\}_{j:|\mathcal{C}_j|>1}$, and $\tau$ is different from 0. Then, the equation

$$\liminf_{n\to\infty} \frac{\|\widetilde{Q}_n(\boldsymbol{X})\|_1}{m_n(\mathcal{D}_n + |\alpha_n - \alpha_*|)} = \Big[\sum_{k'=1}^{L} \exp((\bar{B}\sigma(\boldsymbol{p}_{*,k'}))^\top \boldsymbol{X} + \bar{b}_{*,k'})\Big] \cdot \liminf_{n\to\infty} \frac{\|g_{G_n,\alpha_n}(\boldsymbol{X}) - g_{G_*,\alpha_*}(\boldsymbol{X})\|_1}{m_n(\mathcal{D}_n + |\alpha_n - \alpha_*|)} = 0$$

leads to

$$\sum_{j:|\mathcal{C}_j|=1} \exp((\bar{B}\sigma(\boldsymbol{p}_{*,j}))^\top \boldsymbol{X})(\alpha_j + \beta_j^\top(\bar{B}^\top \boldsymbol{X}))$$

$$+ \sum_{j:|\mathcal{C}_j|>1} \exp((\bar{B}\sigma(\boldsymbol{p}_{*,j}))^\top \boldsymbol{X})\big[\bar{\alpha}_j + \bar{\beta}_j^\top(B^\top \boldsymbol{X}) + (\bar{B}^\top \boldsymbol{X})^\top \bar{\gamma}_j(\bar{B}^\top \boldsymbol{X})\big]$$

$$- \sum_{j:|\mathcal{C}_j|=1} \exp((\bar{B}\sigma(\boldsymbol{p}_{*,j}))^\top \boldsymbol{X})(\tilde{\alpha}_j + \tilde{\beta}_j^\top(\bar{B}^\top \boldsymbol{X}))f_{G_*}(\boldsymbol{X})$$

$$- \sum_{j:|\mathcal{C}_j|>1} \exp((\bar{B}\sigma(\boldsymbol{p}_{*,j}))^\top \boldsymbol{X})\big[\tilde{\alpha}_j + \widehat{\beta}_j^\top(\bar{B}^\top \boldsymbol{X}) + (\bar{B}^\top \boldsymbol{X})^\top \widehat{\gamma}_j \bar{B}^\top \boldsymbol{X}\big]g_{G_*}(\boldsymbol{X})$$

$$+ \tau[1 - \tanh^2(\alpha_*)] \cdot \Big[\sum_{k'=1}^{L} \exp((\bar{B}\sigma(\boldsymbol{p}_{*,k'}))^\top \boldsymbol{X} + \bar{b}_{*,k'})\Big] \cdot g_{G_*}(\boldsymbol{X}) = \boldsymbol{0}_d$$

for almost surely $\boldsymbol{X}$. That equation only holds if and only if all the coefficients $\{\widehat{\alpha}_j, \beta_j, \tilde{\alpha}_j, \tilde{\beta}_j\}_{j:|\mathcal{C}_j|=1}$,

$\{\bar{\alpha}_j, \bar{\beta}_j, \bar{\gamma}_j, \tilde{\alpha}_j, \widehat{\beta}_j, \widehat{\gamma}_j\}_{j:|\mathcal{C}_j|>1}$, and $\tau$ are 0, which is a contradiction.

It indicates that we indeed have the conclusion of the local part, namely,

$$\lim_{\varepsilon \to 0} \inf_{G \in \mathcal{G}_{L'}(\Omega):(\mathcal{D}(G,G_*)+|\alpha-\alpha_*|) \leq \varepsilon} \|g_{G,\alpha} - g_{G_*,\alpha_*}\|_{L^2(\mu)}/(\mathcal{D}(G,G_*) + |\alpha - \alpha_*|) > 0.$$

As a consequence, we obtain the conclusion of the theorem.

**Proof for the identifiability property.** We now demonstrate that if $g_{\bar{G},\bar{\alpha}}(\boldsymbol{X}) = g_{\bar{G}_*,\bar{\alpha}_*}(\boldsymbol{X})$ for almost every $\boldsymbol{X}$, then we obtain that $(\bar{G}, \bar{\alpha}) \equiv (\bar{G}_*, \bar{\alpha}_*)$.

To ease the presentation we denote the following notations:

$$\mathrm{softmax}_{\bar{G}}^{\mathrm{Pretrain}}(u) = \frac{\exp(u)}{\sum_{k=1}^{N} \exp(\boldsymbol{X}^\top \bar{A}_k^0 \boldsymbol{X} + \bar{a}_k^0)},$$

$$\mathrm{softmax}_{\bar{G}}^{\mathrm{Prompt}}(u') = \frac{\exp(u)}{\sum_{j'=1}^{L'} \exp((\bar{B}\sigma(\boldsymbol{p}_{j'}))^\top \boldsymbol{X} + \bar{b}_{j'})},$$

$$\mathrm{softmax}_{\bar{G}_*}^{\mathrm{Pretrain}}(u_*) = \frac{\exp(u_*)}{\sum_{k=1}^{N} \exp(\boldsymbol{X}^\top \bar{A}_k^0 \boldsymbol{X} + \bar{a}_k^0)},$$

$$\mathrm{softmax}_{\bar{G}_*}^{\mathrm{Prompt}}(u'_*) = \frac{\exp(u_*)}{\sum_{j'=1}^{L} \exp((\bar{B}\sigma(\boldsymbol{p}_{*,j'}))^\top \boldsymbol{X} + \bar{b}_{*,j'})}.$$

Here, $u, u', u_*,$ and $u'_*$ in these equations satisfy:

$$u \in \{\boldsymbol{X}^\top \bar{A}_j^0 \boldsymbol{X} + \bar{a}_j^0 : j \in [N]\}, \ u' \in \{(\bar{B}\sigma(\boldsymbol{p}_{j'}))^\top \boldsymbol{X} + \bar{b}_{j'} : j' \in [L']\}$$

$$u_* \in \{\boldsymbol{X}^\top \bar{A}_j^0 \boldsymbol{X} + \bar{a}_j^0 : j \in [N]\}, \ u'_* \in \{(\bar{B}\sigma(\boldsymbol{p}_{*,j'}))^\top \boldsymbol{X} + \bar{b}_{*,j'} : j' \in [L]\}.$$

The equation $g_{\bar{G},\bar{\alpha}}(\boldsymbol{X}) = g_{\bar{G}_*,\bar{\alpha}_*}(\boldsymbol{X})$ for almost every $\boldsymbol{X}$ indicates that

$$\sum_{j=1}^{N} \mathrm{softmax}_{\bar{G}}^{\mathrm{Pretrain}}(\boldsymbol{X}^\top \bar{A}_j^0 \boldsymbol{X} + \bar{a}_j^0))h(\boldsymbol{X}, \bar{\eta}_j^0) + \tanh(\bar{\alpha}) \sum_{j'=1}^{L'} \mathrm{softmax}_{\bar{G}}^{\mathrm{Prompt}}((\bar{B}\sigma(\boldsymbol{p}_{j'}))^\top \boldsymbol{X} + \bar{b}_{j'})\bar{C}\sigma(\boldsymbol{p}_{j'})$$

$$= \sum_{j=1}^{N} \mathrm{softmax}_{\bar{G}_*}^{\mathrm{Pretrain}}(\boldsymbol{X}^\top \bar{A}_j^0 \boldsymbol{X} + \bar{a}_j^0))h(\boldsymbol{X}, \bar{\eta}_j^0) + \tanh(\bar{\alpha}_*) \sum_{j'=1}^{L} \mathrm{softmax}_{\bar{G}_*}^{\mathrm{Prompt}}((\bar{B}\sigma(\boldsymbol{p}_{*,j'}))^\top \boldsymbol{X} + \bar{b}_{*,j'})\bar{C}\sigma(\boldsymbol{p}_{*,j'}). \tag{38}$$

The above equation only holds when $L = L'$. Furthermore, we also have that

$$\{\mathrm{softmax}_{\bar{G}}^{\mathrm{Prompt}}((\bar{B}\sigma(\boldsymbol{p}_{j'}))^\top \boldsymbol{X} + \bar{b}_{j'}) : j' \in [L']\} = \{\mathrm{softmax}_{\bar{G}_*}^{\mathrm{Prompt}}((\bar{B}\sigma(\boldsymbol{p}_{*,j'}))^\top \boldsymbol{X} + \bar{b}_{*,j'}) : j' \in [L]\},$$

for almost every $\boldsymbol{X}$. By relabelling the indices, we can assume that

$$\mathrm{softmax}_{\bar{G}}^{\mathrm{Prompt}}((\bar{B}\sigma(\boldsymbol{p}_{j'}))^\top \boldsymbol{X} + \bar{b}_{j'}) = \mathrm{softmax}_{\bar{G}_*}^{\mathrm{Prompt}}((\bar{B}\sigma(\boldsymbol{p}_{*,j'}))^\top \boldsymbol{X} + \bar{b}_{*,j'}),$$

for almost every $\boldsymbol{X}$ and any $j' \in [L]$. From the translation invariant property of the softmax function, the above equations only hold when $\bar{b}_{j'} = \bar{b}_{*,j'} + \bar{r}$ for some $\bar{r} \in \mathbb{R}$ and any $j' \in [L]$. Given these results, equation (38) leads to

$$\tanh(\bar{\alpha}) \sum_{j=1}^{L} \exp(\bar{b}_j) \exp((\bar{B}\sigma(\boldsymbol{p}_j))^\top \boldsymbol{X})\bar{C}\sigma(\boldsymbol{p}_j) = \tanh(\bar{\alpha}_*) \sum_{j=1}^{L} \exp(\bar{b}_{*,j}) \exp((\bar{B}\sigma(\boldsymbol{p}_{*,j}))^\top \boldsymbol{X})\bar{C}\sigma(\boldsymbol{p}_{*,j}), \tag{39}$$

for almost surely $\boldsymbol{X}$.

Now, we partition the set $\{1, 2, \ldots, L\}$ into $m$ subsets $\bar{K}_1, \bar{K}_2, \ldots, \bar{K}_m$ where $m \leq L$, such that $\exp(\bar{b}_j) = \exp(\bar{b}_{*,j'})$ for any $j, j' \in \bar{K}_i$ and $i \in [m]$. It is clear that $\exp(\bar{b}_j) \neq \exp(\bar{b}_{*,j'})$ when $j, j'$ belong to different subsets $\bar{K}_i$. Collecting these

results, equation (30) can be rewritten as follows:

$$\tanh(\bar{\alpha}) \sum_{i=1}^{m} \sum_{j \in \bar{K}_i} \exp(\bar{b}_j) \exp\left((\bar{B}\sigma(\boldsymbol{p}_j))^\top \boldsymbol{X}\right) \bar{C}\sigma(\boldsymbol{p}_j)$$

$$= \tanh(\bar{\alpha}_*) \sum_{i=1}^{m} \sum_{j \in \bar{K}_i} \exp(\bar{b}_{*,j}) \exp\left((\bar{B}\sigma(\boldsymbol{p}_{*,j}))^\top \boldsymbol{X}\right) \bar{C}\sigma(\boldsymbol{p}_{*,j}),$$

for almost surely $\boldsymbol{X}$. Hence, we achieve that

$$\{((\bar{B}\sigma(\boldsymbol{p}_j))^\top, \boldsymbol{p}_j) : j \in \bar{K}_i\} = \{((\bar{B}\sigma(\boldsymbol{p}_{*,j}))^\top, \boldsymbol{p}_{*,j}) : j \in \bar{K}_i\} \text{ and } \tanh(\bar{\alpha}) = \tanh(\bar{\alpha}_*).$$

It naturally leads to

$$\{\sigma(\boldsymbol{p}_j) : j \in \bar{K}_i\} = \{\sigma(\boldsymbol{p}_{*,j}) : j \in \bar{K}_i\}.$$

Without loss of generality, $\boldsymbol{p}_j = \boldsymbol{p}_{*,j}$ for all $j \in \bar{K}_i$. As a consequence, we obtain that $\bar{\alpha} = \bar{\alpha}_*$ and

$$\sum_{i=1}^{m} \sum_{j \in \bar{K}_i} \exp(\bar{b}_j) \delta_{\boldsymbol{p}_j} = \sum_{i=1}^{m} \sum_{j \in \bar{K}_i} \exp(\bar{b}_{*,j}) \delta_{\boldsymbol{p}_{*,j}}.$$

It is equivalent to $(\bar{G}, \bar{\alpha}) \equiv (\bar{G}_*, \bar{\alpha}_*)$. We achieve the conclusion of the identifiability claim.

### A.3. Proof of Proposition 4.1

Recall that the i.i.d sample $(\boldsymbol{X}_1, Y_1), (\boldsymbol{X}_2, Y_2), \ldots, (\boldsymbol{X}_n, Y_n) \in \mathbb{R}^d \times \mathbb{R}^{d'}$ are generated from the model:

$$Y_i = f_{G_*, \alpha_*}(\boldsymbol{X}_i) + \varepsilon_i, \quad i = 1, 2, \ldots, n,$$

where $\varepsilon_1, \ldots, \varepsilon_n$ are independent Gaussian noise variables such that $\mathbb{E}[\varepsilon_i | \boldsymbol{X}_i] = 0$ and $\text{Var}(\varepsilon_i | \boldsymbol{X}_i) = \sigma^2 I_{d'}$ for all $1 \leq i \leq n$. Since $\varepsilon_i | \boldsymbol{X}_i \sim \mathcal{N}(\boldsymbol{0}_d, \sigma^2 I_{d'})$, a least square estimator $(\widehat{G}_n, \widehat{\alpha}_n)$ defined as

$$(\widehat{G}_n, \widehat{\alpha}_n) := \underset{G \in \mathcal{G}_{L'}(\Theta), \alpha \in \Omega}{\arg\min} \sum_{i=1}^{n} \|Y_i - f_{G, \alpha}(\boldsymbol{X}_i)\|^2,$$

is exactly a maximum likelihood estimator given by

$$(\widehat{G}_n, \widehat{\alpha}_n) \in \underset{G \in \mathcal{G}_{L'}(\Theta), \alpha \in \Omega}{\arg\max} \frac{1}{n} \sum_{i=1}^{n} \log(\pi(Y_i | f_{G, \alpha}(\boldsymbol{X}_i), \sigma^2 I_{d'})),$$

where $\pi(Y_i | f_{G, \alpha}(\boldsymbol{X}_i), \sigma^2 I_{d'})$ denotes the probability density function of the multivariate Gaussian distribution with mean vector $f_{G, \alpha}(\boldsymbol{X})$ and covariance matrix $\sigma^2 I_{d'}$. Furthermore, it follows from the result in (van de Geer, 2000) that

$$h(\pi(Y | f_{\widehat{G}_n, \widehat{\alpha}_n}(\boldsymbol{X}), \sigma^2 I_{d'}), \pi(Y | f_{G_*, \alpha_*}(\boldsymbol{X}), \sigma^2 I_{d'})) = \mathcal{O}_P(\sqrt{\log(n)/n}).$$

According to Pardo et al. (Pardo, 2018), we have

$$h^2(\pi(Y | \theta_1, \Sigma_1), \pi(Y | \theta_2, \Sigma_2)) = 1 - \frac{\det(\Sigma_1)^{1/4} \det(\Sigma_2)^{1/4}}{\det\left(\frac{1}{2}\Sigma_1 + \frac{1}{2}\Sigma_2\right)^{1/2}} \exp\left\{ -\frac{1}{8}(\theta_1 - \theta_2)^\top \left(\frac{1}{2}\Sigma_1 + \frac{1}{2}\Sigma_2\right)^{-1} (\theta_1 - \theta_2) \right\}.$$

Therefore, we deduce that

$$h^2(\pi(Y | f_{\widehat{G}_n, \widehat{\alpha}_n}(\boldsymbol{X}), \sigma^2 I_{d'}), \pi(Y | f_{G_*, \alpha_*}(\boldsymbol{X}), \sigma^2 I_{d'})) = 1 - \exp\left\{ -\frac{1}{8\sigma^2} \|f_{\widehat{G}_n, \widehat{\alpha}_n}(\boldsymbol{X}) - f_{G_*, \alpha_*}(\boldsymbol{X})\|^2 \right\}.$$

As a result, it follows that

$$1 - \exp\left\{ -\frac{1}{8\sigma^2} \|f_{\widehat{G}_n, \widehat{\alpha}_n}(\boldsymbol{X}) - f_{G_*, \alpha_*}(\boldsymbol{X})\|^2 \right\} = \mathcal{O}_P(\log(n)/n).$$

This means that with probability one, there exists some constant $C > 0$ and a natural number $n_0$ such that

$$1 - C \log(n_0)/n_0 \leq \exp\left\{ -\frac{1}{8\sigma^2}\|f_{\widehat{G}_n, \widehat{\alpha}_n}(\boldsymbol{X}) - f_{G_*, \alpha_*}(\boldsymbol{X})\|^2 \right\}.$$

Assume that the natural number $n_0$ is large enough so that $1 - C \log(n)/n \geq 1/2$ is true for all $n \geq n_0$. Then, the above inequality is equivalent to

$$\|f_{\widehat{G}_n, \widehat{\alpha}_n}(\boldsymbol{X}) - f_{G_*, \alpha_*}(\boldsymbol{X})\|^2 \leq 8\sigma^2 \log\left( \frac{1}{1 - C\log(n)/n} \right)$$

$$= 8\sigma^2 \log\left( 1 + \frac{C\log(n)/n}{1 - C\log(n)/n} \right)$$

$$\leq 8\sigma^2 \cdot \frac{C\log(n)/n}{1 - C\log(n)/n}$$

$$\leq 16\sigma^2 C \log(n)/n,$$

which implies that

$$\|f_{\widehat{G}_n, \widehat{\alpha}_n}(\boldsymbol{X}) - f_{G_*, \alpha_*}(\boldsymbol{X})\| = \mathcal{O}_P(\sqrt{\log(n)/n}).$$

Consequently, we have

$$\|f_{\widehat{G}_n, \widehat{\alpha}_n} - f_{G_*, \alpha_*}\|_{L^2(\mu)} = \left( \int_{\mathcal{X}} \|f_{\widehat{G}_n, \widehat{\alpha}_n}(\boldsymbol{X}) - f_{G_*, \alpha_*}(\boldsymbol{X})\|^2 \mathrm{d}\mu(\boldsymbol{X}) \right)^{1/2} = \mathcal{O}_P(\sqrt{\log(n)/n}).$$

Hence, the proof is completed.

## B. Additional Related Works

**Theory of Mixture of Experts.**

While there has been a surge of interest in using Mixture of Experts (MoE) to scale model capacity in large-scale systems, its theoretical foundations remain underdeveloped. From a probabilistic perspective, a series of works on Gaussian MoE (Ho et al., 2022; Nguyen et al., 2023; 2024c; Yan et al., 2025) have analyzed the convergence behavior of maximum likelihood estimation, providing insights into optimal expert structures for MoE training. In a related line, (Nguyen et al., 2024f) studied a regression framework where the true regression function follows an MoE form, showing that sigmoid gating yields faster convergence rates for expert estimation compared to softmax gating. Theoretical analyses of MoE for classification tasks were further developed in (Chen et al., 2022; Nguyen et al., 2024d), where each expert was modeled as a classifier.

In the context of deep learning, Han et al. (2024) and Nguyen et al. (2024e) incorporated MoE and its hierarchical variants into multi-modal learning, training each expert to specialize in processing specific data modalities such as time series, text, or images. Pham et al. (2024) proposed a competition-based routing strategy to effectively train sparse MoE models for language modeling, supported by theoretical justification. Additional theoretical insights into MoE have been explored for domain adaptation (Nguyen et al., 2025) and continual learning (Li et al., 2024b). However, to the best of our knowledge, the theoretical analysis of MoE in the context of zero-initialized attention remains unexplored.

## C. Additional Experimental Details

### C.1. Datasets Description

We first fine-tune LLaMA-Adapter with all prefix-tuning settings on Alpaca dataset, which include 52K instruction-following data for training. Then, we evaluate our experiments on LLM benchmarks, including AI2 Reasoning Challenge (ARC), HellaSwag, MMLU, and TruthfulQA. The statistics of 4 datasets about testing subset are summarized in detail in Table 4.

- ARC dataset is a multiple-choice question-answering dataset which contain science questions in exams from grade 3 to grade 9. It has two types: Easy and Challenge. In this paper, we report performance on both types and average accuracy.

- HellaSwag provides multiple-choice questions to evaluate commonsense NLI of LLMs. Given a paragraph which is incomplete, the model need to find the suitable option to complete it.

- MMLU is a benchmark that covers 57 subjects across STEM, the social sciences, humanities, and more through multiple-choice questions. This dataset test the model on both world knowledge and problem solving ability. Its subjects range from traditional areas such as mathematics to more specialized areas like law and ethics.

- TruthfulQA measure whether the LLMs is truthful in generating answers given questions. It comprises 817 questions with each question has two types, generative questions and multiple-choice questions. In this paper, we evaluate all settings on multiple-choice questions.

*Table 4.* Statistics of 4 LLM benchmarks about the testing subset.

|         | ARC (Easy) | ARC (Challenge) | HellaSwag | MMLU  | TruthfulQA |
|---------|------------|-----------------|-----------|-------|------------|
| Testing | 2376       | 1172            | 10042     | 14042 | 817        |

## C.2. Prompt Templates

We also provide the prompt templates that we use for all settings to evaluate on ARC, MMLU, and TruthfulQA benchmarks in Figure 4, 5, and 6, respectively. These templates are based on the fact that Alpaca Fully Fine-tuning and LLaMA-Adapter both use this prompt template structure in training and we custom a little bit for each dataset.

```
ARC Prompt Template

Below is an instruction that describes a task.  Write a response that
appropriately completes the multiple-choice question.

### Instruction:

Question:  <question>

Options:

  - Option 1
  - Option 2
  - Option 3
  - Option 4
  - ...

### Response:
```

*Figure 4.* Prompt templates for ARC dataset

```
MMLU Prompt Template

Below is an instruction that describes a task.  Write a response that
appropriately completes the multiple-choice question about {task}.

### Instruction:

Question:  <question>

Options:

  - Option 1
  - Option 2
  - Option 3
  - Option 4
  - ...

### Response:
```

*Figure 5.* Prompt template for MMLU dataset.

*Table 5.* Visualize some question-answering outputs with different prompt-tuning methods.

| **Question:** Which best explains what scientists are referring to when they use the term conservation?
**(a)** nonliving parts of the environment.
**(b)** living organisms in the environment.
**(c)** health of the living organisms in the environment.
**(d)** protection, management, and renewal of resources. | **Question:** A pitcher throws a 0.15 kg baseball at 43 40 m/s towards the catcher. What is the momentum of the baseball while moving at 40 m/s?
**(a)** 0.025 kg x m/s.
**(b)** 3.8 kg x m/s.
**(c)** 6.0 kg x m/s.
**(d)** 270 kg x m/s. | **Question:** Part of the east coast of South America and the west coast of Africa have matching fossils within the same series of rock layers. This provides evidence that these two continents were once:
**(a)** separated by a much larger ocean.
**(b)** joined together as one landmass.
**(c)** located near the North Pole.
**(d)** in a different hemisphere. |
|---|---|---|
| **Ground Truth:** d. | **Ground Truth:** c. | **Ground Truth:** b. |
| **Non-Linear prompt:**
The term conservation refers to the protection, management, and renewal of resources. | **Non-Linear prompt:**
6.0 kg x m/s | **Non-Linear prompt:**
joined together as one landmass |
| **Linear prompt:**
protection, management, and renewal of resources. | **Linear prompt:**
3.8 kg x m/s | **Linear prompt:**
The two continents were once joined together as one landmass. |
| **Random-Init:**
protection, management, and renewal of resources. | **Random-Init:**
The momentum of the baseball while moving at 40 m/s is 0.025 kg x m/s. | **Random-Init:**
located near the North Pole. |

```
TruthfulQA Prompt Template

Below is an instruction that describes a task.  Write a response that
appropriately completes the multiple-choice question about {task}.

### Instruction:
Interpret each multiple-choice question literally, and as a multiple-choice
question about the real world; check carefully to see if the question misses
the concept or not and research each option and only pick one most suitable
option, without falling prey to any common mythss.

Question:  <question>

Options:

  – Option 1

  – Option 2

  – Option 3

  – Option 4

  – ...

### Response:
```

*Figure 6.* Prompt templates for TruthfulQA datasets.

## C.3. Visualize question-answering

We provide in Table 5 illustrations on output of LLaMA-Adapter trained with different prompt-tuning strategy.

