# OpenReview forum: "On Zero-Initialized Attention: Optimal Prompt and Gating Factor Estimation"
_ICML.cc/2025/Conference — ICML 2025 poster_

### Official Review · Reviewer_jpno · 2025-03-14

**Overall Recommendation:** 3

**Summary:**

This paper provides extensive theoretical analysis for the zero-initialized attention in LLaMA-Adapters and connects it with mixture-of-experts models. Based on this, the author introduces non-linear prompts. Plus, experiments are conducted to demonstrate the performance of non-linear prompts.

**Claims And Evidence:**

The claims of the article are consistent with the content he provided. The structure in attention can indeed be regarded as a mixture-of-experts structure (at least similar in form), and the experiments have confirmed the superiority of the method proposed in the paper.

**Essential References Not Discussed:**

None.

**Experimental Designs Or Analyses:**

Regarding the experimental part, the experiments on most data sets have verified that zero-init has better results than random-init, and non-linear has greater potential than linear, which to a certain extent can explain the superiority of the method proposed in the paper. However, in the experiments of HellaSwag and LLaMA-7B in TruthfullQA, it seems that non-linear prompts are worse than linear prompts. Can the author explain this part of the problem?

**Methods And Evaluation Criteria:**

Yes. This article focuses on treating the zero-initialized attention in LLaMA-Adapters as a mixture-of-experts model and optimizing LLaMA-Adapters using nonlinear prompts. In the article, the author provides the connection between mixture-of-experts and attention and gives proof.

**Other Comments Or Suggestions:**

I feel that the author could compare several initialization methods, such as all 1, or other forms of random initialization, and discuss the sparsity of initialization. Intuitively, I feel that sparsity may be a more important point. For example, most of the values are 0 but a small number of values exist.

**Other Strengths And Weaknesses:**

None.

**Questions For Authors:**

None.

**Relation To Broader Scientific Literature:**

The more important finding of the article may be that the introduction of non-linear prompts to LLaMA-Adapters provides better performance, which can inspire new prompt tuning.

**Theoretical Claims:**

I have partially checked the relevant derivations and there should be no problem. However, if there are any more questions, I will add them later.

---

> ### Author Rebuttal · Authors · 2025-03-29
>
> Thank you for your constructive feedback and insightful comments. We hope that we can address your concerns with the responses below.
>
> ### **Q1: Comparison between Non-Linear and Linear prompt on HellaSwag and TruthfullQA with LLaMA-7B setting:**
> Thank you for your comments. Our main study in this paper is to **focus on providing detailed theoretical analysis and experiments to understand the benefits of the zero-initialized attention over the random-initialized attention (conventional prefix-tuning approach)** based on their connection to MoE models. Additionally, our analysis also indicates that **non-linear prompts can be also optimally estimated as linear prompt with greater flexibility**, suggesting that non-linear prompts are potential alternative to linear prompts.
>
> To justify the potentials of non-linear prompts for LLaMA-Adapter models, we perform several experiments to compare the performance when using the linear prompts versus when using non-linear prompts. From these experimental results, we observe that non-linear prompts achieve **higher performance in most settings (ranging from 0.5% to 4%)**, specifically across **all results on LLaMA-13B and the first two datasets on LLaMA-7B**. However, there are certain cases where non-linear prompts yield only **comparable results**, with performance differences **ranging from 0.1% to 0.5% lower than the linear prompt settings**.
>
> This variation is expected because no single prompting method universally excels across all tasks and models. The effectiveness of non-linear prompts can depend on several factors, such as the dataset characteristics, model capacity, and the complexity of task-specific adaptations. In some cases, the additional expressivity of non-linear prompts may not be necessary, leading to performance that closely approximates linear prompts.
>
> ### **Q2: Comparison among several initialization methods:**
> Thank you for your suggestion. Indeed, we would like to clarify that the Random and Zero Initializations in the manuscript do not refer to the initialization of some vector. Let us take this opportunity to explain each of these initializations.
>
> First of all, **Random Initialization (equivalently, Random-Init) follows the conventional prefix-tuning approach**, where traditional attention is applied to all tokens, including prompt vectors and previous output tokens, without incorporating the zero-initialization mechanism introduced in the original paper. The term "Random-Init" is used because, at the initial stage, all prompt vectors are randomly initialized. When combined with traditional attention mechanisms, this randomness affects the model’s convergence robustness. **The terms "Random-Init" and "Zero-Init" are also used in the original LLaMA-Adapter paper**.
>
> On the contrary, **Zero Initialization (in short, Zero-Init) setting introduces a learnable gating factor within the attention layers that use prompt, which is initially set to zero**. Then, tanh activation function is applied to this factor to regulate the scale of this factor into [−1,1]. Additionally, separate softmax operations are applied to the attention scores of prompt tokens and word tokens independently, after which the gating factor is applied to the attention score of prompt tokens. This mechanism, as detailed in Equation (7) of the original LLaMA-Adapter paper, plays a crucial role in controlling the contribution of prompt tokens to the overall attention mechanism. **By initializing this gating factor at zero**, this factor can first eliminate the influence of under-fitted prompts at the early stages of training, allowing the model to gradually adjust its magnitude to incorporate meaningful instruction semantics into LLaMA.
>
> Finally, in our study, we provide a detailed theoretical analysis and experiments to understand the benefits of zero-initialized attention over the random-initialized attention (conventional prefix-tuning approach). We also provide a theoretical analysis and experiments to show the effectiveness and flexibility of the Non-Linear prompt combined with the zero-initialized mechanism.
>
> We hope our response answers your question about the initialization. Otherwise, please feel free to let us know, we are more than happy to address your further concerns.

---

### Official Review · Reviewer_XuJQ · 2025-03-14

**Overall Recommendation:** 3

**Summary:**

The paper provides a rigorous theoretical foundation for zero-initialized attention, which has been successfully used in fine-tuning large language models (LLMs), particularly in LLaMA-Adapter. Establishes a connection between zero-initialized attention and mixture-of-experts (MoE) models. It additionally proves that both linear and non-linear prompts, along with gating functions, can be optimally estimated.

**Claims And Evidence:**

The claims in the submission are largely supported by clear and convincing evidence, as the paper provides both theoretical justifications and empirical validations for its main arguments. The connection between zero-initialized attention and the mixture-of-experts (MoE) framework is rigorously established through mathematical derivations, and the statistical benefits of optimal prompt and gating factor estimation are backed by well-defined regression-based analyses. The experimental results on open LLM benchmarks align with the theoretical findings, demonstrating improved performance of zero-initialized attention over random initialization and highlighting the advantages of non-linear prompts.

**Essential References Not Discussed:**

the paper overlooks the Switch Transformer model, which employs a simplified MoE approach to achieve efficient scaling

**Experimental Designs Or Analyses:**

The performance gain is notable, with a clear highlight in Table 1.

It is strange that LLaMA-7B + zero-init + linear, LLaMA-7B + zero-init + non-linear is worse than LLaMA-7B, Fully Fine-tuning Alpaca. Usually PeFT is much worse than full fine-tuning. Is the full fine-tuned model well-tuned (and overfitted), and what is the trade-off here?

**Methods And Evaluation Criteria:**

The paper effectively justifies its focus on zero-initialized attention by connecting it to the mixture-of-experts (MoE) framework and demonstrating its theoretical advantages in prompt estimation. The evaluation is conducted on widely recognized open LLM benchmarks, including AI2 Reasoning Challenge (ARC), HellaSwag, MMLU, and TruthfulQA, which are appropriate for assessing the model’s ability to follow instructions, reason, and generate accurate responses. The choice of LLaMA-7B and LLaMA-13B as base models is also reasonable, as they represent strong open-source LLMs used in real-world applications. Furthermore, the paper compares zero-initialized attention against conventional random-initialized attention and other fine-tuning strategies such as LoRA and full fine-tuning, providing a comprehensive evaluation. However, while the experimental setup is robust, additional analysis on different model scales or alternative PEFT techniques could further strengthen the generalizability of the findings.

**Other Comments Or Suggestions:**

​The paper demonstrates strengths in its theoretical analysis and empirical validation of zero-initialized attention within large language models (LLMs). By establishing a connection between zero-initialized attention and mixture-of-experts (MoE) models, it offers a novel perspective that could inform future research in parameter-efficient fine-tuning. The empirical results, showing improved performance with zero-initialized attention and the effectiveness of non-linear prompts, add practical value to the theoretical insights. However, the paper's originality is somewhat tempered by its reliance on and builds upon PeFT approaches with on-linear prompt structures, which is not entirely new.

**Other Strengths And Weaknesses:**

One strength from the theory side is that the paper rigorously establishes a connection between zero-initialized attention and the mixture-of-experts (MoE) framework, providing a solid mathematical foundation for understanding its benefits in prompt tuning. By leveraging regression-based analysis and deriving optimal estimation rates for both linear and non-linear prompts, the paper offers a theoretical justification for why zero-initialized attention improves sample efficiency and stability in parameter-efficient fine-tuning.

**Questions For Authors:**

No

**Relation To Broader Scientific Literature:**

By establishing a theoretical connection between zero-initialized attention and MoE models, the paper builds upon the foundational principles of MoE, where a gating mechanism dynamically selects a subset of experts (or parameters) for each input, thereby improving computational efficiency and model capacity. This approach resonates with recent advancements in sparsely-gated MoE layers, which have been employed in scaling transformer models efficiently by activating only pertinent subsets of parameters during processing. Also, the exploration of both linear and non-linear prompts in the context of zero-initialized attention extends the current understanding of prompt-based learning and its integration with attention mechanisms, offering a nuanced perspective on optimizing prompt and gating factor estimation in LLM fine-tuning.

**Theoretical Claims:**

The proofs for Theorems 4.2 and 5.2, which demonstrate the optimality of prompt and gating factor estimation, follow a logical structure, using techniques such as Voronoi loss functions and parametric convergence analysis.

---

> ### Author Rebuttal · Authors · 2025-03-29
>
> Thanks for your constructive feedback and insightful comments. We would like to address your concerns as follows:
> ### **Q1: Explanation for results in Table 2.**
> Thank you for your comments. **We want to clarify that most PEFT methods (e.g., LLaMA-7B + zero-init + linear, LLaMA-7B + zero-init + non-linear) perform worse than fully fine-tuning of LLaMA-7B, as shown in Table 2 (e.g., ARC dataset)**. While PEFT sometimes matches full fine-tuning, this is likely because the pre-trained model already contains fundamental knowledge for certain downstream tasks. However, in general, PEFT methods lag behind in performance compared to full fine-tuning.
>
> However, it's important to emphasize that fully fine-tuning models like LLaMA is computationally expensive, making it impractical for low-resource settings. PEFT significantly reduces the number of learnable parameters while achieving comparable results, offering a more efficient alternative in such environments.
>
> ### **Q2: Additional analysis on other Parameter-Efficient Fine-Tuning (PEFT) techniques.**
> Thank you for your suggestion. To further validate our method, we conducted additional experiments comparing it with other PEFT methods, including **Prompt Tuning [1]**, **IA3 [2]**, and **VeRA (r=128 and applied to the same modules as LoRA) [3]**, on LLaMA-7B setting. These comparisons provide a broader perspective on how our approach performs relative to established fine-tuning techniques. Additionally, we would like to clarify that the **name Random-Init prompt in our study follows the traditional prefix-tuning approach**. The results is presented in table below.
> |Method|ARC (eas)|ARC (cha)|ARC (aver)|MMLU|Hellaswag|TruthfullQA|Average|
> |-|-|-|-|-|-|-|-|
> |LLaMA-7B, Fully Fine-tuning Alpaca|67.47|46.25|56.86|37.25|77.09|42.35|53.39|
> |LLaMA-7B, LoRA Alpaca|61.91|42.15|52.03|34.87|**77.53**|**46.14**|52.64|
> |LLaMA-7B, Prompt Tuning|55.35|37.46|46.41|32.85|75.88|34.76|47.48|
> |LLaMA-7B, IA3|52.06|35.92|43.99|31.65|75.73|32.8|46.04|
> |LLaMA-7B, VeRA|49.2|35.49|42.35|30.88|75.59|31.95|45.19|
> |LLaMA-7B, Prefix-Tune (Random-Init)|60.65|40.7|50.67|35.12|72.62|37.82|49.06|
> |LLaMA-7B + zero-init + linear|62.29|43.17|52.73|36.28|76.79|45.53|52.83|
> |LLaMA-7B + zero-init + non-linear|**63.51**|**45.39**|**54.45**|**36.95**|76.67|45.04|**53.28**|
>
> From our results, we observe that except Fully Fine-tuning setting, **the Zero-Init approach combined with a non-linear prompt consistently outperforms other PEFT methods overall**. This further point out the effectiveness of our method in achieving stability compared to traditional fine-tuning techniques.
>
> [1] The Power of Scale for Parameter-Efficient Prompt Tuning. EMNLP, 2021
>
> [2] Few-Shot Parameter-Efficient Fine-Tuning is Better and Cheaper than In-Context Learning. NeurIPS, 2022
>
> [3] VeRA: Vector-based Random Matrix Adaptation. ICLR, 2024
> ### **Q3: The paper overlooks the Switch Transformer model.**
> Thanks for your comment. There seems to be a misunderstanding about the contributions of the paper. **Our main focus is to study the LLaMA-Adapter for parameter-efficiently fine-tuning the LLaMA models rather than efficiently scaling LLMs with MoE models.**
>
> In particular, we study the LLaMA-Adapter, a PEFT method for LLaMA models. Since the LLaMA models do not replace feed-forward network layers with sparse MoE layers in their Transformer architecture, we do not consider any MoE-based Transformer variants, including the Switch Transformer. Instead, we establish a connection between the zero-initialized attention and MoE models in Section 3. We show that the zero-initialized attention can be represented as an MoE model. From that perspective, we demonstrate that using the zero-initialized attention with either linear prompts or non-linear prompts is more sample efficient than using the random-initialized attention (traditional attention).
> ### **Q4: The novelty of non-linear prompts.**
> Thanks for your comment. We would like to emphasize that our main contribution is to provide a theoretical study for understanding the benefits of the zero-initialized attention over the random-initialized attention based on their connection to MoE models. To the best of our knowledge, there had not been any similar studies in the literature prior to our work.
>
> Furthermore, our analysis indicates that in addition to linear prompts in the original LLaMA-Adapter, non-linear prompts can be also optimally estimated with greater flexibility. Therefore, we perform several experiments to justify the efficacy of the LLaMA-Adapter with non-linear prompts. Although the idea of using non-linear prompts in PEFT methods may not be new, **our paper is the very first work to propose employing non-linear prompts in the LLaMA-Adapter to enhance its performance with both theoretical guarantee and empirical evidence.**

---

### Official Review · Reviewer_oCcc · 2025-03-25

**Overall Recommendation:** 3

**Summary:**

This paper investigates a specific aspect of LLaMA-Adapter, focusing on zero-initialized attention. The zero-initialized attention mechanism is not only initialized with zero values but also involves a structural modification that replaces the traditional softmax function. Instead, softmax is computed independently over two components: the input tokens  $X_l$ and the learnable adaptation prompt $P_l$, i.e., output softmax equals to $S_g=[{softmax}(S_p) \cdot {tanh}(\alpha), {softmax}(S_X)]$. The authors demonstrate that this zero-initialized attention can be interpreted as a specialized form of a mixture of experts (MoE). Building on this insight, they also prove that non-linear prompts can offer advantages over linear prompts in the context of zero-initialized attention. Extensive experiments were conducted, and the results were validated.

## update after rebuttal

Although the rebuttal responses are satisfactory, I acknowledge that the issues raised necessitate significant revisions to the manuscript’s writing and presentation. As such, I will retain my current evaluation.

**Claims And Evidence:**

While the paper presents several claims, I believe the main arguments can be summarized as follows:

**(Theoretical Claim)** Zero-initialized attention can be interpreted as a mixture of experts (MoE), and this interpretation suggests that non-linear prompts are more suitable than linear prompts.

**(Empirical Claim)** Zero-initialized attention outperforms random-initialized attention (that is traditional attention), and non-linear prompts outperform linear prompts for zero-initialized attention.

Both claims are valuable contributions to the field, and I find sufficient supporting evidence for each. As a result, I am inclined to accept the paper; however, I have some minor suggestions.

**Suggestions:**
1) Alignment of Theory and Experiments: Additionally, the proof in the paper does not directly claim that zero-initialized attention is superior to random-initialized attention. As I understand it, the primary argument is that zero-initialized attention resembles MoE. Given this, I am uncertain if the experiments in Table 1, which compare zero-initialized and random-initialized attention, are the most appropriate. Overall, the theoretical argument revolves around the similarity between zero-initialized attention and MoE, and the benefits of non-linear prompts. However, the experimental results seem to focus on comparing zero-initialized attention with random-initialized attention. I believe it would strengthen the paper if the theory and experiments were more closely aligned.
Moreover, regarding Table 1, the results presented are indeed similar to those found in the original paper's "Table 5: Effectiveness of Zero-initialized Attention in our method," which specifically covers the ScienceQA dataset. However, the performance improvement in this paper does not seem to be as significant as in LLaMA-Adapter. I am curious if there is a specific reason for this discrepancy.

2) Fragmentation of Sections: The structure of the paper feels somewhat fragmented. In particular, Sections 4 and 5 appear to share similar content and proofs. A possible reorganization could involve grouping these under a common heading, such as "Linear and Non-linear Prompts."

**Essential References Not Discussed:**

N/A

**Experimental Designs Or Analyses:**

Refer to Claims And Evidence.

**Methods And Evaluation Criteria:**

Refer to Claims And Evidence.

**Other Comments Or Suggestions:**

Refer to Claims And Evidence.

**Other Strengths And Weaknesses:**

Refer to Claims And Evidence.

**Questions For Authors:**

Refer to Claims And Evidence.

**Relation To Broader Scientific Literature:**

N/A

**Theoretical Claims:**

Although I haven't examined all the proofs, there don't seem to be any major issues.

---

> ### Author Rebuttal · Authors · 2025-03-29
>
> Thank you for your constructive feedback and insightful comments. We hope that we can address your concerns with the responses below.
>
> ### **Q1: Alignment of Theory and Experiments:**
> Thanks for your comments. We would like to clarify that the convergence analysis of prompt estimation under the random-initialized attention has been conducted in prior work (see Appendix C in [1] or Appendix A in [2]). Thus, in response to your concern, we will include the following comparison in the revision of our manuscript (below Theorem 4.2), indicating that **using the zero-initialized attention is more sample efficient than using the random-initialized attention**:
>
> *1. Prompt convergence in random-initialized attention [1,2]:* The convergence rates of prompt estimation are significantly slow, standing at the order of $O(1/\log^{\tau}(n))$ for some constant $\tau>0$, where $n$ is the sample size. Thus, to approximate prompts with a given error $\epsilon$, we need exponentially many data points $O(\exp(\epsilon^{-1/\tau}))$, which is not sample efficient.
>
> *2. Prompt convergence in zero-initialized attention (Ours):* As shown in Theorem 4.2 and Theorem 5.2 in our manuscript, the convergence rates of prompt estimation are of polynomial orders, ranging from $O(n^{-1/2})$ to $O(n^{-1/4})$. Therefore, we only need polynomially many data points to approximate the prompts with a given error $\epsilon$.
>
> **Hence, in the experiments, we conduct comparisons between *Zero-init* and *random-initialized attention* to validate the above theoretical analytic as presented in Table 1.** Therefore, we believe that our work consistently aligns with our theoretical results.
>
> [1] P. Akbarian et al. Quadratic gating functions in mixture of experts: A statistical insight. arXiv preprint, 2024.
>
> [2] M. Le et al. Mixture of experts meets prompt-based continual learning. Advances in NeurIPS, 2024.
>
> ### **Q2: Discrepancy of LLaMA-Adapter on LLM benchmarks and ScienceQA (Multi-Modal):**
> The discrepancy in performance improvement between our paper and the original study on LLaMA-Adapter can be attributed to **the differences in the datasets and task settings**. The original paper only conduct ablation study to compare Zero-initialized Attention to Random-Init setting (without the zero-initialized mechanism) on the ScienceQA dataset, which is a **multi-modal dataset used for Visual Question Answering**. In contrast, we extend more experiments on **language-only tasks**, specifically fine-tuning on the Alpaca dataset  **based on the original code of  LLaMA-Adapter paper**. The **Reviewer XuJQ** also mention that in these language-only tasks, **the performance gain in Table 1 is notable**. This **fundamental difference in task type** can influence how the zero-initialized mechanism impacts model performance.
>
> Moreover, ScienceQA consists of 21,208 questions, making model convergence slow and unstable when the zero-initialized mechanism is not applied. This issue is **highlighted in Figure 7 of the original paper**, where models **without zero initialization struggle with robustness and efficiency**. The multi-modal nature of ScienceQA, which involves both visual and textual inputs, further complicates the training process, making stabilization techniques like Zero-Init more beneficial.
>
> For language-only tasks, we fine-tuned the Random-Init setting (without the zero-initialized mechanism) on the Alpaca dataset, which contains 52,000 samples. Although the Random-Init setting led to **unrobust convergence in sample efficiency, as shown in Figure 2 and 3 in our paper**, the final convergence values when fine-tune the Random-Init setting on 100% Alpaca dataset remained **sufficiently low compared to the Zero-Init setting**. This indicates that when fine-tuning a random-initialized model on a sufficiently large language-only dataset, the model can still achieve reasonable stability despite the slower convergence.
>
> Overall, the smaller performance gap observed in our study compared to the original paper can be explained by the **difference in dataset size and modality**. While Zero-Init plays a crucial role in stabilizing multi-modal training on ScienceQA, its impact is **less pronounced in large-scale language-only fine-tuning, where instability can be mitigated to some extent by the sheer volume of training data**.
>
> ### **Q3: Fragmentation of Sections:**
> Thanks for your suggestion. We agree that both Section 4 and Section 5 focus on the convergence analysis of prompt estimation. Therefore, in the revision of our manuscript, we will merge them into one big section by relabeling Subsection 5.1 and Subsection 5.2 as Subsection 4.2 and Subsection 4.3, respectively.

---

> > ### Comment · Reviewer_oCcc · 2025-04-09
> >
> > Thank you to the authors for their detailed response.
> >
> > The concern raised in Q2 has been adequately addressed through the discussion. However, given that the prior work in question is closely connected to the core contribution of this paper, I believe the manuscript must explicitly explain why the experimental setups differ, if such differences exist. This context is essential for clarity and completeness.
> >
> > Regarding Q1, I view the clarification provided in this rebuttal as effectively the main theorem of the paper. Specifically, the statement that "zero-initialized attention is more sample efficient than random-initialized attention" should be highlighted more prominently in the theoretical sections (Sections 3–4). Doing so would help readers more clearly understand the connection between the theoretical analysis and the empirical results. As it currently stands, this connection is not sufficiently emphasized in the flow of the manuscript.
> >
> > Overall, I find the rebuttal responses satisfactory, and I believe that—if fully incorporated—the paper will be significantly strengthened. However, I also recognize that these points require substantial revisions to the writing and presentation. Therefore, I will maintain my current score.

---

> > > ### Author Response · Authors · 2025-04-09
> > >
> > > Dear Reviewer oCcc,
> > >
> > > We are glad to hear that our response addresses your concerns. In the revision of our manuscript, we will carefully incorporate your suggestions, as well as those provided by the other reviewers. If you have any further concerns, please feel free to let us know.
> > >
> > > Thank you,
> > >
> > > The Authors

---

### Decision · Program_Chairs · 2025-05-01

**Decision:**

Accept (poster)

**Comment:**

This paper studies the theoretical understanding of zero-initialized attention in LLaMA-Adapter and its connection to mixture-of-expert models, and demonstrates the statistical benefits of zero-initialized attention over conventional ones. It also argues that non-linear prompts offer greater flexibility for a broader range of applications and shows that non-linear prompts outperforms the linear ones.

The reviewers agreed upon positive ratings. Overall I recommend accept.